# Systematic assessment of ISWI subunits shows that NURF creates local accessibility for CTCF

Mario Iurlaro [1,4,5], Francesca Masoni [1,2,5], Ilya M. Flyamer[1], Christiane Wirbelauer[1], Murat Iskar [1], Lukas Burger [1,3], Luca Giorgetti [1] & Dirk Schübeler [1,2]✉

Catalytic activity of the imitation switch (ISWI) family of remodelers is critical for nucleosomal organization and DNA binding of certain transcription factors, including the insulator protein CTCF. Here we define the contribution of individual subcomplexes by deriving a panel of isogenic mouse stem cell lines, each lacking one of six ISWI accessory subunits. Individual deletions of subunits of either CERF, RSF, ACF, WICH or NoRC subcomplexes only moderately affect the chromatin landscape, while removal of the NURF-specific subunit BPTF leads to a strong reduction in chromatin accessibility and SNF2H ATPase localization around CTCF sites. This affects adjacent nucleosome occupancy and CTCF binding. At a group of sites with reduced chromatin accessibility, CTCF binding persists but cohesin occupancy is reduced, resulting in decreased insulation. These results suggest that CTCF binding can be separated from its function as an insulator in nuclear organization and identify a specific role for NURF in mediating SNF2H localization and chromatin opening at bound CTCF sites.

Chromatin remodelers can move, slide or evict nucleosomes, an essential activity for all aspects of genome regulation[1]. This includes the ability of DNA-binding factors to access the genome, which is linked to the presence and position of nucleosomes[2]. Indeed, sites bound by transcription factors (TFs) are characterized by high local accessibility, which is assumed to be required for their binding[3].

How remodeling activity is targeted to specific genomic sites has been difficult to address due to the large number of complexes and the lack of a systematic assessment of their contributions. Among these, imitation switch (ISWI) represents one of four remodeler families besides SWItch/sucrose nonfermentable (SWI/SNF), chromodomain helicase DNA-binding (CHD) and INO80 (ref. 1). ISWI members consist of one of two ATPases, SNF2L (SMARCA1) or SNF2H (SMARCA5), that alternatively associate with complex-specific accessory subunits. Main ISWI accessory subunits are RSF1, ACF1 (BAZ1A), WSTF (BAZ1B), TIP5

(BAZ2A), CECR2 and BPTF, which together with one ATPase create the RSF, ACF and CHRAC, WICH, NoRC, CERF or NURF complexes, respectively[4–12], which have been suggested to engage their substrate differently in vitro[13]. In vivo, ISWI complexes have been linked to both gene activation and repression[4,5] with NURF, RSF, CERF and ACF impacting genes transcribed by RNA polymerase II, while NoRC and WICH impact RNA polymerase I-transcribed genes[4,5]. Altogether, this suggests that ISWI complexes present distinct enzymatic abilities in vitro and are associated with distinct functions in vivo. It has been proposed that accessory subunits could differentially regulate the ATPase activity in terms of recruitment or activation at specific chromatin regions[5]; however, a systematic assessment of this model is presently missing.

Loss of SNF2H in mammalian cells causes genome-wide changes in nucleosome organization with increased nucleosomal repeat length and reduced binding of specific TFs, such as CTCF, coinciding with

[1]Friedrich Miescher Institute for Biomedical Research, Basel, Switzerland. [2]Faculty of Science, University of Basel, Basel, Switzerland. [3]Swiss Institute of Bioinformatics, Basel, Switzerland. [4]Present address: Disease Area Oncology, Novartis Biomedical Research, Basel, Switzerland. [5]These authors contributed equally: Mario Iurlaro, Francesca Masoni. ✉e-mail: dirk@fmi.ch

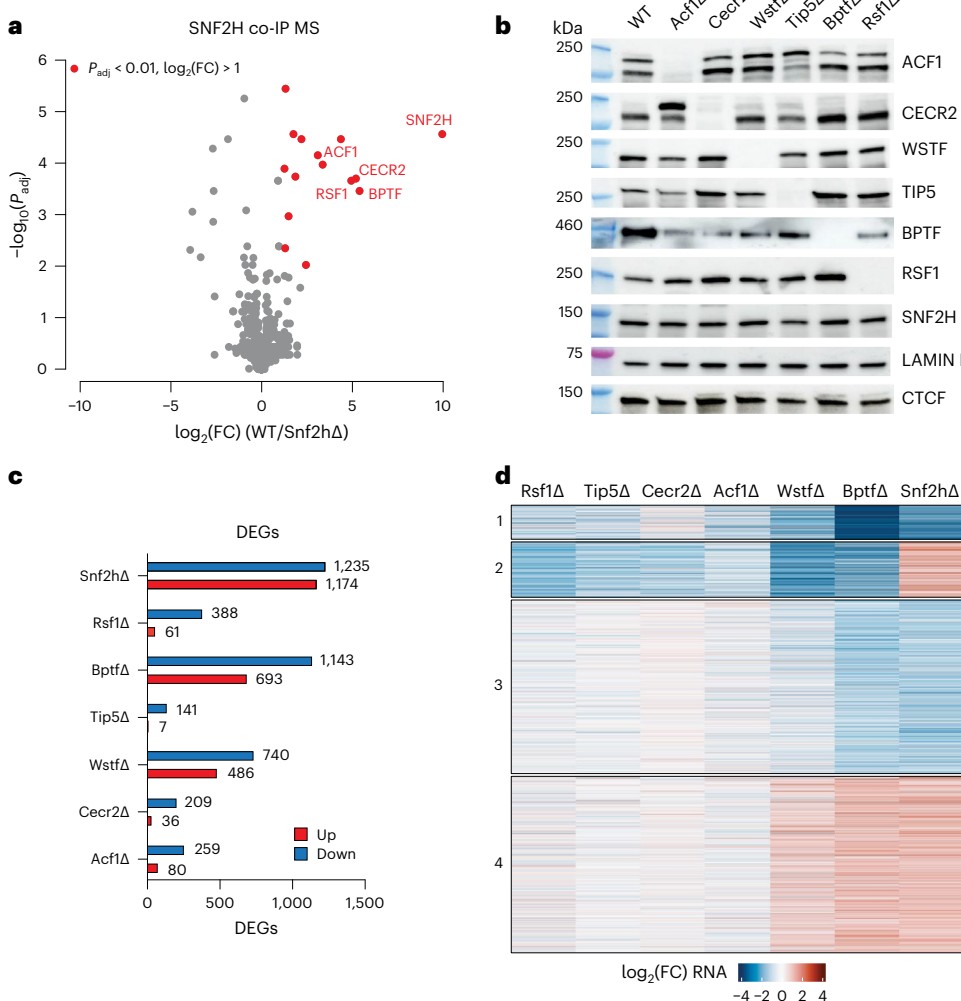

**Fig. 1 | Comprehensive deletion and transcriptome analysis of ISWI subunits in mESCs. a**, MS quantification of proteins co-immunoprecipitated using an anti-SNF2H antibody in WT and Snf2hΔ cells. Highlighted in red are proteins with $P < 0.01$ and $\log_2(FC)$ (WT/Snf2hΔ) > 1. Only the names of ISWI subunits are shown. Statistical significance was calculated using a two-sided $t$-test with Benjamini–Hochberg correction for multiple comparisons (Methods). **b**, Western blot for ISWI subunits and controls (LAMIN B, CTCF) in all deletion cell lines and WT control. While the deletions show overall no effect on other subunits, we note a change in band stoichiometry for CECR2, ACF1 and BPTF in some of the mutant lines. Blots are representative of at least two experiments. **c**, Number of DEGs (Methods) upon deletion of ISWI subunits, upregulated in red and downregulated in blue. **d**, Heatmap of DEGs shown as $\log_2(FC)$ with respect to parental cell line control. DEGs are clustered based on expression changes, with cluster numbering indicated on the left.

reduced chromatin accessibility[14,15]. Here we generated a set of isogenic mouse stem cell lines with loss-of-function mutations of ISWI accessory subunits. Molecular phenotyping of the transcriptome and the epigenetic landscape of accessibility and nucleosomal organization show that loss of either ACF1, RSF1, CECR2 or TIP5 only mildly affects nucleosomal positioning or transcription, indicative of functional redundancy between the respective subcomplexes. However, we identify a specific role for the NURF subunit BPTF in generating accessible chromatin around CTCF-bound sites. We additionally show that, at a group of sites, in the absence of chromatin opening via NURF, CTCF binding persists, while localization of cohesin and its release factor WAPL is reduced, together with a slight reduction in insulation. This suggests a mechanistic link between chromatin remodeling by NURF and CTCF structural function.

## Results

### Generation of isogenic lines with individual ISWI deletions

Mammalian ISWI subcomplexes have been considered mutually independent, with each consisting of either SNF2H or SNF2L ATPase together with one or more noncatalytic subunits[5]. We previously

showed a critical role for SNF2H in mouse embryonic stem cells (mESCs)[14], where it has considerably higher expression than SNF2L (Extended Data Fig. 1a). Loss of SNF2L in mESCs indeed results in negligible transcriptional (Extended Data Fig. 1b–d) and genome-wide accessibility changes, including at CTCF-binding sites (Extended Data Fig. 1e,f), letting us focus on SNF2H-containing complexes. We first determined their protein composition by co-immunoprecipitation (co-IP) under native conditions using the catalytic subunit SNF2H as bait and quantified associated proteins by mass spectrometry (MS). This identified interactions with previously reported ISWI accessory subunits (Fig. 1a). These include CECR2 and BPTF, which had initially been suggested to engage with SNF2L but were also recently reported to interact with human SNF2H[16,17] (Fig. 1a). In parallel, we also performed SNF2H co-IP followed by western blot detection for ISWI accessory subunits. This further validates previously reported interactions of SNF2H with WSTF and BAZ2A/TIP5 subunits (wild-type (WT) lane in Extended Data Fig. 2a). While we detected BPTF interaction by MS, this proved difficult to validate via western blotting due to the large size of the BPTF protein (predicted to be higher than 300 kDa; Fig. 1b and Extended Data Fig. 2a). Taken together, these experiments suggest

that in mESCs, SNF2H not only interacts with its canonical partners (RSF1, TIP5, WSTF and ACF1) but also with CECR2 and BPTF (Fig. 1a and Extended Data Fig. 2a).

Having established the set of SNF2H-containing complexes present in mESCs, we generated isogenic loss-of-function mutants using CRISPR–Cas9 for six noncatalytic subunits that distinguish mammalian ISWI subcomplexes, namely *Wstf/Baz1b* for WICH (WstfΔ), *Baz2a/Tip5* for NoRC (Tip5Δ), *Acf1/Baz1a* for the ACF and CHRAC complexes (Acf1Δ), *Rsf1* for RSF (Rsf1Δ), *Bptf* for NURF (BptfΔ) and *Cecr2* for CERF (Cecr2Δ). All deletion cell lines show no detectable protein by western blot, display self-renewal and express pluripotency markers comparable to WT cells (Fig. 1b and Extended Data Fig. 2b). Next, we asked whether loss of one subunit would affect the formation of other subcomplexes by performing SNF2H co-IP followed by western blot detection of tested ISWI subunits. As expected, SNF2H co-IP failed to co-immunoprecipitate the deleted subunits (Extended Data Fig. 2a), while other complexes are detected with the abovementioned exception of BPTF, arguing that any molecular phenotype is subcomplex-specific (Extended Data Fig. 2a).

To determine the effects on the transcriptome, we performed RNA-seq in each of the generated mutant lines. Here Tip5Δ, Cecr2Δ, Acf1Δ and Rsf1Δ show moderate transcriptional phenotypes (with 148, 245, 339 and 449 misregulated genes, respectively) in contrast with WstfΔ and BptfΔ, with 1,226 and 1,836 misregulated genes (Fig. 1c and Extended Data Fig. 2c). Notably, none of the accessory subunit deletions fully recapitulate the transcriptional profile resulting from loss of SNF2H. However, loss of the NURF component BPTF causes transcriptional changes that substantially overlap with those of Snf2hΔ (Extended Data Fig. 2d–f). When inquiring about the chromatin state of the respective promoters using chromHMM[18,19], we observe over-representation of promoters classified as bivalent among affected genes in all deletion lines (Supplementary Fig. 1). Grouping affected genes further (Fig. 1d) identifies two large clusters that change similarly in BptfΔ and Snf2hΔ (Fig. 1d, clusters 3 and 4) but are not enriched for particular gene ontology terms (Supplementary Fig. 2a,b), while differentially expressed genes (DEGs) in Rsf1Δ, Cecr2Δ and Tip5Δ are involved in similar biological processes (Supplementary Fig. 2a,b). Taken together, the transcriptional responses suggest redundancy among several ISWI subcomplexes under the tested culture conditions[20] yet point to a larger and distinct role for NURF.

**Absence of BPTF causes changes in nucleosome organization**
Because the absence of SNF2H causes a global reduction in nucleosome phasing, with a coinciding increase in nucleosome repeat length (NRL) by ~9 to 10 bp (refs. 14,15), we asked if this could be assigned to individual subcomplexes. Micrococcal-nuclease sequencing (MNase-seq) showed that average NRL is largely unaffected in our deletion lines (Fig. 2a). Similarly to the Snf2hΔ cells[14], nucleosomal phasing at transcription start sites (TSSs) is unaffected by loss of individual ISWI subcomplexes (Fig. 2b and Extended Data Fig. 3a). At distal DNaseI hypersensitive sites (DHSs; Supplementary Table 1), deletion of ISWI subunits had little to no effect, with the exception of BPTF (Fig. 2c and Extended Data Fig. 3b). Loss of this NURF subunit causes increased nucleosomal signal over distal DHSs and coinciding reduction in phasing at the flanking regions (Fig. 2c and Extended Data Fig. 3b).

Because a large fraction of distal DHSs in mammalian genomes are bound by CTCF (in our dataset 34%), we repeated this analysis with a focus on bound CTCF sites, as determined by chromatin immunoprecipitation followed by sequencing (ChIP–seq). This shows that the reduction of nucleosomal phasing observed only in BptfΔ cells is indeed concentrated at distal DHSs bound by CTCF (Extended Data Fig. 3c–k), mirroring our previous observation upon loss of SNF2H[14].

Next, we asked how the regulatory landscape, as defined by open chromatin, is affected by performing an assay for transposase-accessible chromatin using sequencing (ATAC–seq)[21]. *K*-means

clustering of the resulting dataset upon loss of each ISWI subunit categorized open chromatin regions (ATAC–seq peaks) into three clusters (Supplementary Table 2). This shows that chromatin accessibility is largely unaltered in Cecr2Δ, Acf1Δ, Rsf1Δ or Tip5Δ, in line with the observed limited transcriptional response (Fig. 2d). Again, BptfΔ displays major changes, which extensively overlap with those in Snf2hΔ (Fig. 2d and Extended Data Fig. 4a).

Regions with reduced accessibility in both BptfΔ and Snf2hΔ (cluster 2 in Fig. 2d) are highly enriched for CTCF binding, compared to any other cluster or functional annotation (Extended Data Fig. 4b). In fact, 47.5% of cluster 2 regions are bound by CTCF, whereas clusters 1 and 3 show binding in less than 1% of their regions. Indeed, the CTCF motif is strongly enriched in cluster 2 regions, suggesting that loss of accessibility driven by deletion of either BPTF or SNF2H mostly localizes to CTCF-bound sites (Fig. 2e,f and Extended Data Fig. 4c,d). Taken together, genome-wide accessibility maps identify only modest effects upon loss of subunits of NoRC, RSF, CERF and ACF complexes in mESCs, while loss of BPTF causes large changes at CTCF sites that resemble the phenotype observed in Snf2hΔ. Indeed, upon loss of either BPTF or SNF2H, more than 85% of bound CTCF sites show a reduction in accessibility.

To determine if this reflects the local presence of remodeler activity, we measured genome-wide binding of BPTF and SNF2H. Indeed, BPTF is readily detected at CTCF sites regardless of whether these reside distal or proximal to promoters (Extended Data Fig. 5a,b). SNF2H is similarly present at CTCF sites, but this enrichment is dependent on BPTF, as it is lost in BptfΔ cells (Fig. 2g,h and Extended Data Fig. 5c–e). Notably, the absence of BPTF does not alter SNF2H localization at other distal regulatory regions (Extended Data Fig. 5f,g). Altogether, BPTF localization to CTCF sites and the reduction in SNF2H binding upon BPTF depletion mirror the observed accessibility changes and argue that the NURF remodeling complex mediates SNF2H localization and resulting accessibility at CTCF sites (Fig. 2g,h and Extended Data Fig. 5a–e).

**CTCF sites largely remain bound in the absence of BPTF**
Reduced accessibility upon remodeler deletions has thus far been reported to coincide with and reflect loss of TF binding[14,22,23]. This was also the case for CTCF upon loss of SNF2H[14] and is similarly expected to be the case in the absence of BPTF.

To test this, we performed ChIP–seq for CTCF in BptfΔ cells. Unexpectedly, most sites with a strong reduction in accessibility remain either bound by CTCF or display only a minor reduction in binding when compared with the drastic loss in Snf2hΔ (Extended Data Fig. 6a,b). This is evident from the average CTCF ChIP–seq profile at bound sites, which shows a stronger loss in Snf2hΔ versus BptfΔ (Fig. 3a), but also at individual sites where CTCF binding is consistently lost in Snf2hΔ with a weaker or absent change in BptfΔ (Fig. 3b). This indicates that loss of the NURF component BPTF results in a state distinct from complete loss of ISWI function, where CTCF binding can persist, but chromatin opening at the same sites is strongly impaired.

**Footprinting confirms persistent CTCF binding upon loss of BPTF**
Thus far, we have used enrichment-based methods (ChIP–seq and ATAC–seq) to identify the role of BPTF at sites of CTCF binding. This leaves the possibility that the observed disconnect between binding and accessibility could be confounded by the differential sensitivities of these assays. Testing of this possibility requires an orthologous approach that does not rely on enrichment and that simultaneously detects TF binding and nucleosomal organization. To this end, we performed amplicon-based single-molecule footprinting (SMF; or NOMe-seq) that allows single-molecule and base-resolution footprinting of chromatin-bound proteins[24–26]. With this technique, we tested selected loci in WT and BptfΔ cells. The resulting footprinting

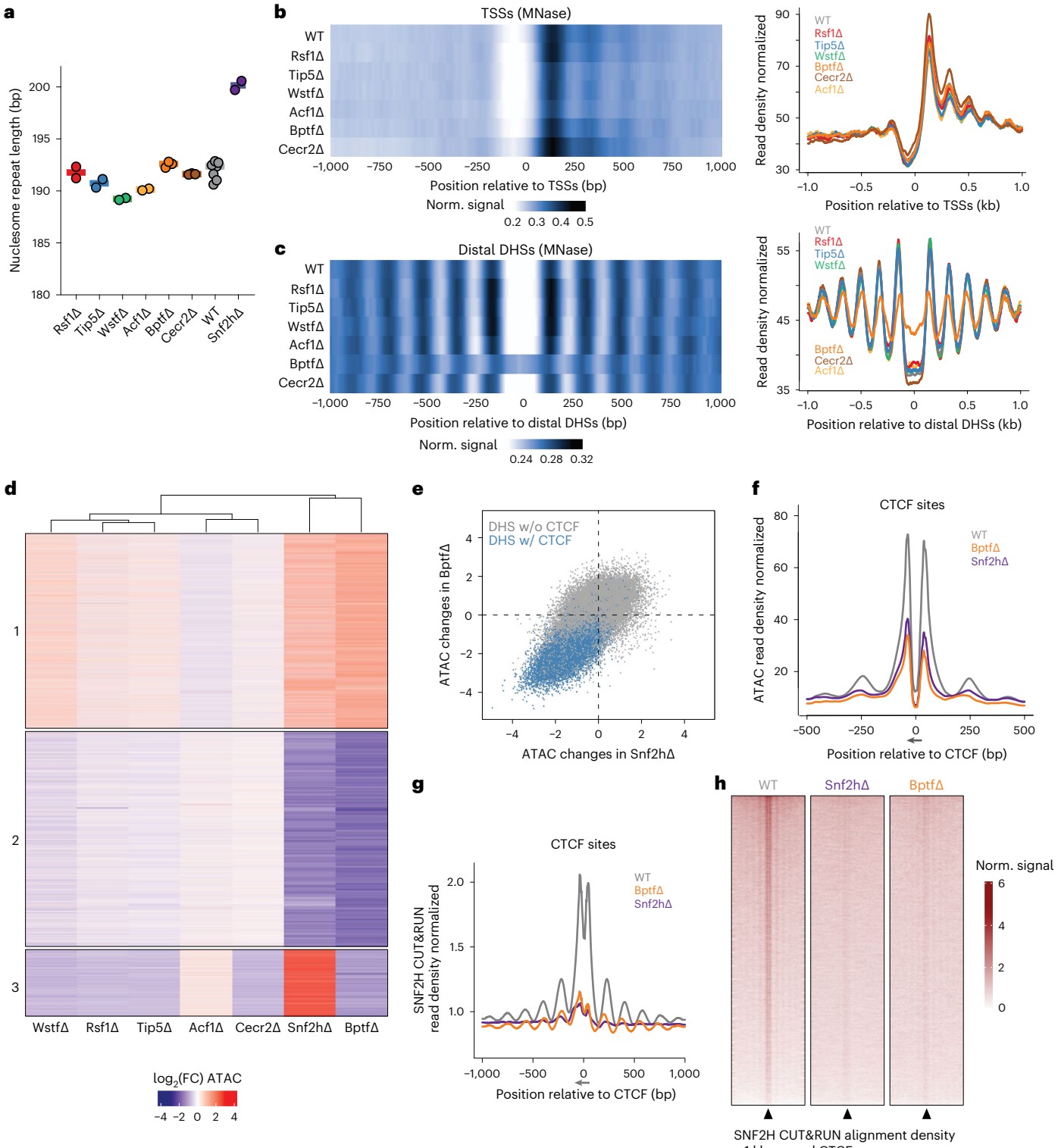

**Fig. 2 | Genome-wide nucleosome position and accessibility profiling identifies subcomplex-specific chromatin functions. a**, Average NRL for each generated deletion cell line, for SNF2H deleted line (data from ref. 14) and parental line control, as measured by MNase-seq. Shaded bar represents median value of multiple replicates. **b**,**c**, Average nucleosomal profile at TSSs (**b**) or distal DHSs (**c**) shown as heatmap (left) and profile plot (right). **d**, Heatmap displaying log$_2$(FC) of ATAC–seq signal at differentially accessible regions for each deletion cell line with respect to parental line control. Regions are clustered based on accessibility changes. Each cluster contains 36,386, 40,166 and 12,426 sites, respectively, for clusters 1, 2 and 3. Cluster numbers are reported on the left.

**e**, Quantitative comparison of chromatin accessibility changes (log$_2$(FC) over parental control) in Snf2hΔ (*x* axis) and BptfΔ (*y* axis) at DHSs. DHSs containing a CTCF motif are highlighted in blue. **f**, Average ATAC–seq signal at bound CTCF motifs in WT control (gray), BptfΔ (orange) and Snf2hΔ (purple) cells. Canonical motif orientation (5′–3′) indicated by the arrow. **g**, Average SNF2H CUT&RUN signal at bound CTCF motifs (as in **f**) in WT control (gray), BptfΔ (orange) and Snf2hΔ (purple) cells. Canonical motif orientation (5′–3′) indicated by the arrow. **h**, SNF2H CUT&RUN alignment densities in WT, Snf2hΔ and BptfΔ cells, centered on CTCF-bound motifs (black arrowheads).

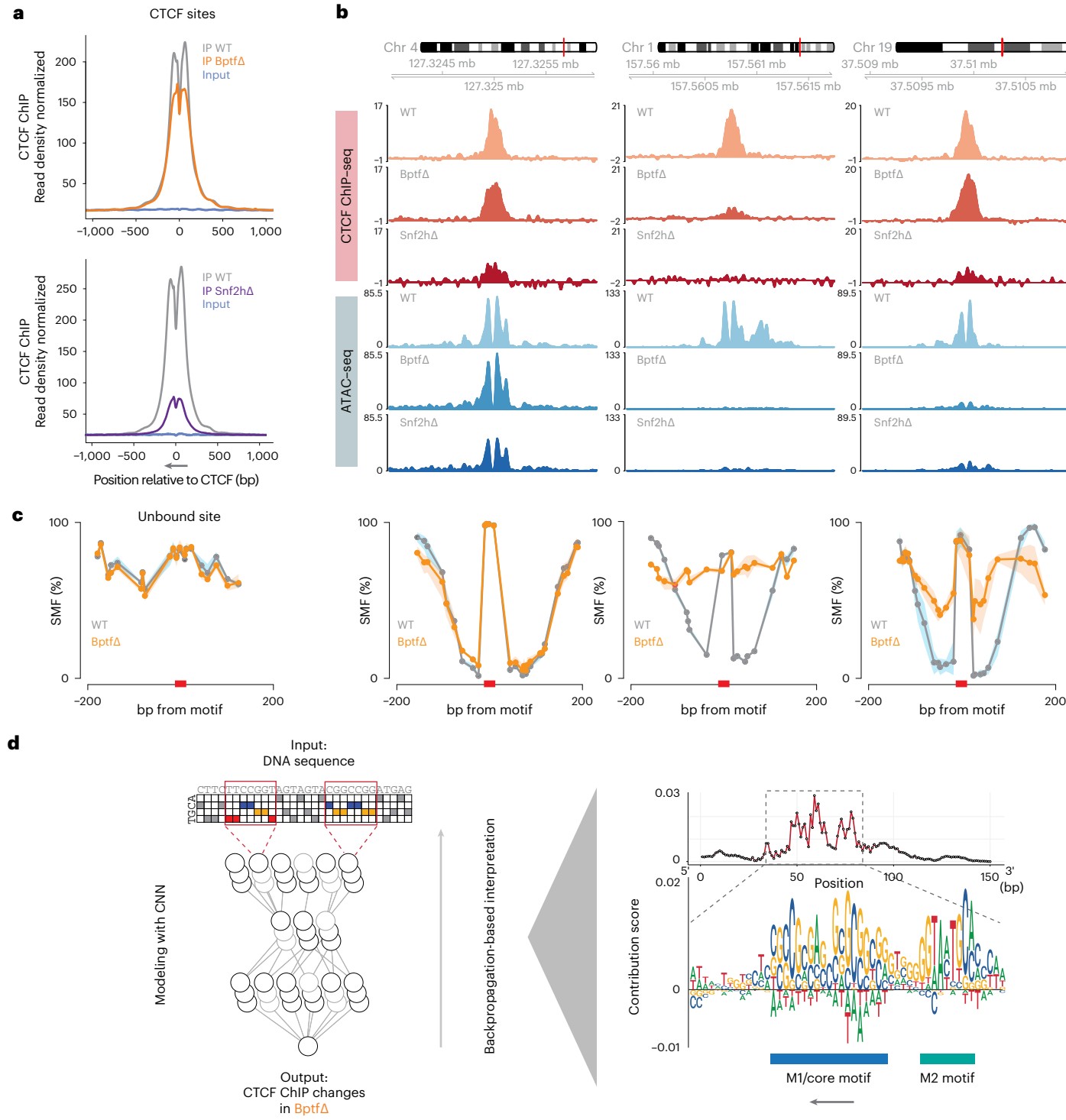

**Fig. 3 | CTCF binding at strong motifs largely persists in the absence of BPTF despite loss of accessibility. a**, Average CTCF ChIP–seq signal at bound CTCF sites in BptfΔ (orange) and parental ES cells (gray; top). The same analysis is in Snf2hΔ (purple) and parental ES cells (gray; bottom; data from ref. 14). Inputs are shown as control (blue). Canonical motif orientation (5′–3′) indicated by the arrow. **b**, Representative genomic loci illustrating changes in CTCF binding (ChIP–seq indicated by shades of red) and chromatin accessibility (ATAC–seq indicated by shades of blue) in BptfΔ, Snf2hΔ and parental ES cells. **c**, Average SMF signal at an unbound site (left) and at the same sites (as in **b**; right), in BPTF-deleted cells (orange) and WT control (gray). Shaded line represents s.d. **d**, A CNN-based model used to predict changes in CTCF binding in BptfΔ cells. Influence of particular nucleotides is shown as average contribution scores, highlighting the role of an extended CTCF motif (M1 and M2) in retaining binding in the absence of BPTF. Canonical motif orientation (5′–3′) indicated by the arrow.

patterns at the tested locations show that sites with unchanged CTCF ChIP–seq and ATAC–seq signals in BptfΔ also show a footprint identical to WT cells (Fig. 3b,c, second panel from the left). Loss of CTCF binding in BptfΔ as measured by ChIP is reflected in a flattening of average footprinting signal, similar to the signal obtained at unbound sites (Fig. 3c, third and first panels from the left). Crucially, sites that maintain CTCF binding as measured by ChIP but lose accessibility maintain a clear footprint of similar height over the CTCF motif in BptfΔ,

suggesting the same factor occupancy as in WT cells. Adjacent flanking regions, however, show lower methylation reflecting the reduced accessibility for the footprinting enzyme adjacent to CTCF, which was the case for multiple tested loci (Fig. 3c, fourth panel from the left, and Extended Data Fig. 6c,d) mirroring the accessibility reduction in ATAC−seq (Fig. 3b and Extended Data Fig. 6c). If this reflects increased nucleosome occupancy, it should result in higher abundance of longer MNase fragments over bound CTCF sites. Indeed, analysis of MNase-seq reads as a function of fragment size shows a clear accumulation of longer (>200 bp) fragments that span bound motifs but only in BptfΔ (Extended Data Fig. 6e). Taken together, without BPTF, binding of CTCF largely persists; however, at a class of sites, it is not accompanied by canonical highly accessible chromatin.

### An extended motif is a feature of persistent CTCF sites
Next, we asked how persistent CTCF binding upon loss of NURF relates to initial binding strength or DNA sequence. Interestingly, CTCF ChIP−seq signal in WT cells did not correlate with loss of CTCF binding in BptfΔ (Extended Data Fig. 7a). Motif strength, however, showed a weak yet noticeable trend, in that CTCF sites with better motif scores tend to retain more CTCF binding in BptfΔ (Extended Data Fig. 7b). To move beyond prior motif definitions, we applied a convolutional neural network (CNN) approach similar to DeepSTARR (Methods; ref. 27; Fig. 3d) and took 150 bp of DNA sequence around CTCF motifs as independent variable and response in CTCF ChIP−seq signal in BptfΔ cells as dependent variable. The resulting model shows relatively high predictive power for the CTCF ChIP-response in BptfΔ (Extended Data Fig. 7c,d, Pearson correlation coefficient = 0.43 for observed versus predicted binding in the test set). As expected from the initial trend described above, the deep learning approach identifies that the canonical CTCF motif has predictive value explaining the persistent CTCF binding upon BPTF depletion (Fig. 3d). In addition, this approach revealed the contribution of an additional 9-nucleotide (nt) stretch located ~21 bp from the center of the canonical CTCF motif (Fig. 3d and Extended Data Fig. 7e). This sequence corresponds to an additional CTCF motif component (M2), reported at a minority of CTCF sites and first discovered in DNAseI datasets[28,29]. M2-containing CTCF sites have been previously associated with highly conserved binding events that also tend to be less sensitive to CTCF protein knockdown[29].

Here the deep learning approach identified the contribution of both M1 and M2 motifs to the persistent CTCF binding in BPTF-depleted cells. Of note, most CTCF sites do not harbor an M2 motif. While this is the case for the entirety of CTCF sites, we observe that regions containing an M2 sequence are relatively enriched for sites characterized by persistent binding in BptfΔ cells (Extended Data Fig. 7f).

Combined, these findings suggest that not only strong canonical motifs but also the presence of additional motif components, likely contributing to protein−DNA affinity, can reduce dependence on remodeling activity for binding.

### Absence of BPTF impacts long-range chromatin interactions
To further explore the relationship between chromatin accessibility and CTCF binding, we clustered CTCF sites based on changes in accessibility and binding in BptfΔ. Within each cluster, we compared the average ATAC−seq and CTCF ChIP−seq responses to the loss of BPTF or SNF2H (Fig. 4a, Extended Data Fig. 8a,b and Supplementary Table 3). This analysis again illustrates a much stronger reduction of CTCF binding in Snf2hΔ versus BptfΔ within all clusters, while reduced accessibility is similar in both mutants (Fig. 4a). Notably, the resulting clusters showed distinct patterns of chromatin states as previously defined by the combinations of histone marks and TF binding in mESCs[18,19] (Fig. 4b). The cluster consisting of sites with limited accessibility changes in both mutants is enriched for promoter and enhancer states (Fig. 4a,b; Fisher's exact P value < $2.2 \times 10^{-16}$). These CTCF sites

reside within regulatory regions, suggesting that in this context accessibility is maintained by other TFs.

Next, we asked if the CTCF insulator function is affected in BptfΔ, as we previously reported upon deletion of SNF2H[14]. BPTF loss led to small but noticeable changes in physical insulation scores that, however, are less pronounced than in Snf2hΔ (ref. 14) or upon CTCF degradation[30] (Extended Data Fig. 8c−e). When investigating the potential impact on topological-associated domains (TADs)[31], we observe a reduction in average contact enrichment at TAD edges in BptfΔ (Fig. 4c and Extended Data Fig. 8f). Again, this effect was smaller compared to changes in Snf2hΔ (ref. 14) or the loss of TADs after CTCF degradation[30]. Albeit modest, this effect is specific to BptfΔ as it is not detected when comparing biological replicates (Fig. 4c). Genome-wide analysis of contact enrichment around TAD boundaries (Fig. 4d and Extended Data Fig. 8g) and loops (Fig. 4e and Extended Data Fig. 8h) showed similarly BptfΔ-specific reduction of insulation and minor changes in loop formation. To relate these observed changes in 3D genome organization to CTCF binding and accessibility in BptfΔ, we calculated changes in contact frequency separately for the five clusters. CTCF sites losing both accessibility and binding showed the strongest loss of insulation upon BPTF or SNF2H deletion (Fig. 4f and Extended Data Fig. 9a). We can exclude that these changes occur only indirectly via impacting enhancer function as they are observed at CTCF sites regardless of whether they reside within or outside enhancers (Extended Data Fig. 9b)[32,33].

### Insulation is also reduced at sites with persistent binding
The limited reduction in CTCF binding detected in BptfΔ cells raises the possibility that reduced accessibility around bound CTCF sites could contribute to the observed modest impairment of insulator function. If this is indeed the case, there should be a class of sites with persistent binding upon loss of BPTF but reduced insulation.

To test this, we focused on those 15,865 CTCF sites with only very limited variation in binding (less than ±20% variation in binding upon BPTF deletion), grouped these according to their changes in accessibility (Fig. 5a, Extended Data Fig. 10a and Supplementary Table 4) and calculated changes in 3D contact frequencies within these groups (Fig. 5b and Extended Data Fig. 10b). This identifies a group of sites where stronger loss of accessibility corresponds to loss of insulation despite persistent binding, which is evident genome-wide (Fig. 5b and Extended Data Fig. 10b) and at individual loci (Extended Data Fig. 10c,d). Because physical insulation at CTCF sites arises from the stalling of loop-extruding cohesin complexes[34], we asked if reduced insulation corresponded to reduced cohesin enrichment. ChIP−seq for the cohesin subunit RAD21 showed a progressive decrease at CTCF sites proportional to the reduction in accessibility, consistent with the observed modest reduction in insulation (Fig. 5c (top)). This behavior is not restricted to cohesin but could be similarly observed for its unloading factor WAPL (Fig. 5c (bottom)). Notably, this reduction does not coincide with the reduced expression of cohesin subunits (Extended Data Fig. 10e,f).

We conclude that proper insulator function at this class of CTCF sites, at least in part, depends not only on the presence of bound CTCF but also on chromatin opening mediated by NURF. The BptfΔ phenotype further shows that CTCF binding does not necessarily require nor is sufficient for chromatin opening, suggesting a partial separation of CTCF binding from its function as an insulator.

## Discussion
Our systematic analysis identifies a specific function for the NURF component BPTF in creating accessibility at CTCF sites, which we show for a class of regions to reduce insulation despite persistent CTCF binding. Furthermore, we show that, individually, none of the tested ISWI accessory subunits account for the changes in NRL observed when depleting SNF2H[14,15]. This argues for redundancy between subcomplexes in

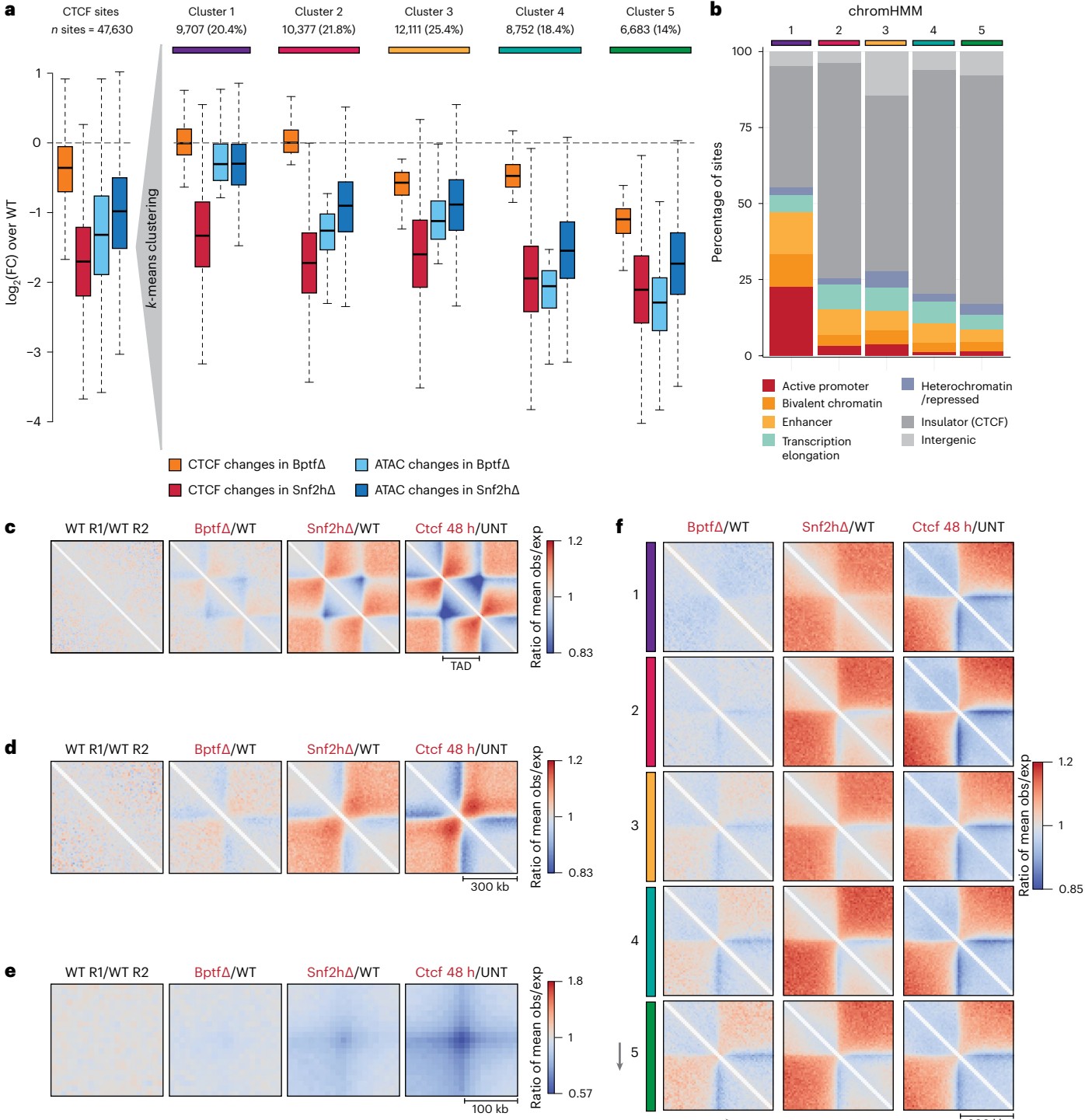

**Fig. 4 | Unsupervised clustering highlights differential responses to the absence of BPTF at the level of CTCF binding, chromatin opening and nuclear organization. a**, Boxplot displaying changes upon loss of BPTF or SNF2H in CTCF ChIP–seq and ATAC–seq signal at all bound CTCF sites, expressed as $\log_2(FC)$ in respect to WT control cells. Measurements are shown for all sites (left) and separately for clusters 1–5. Black lines indicate median, boxes indicate first and third quartiles, and whiskers indicate maximum and minimum values of distribution after the removal of outliers. **b**, Distribution of chromatin states at sites surrounding bound CTCF motifs, as labeled by chromHMM (Methods) and split by clusters (as in **a**). Cluster numbers reported on top. **c**, Changes in observed/expected interactions at TADs (identified in ref. 46 mESCs dataset) following BPTF depletion, SNF2H depletion (data from ref. 14) and CTCF auxin-mediated degradation for 48 h (data from ref. 30), measured using Hi-C (ratios over respective controls are reported), at 10 kb resolution. **d**, Same analysis as in **c** at TAD boundaries (identified in ref. 46 mESCs dataset). **e**, Same analysis as in **c** and **d** at Hi-C loops. **f**, Changes in observed/expected interactions at CTCF sites divided by clusters (as in **a**) following BPTF depletion, SNF2H depletion and CTCF auxin-mediated degradation (48 h), measured using Hi-C (ratios over respective controls are reported). Cluster number is reported on the left. Canonical motif orientation (5'–3') indicated by the arrow.

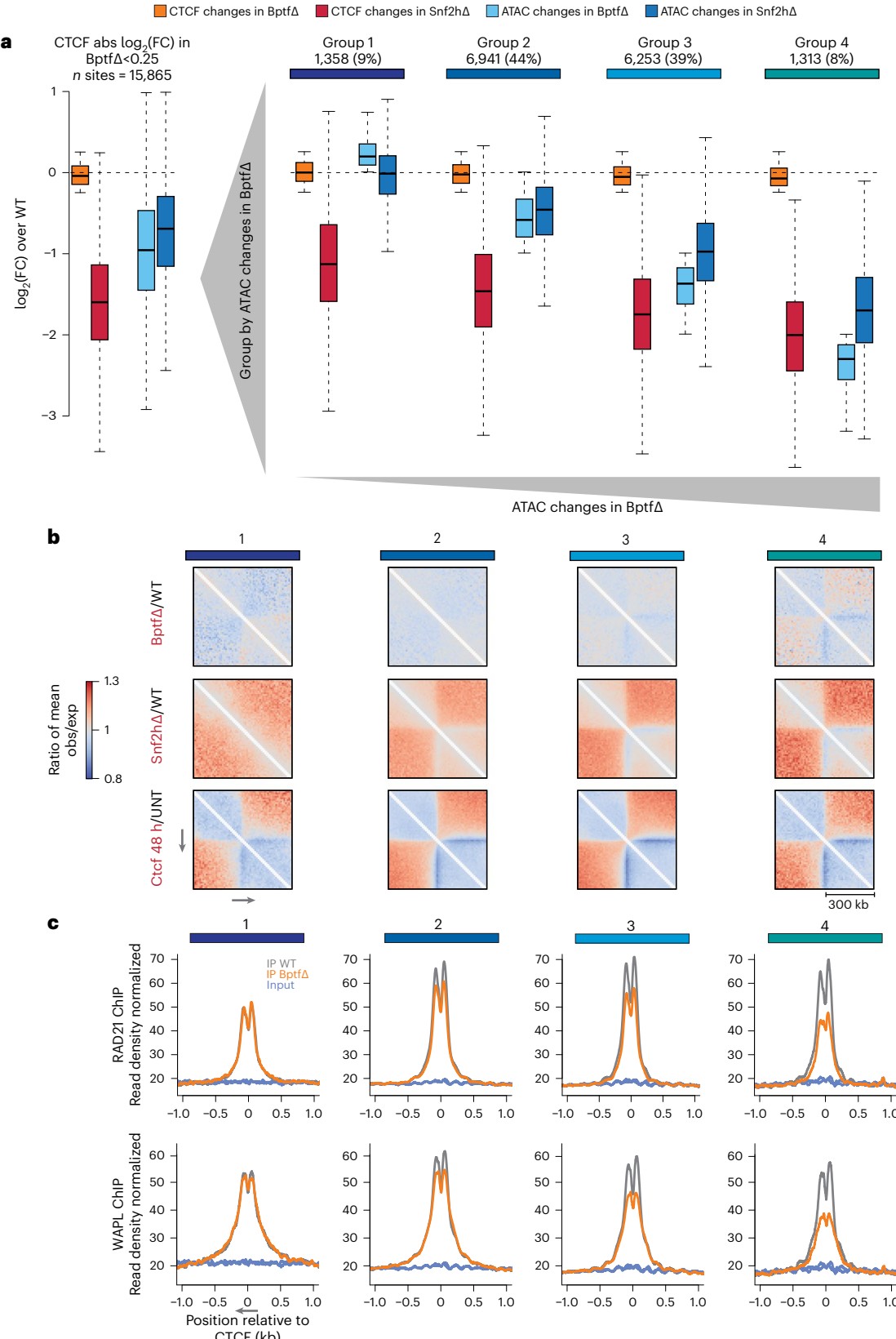

**Fig. 5 | BPTF-dependent accessibility impacts nuclear organization, localization of cohesin and the cohesin-release factor WAPL despite persistent CTCF binding. a**, Boxplot (as in Fig. 4a) summarizing changes in CTCF ChIP–seq and ATAC–seq signal for CTCF sites that retain binding upon loss of BPTF, expressed as log₂(FC) in respect to WT control cells. Measurements are shown for all sites that retain binding (left) and divided into groups 1–4 based on their accessibility changes upon loss of BPTF. **b**, Changes in observed/ expected interactions at CTCF sites not changing in binding divided by group (as in **a**) following BPTF depletion, SNF2H depletion and CTCF auxin-mediated degradation (48 h), measured using Hi-C (ratios over respective controls are reported), at 10 kb resolution. Canonical motif orientation (5′–3′) indicated by the arrow. **c**, Average ChIP–seq signal for RAD21 (top) and WAPL (bottom), at CTCF sites that retain binding, grouped by changes in chromatin accessibility (as in **a**). Canonical motif orientation (5′–3′) indicated by the arrow.

regulating nucleosome distance and appears compatible with the fact that ACF, RSF and WICH mammalian complexes are all able to space nucleosomes in vitro[6–8].

The phenotype observed upon deleting the NURF subunit BPTF is unexpected as it only partially recapitulates the SNF2H loss-of-function phenotype previously observed at CTCF sites[14,15,35], in that BPTF is required for maintaining these sites in an open chromatin state. This argues for a specific and local function for NURF catalyzed by SNF2H, which we show localizes to CTCF sites in a BPTF-dependent manner, in line with their biochemical interaction. In contrast to the phenotype observed upon loss of SNF2H, in the absence of BPTF, CTCF binding largely persists, and accordingly, we identify less severe changes in nuclear organization. These genome-wide observations in a stable and clonal genetic deletion are compatible with previous findings in different mouse and human cellular models at a subset of individual CTCF sites[36], with the phenotype displayed upon depletion of BPTF in leukemic cells[17], as well as with the reported nucleosomal profiles upon siRNA depletion of different ISWI accessory subunits[35].

DNA sequence seems to have a critical role in retaining CTCF binding in the absence of NURF, as indicated by the enrichment in persistent CTCF sites of an additional motif component previously described as the 'M2 motif' that extends the canonical M1 motif of CTCF[29]. This additional sequence component has been proposed to interact with the CTCF zinc fingers 9–11 (refs. [37,38]). CTCF sites containing this motif were shown to be more resistant to reduced CTCF levels in human cells, suggesting higher DNA–protein affinity at these regions[29], which we speculate will lead to more stable CTCF binding and a reduced requirement for remodeling activity.

The difference in phenotype between Snf2hΔ and BptfΔ suggests that ISWI-dependent activity other than NURF also contributes to CTCF binding. The limited phenotype upon deletion argues against the contribution of the homologous ATPase SNF2L in our model system. Because preferential SNF2H occupancy at CTCF sites requires NURF, we can only speculate that unspecific nucleosome mobilization by other SNF2H-containing ISWI subcomplexes might create binding opportunities.

Several scenarios have been proposed about how TF binding and chromatin opening are linked[39–41]. These involve unspecific remodeler activity that is stabilized by TF binding or TF-dependent recruitment of remodeler activity. In the case of CTCF, our findings let us propose a model that envisages both scenarios: unspecific nucleosomal mobility mediated by different SNF2H-containing complexes enables CTCF binding, while specific recruitment of NURF is required for chromatin opening at these regions.

The separation of binding from chromatin opening at a subset of sites in BptfΔ highlighted that, at these sites, CTCF binding alone is not necessarily sufficient for complete insulation, which instead appears to require chromatin opening. Because this coincides with partial loss of binding of cohesin and as a result of its release factor WAPL, it is tempting to speculate that proper cohesin accumulation at CTCF sites requires chromatin opening mediated by NURF. This model might also apply to humans and *Drosophila* where BPTF contributes to insulation[17,42,43].

Since its first description, open chromatin has become an established hallmark of active regulatory regions[44]. It is so canonical that it is successfully used for their comprehensive genome-wide detection in any given cell type[39,41]. Several studies have shown that chromatin opening relies on the activity of very different remodeler families, including ISWI and SWI/SNF, and occurs in a TF-specific fashion[14,22,23,45]. In the reported cases, removal or inhibition of remodeler activity caused reduced binding and accessibility around selected TF sites. However, we are not aware of an example where the absence of a cofactor causes loss of chromatin opening, yet displays largely persistent TF binding as we show here for CTCF in the absence of BPTF.

Our observation at a group of CTCF regions where chromatin opening contributes to complete insulation independently of CTCF binding adds a previously underappreciated variable to the set of requirements for proper nuclear organization. More generally, it is compatible with a model where local chromatin opening enables or assists cofactor interaction for TFs, potentially creating the necessary space and flexibility for proper regulatory function. If true, similar phenotypes might be identified with other TFs either by mutating cofactors as presented here or by mutating cofactor interacting domains of TFs. Ultimately, this should shed further light on how chromatin structure and nucleosome mobility influence TF function beyond DNA binding.

## Online content

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

## Methods

This study complies with all relevant ethical regulations and did not require any approval.

### Cell culture

WT mESCs and derived genetically deleted lines of 129S6/SvEvTac background were maintained as previously described[23]. Briefly, cells were maintained in Dulbecco's modified Eagle's medium (DMEM; Invitrogen), supplemented with 15% fetal calf serum (Invitrogen), GlutaMAX Supplement (Gibco), nonessential amino acids (Gibco), β-mercaptoethanol (Sigma) and leukemia inhibitory factor (produced in house). Cells were grown on plates coated with 0.2% gelatin (Sigma).

### Cell line generation and maintenance

All deleted cell lines were generated starting from mESC lines of 129S6/SvEvTac background using the CRISPR–Cas9 protocols previously described, with modifications[47]. Briefly, $1 \times 10^6$ of mouse ES cells were cotransfected (Lipofectamine 3000; Thermo Fisher Scientific, L3000008) with a pC2P plasmid containing the Cas9-P2A-puromycin cassette and expressing the sgRNA targeting the gene of interest. For each transfection, 125 µl of Opti-MEM (Opti-MEM reduced serum medium; Gibco) and 7.5 µl Lipofectamine 3000 (Lipofectamine 3000; Thermo Fisher Scientific) were combined to a mix containing 125 µl Opti-MEM and 3 µl of p3000 reagent (Lipofectamine 3000; Thermo Fisher Scientific) and 1 µg of plasmid. The resulting mix was incubated for at least 15 min at room temperature and added to cells seeded in a six-well plate while still in suspension. Puromycin selection (2 µg ml$^{-1}$) was carried out 1 day after transfection for 24 h. Following a 2-day recovery, resistant cells were diluted to isolate single clones ($1 \times 10^4$ cells per 15 cm plate). The deriving colonies were manually picked, expanded and validated by western blot and DNA sequencing. One representative clone was then selected for each genotype. Sequence of gRNAs together with sequence of mutated alleles are reported in Supplementary Table 5 (*Acf1* was targeted twice to eliminate a second alternative isoform).

### Nuclear protein extraction for western blotting and co-IP

For nuclear cell lysis, $5 \times 10^6$ cells per condition were resuspended in 1 ml of lysis buffer + protease inhibitor (cOmplete; Roche, 000000011873580001; 10 mM Tris–HCl (pH 7.4), 10 mM NaCl, 3 mM MgCl$_2$, 0.1 mM EDTA and 0.5% NP-40) and incubated on ice for 10 min. Samples were then centrifuged for 5 min at 845$g$ at 4 °C and resuspended in 250 µl of wash buffer + protease inhibitor (10 mM Tris–HCl (pH 7.4), 10 mM NaCl, 3 mM MgCl$_2$ and 0.1 mM EDTA). Samples were then directly centrifuged for 5 min at 845$g$ at 4 °C. Resulting nuclei were resuspended in 250 µl of RIPA buffer + protease inhibitor (50 mM Tris–HCl (pH 8.0), 150 mM NaCl, 1% NP-40, 1% Na-deoxycholate, 0.1% SDS and 1 mM dithiothreitol (DTT)), mixed briefly through vortexing and incubated on ice for 30 min. After this time, samples were sonicated twice (seven cycles of 30 s on and 30 s off) with a Bioruptor Plus sonicator (Diagenode). In between the two sonication cycles, samples were incubated at 12 °C with 0.8 µl of Benzonase (Sigma, E1014). Finally, lysates were centrifuged at 4 °C for 15 min at maximum speed, and the resulting supernatant was used for SDS–PAGE and western blotting directly or for co-IP followed by SDS–PAGE and western blotting. All antibodies used for western blotting are reported in Supplementary Table 6. Antibody dilutions for western blotting are as follows: 1:2,000 for Snf2h, Ctcf, Acf1, Bptf and LaminB antibodies; 1:1,000 for Rsf1, Tip5 and Rad21 antibodies; 1:500 for Cecr2 antibody and 1:5,000 for Wstf antibody.

### Co-IP followed by western blotting

For co-IP, protein G Dynabeads magnetic beads (Thermo Fisher Scientific, 10004D) were washed twice and resuspended in their original volume with RIPA buffer diluted (1:1) in dilution buffer (10 mM Tris–HCl (pH 7.5), 150 mM NaCl, 0.5 mM EDTA, 1 mM DTT and protease inhibitor). Nuclear extracts for the IP were obtained, as described in the previous section. For preclearing, 15 µl of beads were added to nuclear extracts and incubated for 30 min at 4 °C with overhead rotation. Five percent of the precleared lysate was used as input control. To the rest of the precleared lysates, 5 µg of anti-SNF2H antibody was added and samples were left overnight at 4 °C with rotation. The following day, 25 µl of prewashed magnetic beads were added and samples were left for 1 h at 4 °C with rotation. After this time, beads were washed three times with 500 µl IP-wash buffer (20 mM Tris–HCl (pH 8.0), 150 mM NaCl, 1 mM EDTA, 1.5 mM MgCl$_2$, 0.5% NP-40, 1 mM DTT and protease inhibitor). Beads were then transferred to a new tube and washed again twice before being resuspended in an appropriate volume of gel-loading buffer (1:1 mix of RIPA buffer and 5× Laemmli buffer with 5% β-mercaptoethanol) for SDS–PAGE and western blotting. The antibodies used for co-IP and western blotting are reported in Supplementary Table 6.

### Co-IP followed by mass spectrometry

The day before the IP, WT and Snf2hΔ mES cells were seeded into a 15-cm plate ($1 \times 10^7$ cells per plate). For each condition, three independent replicates were prepared. For the lysis, $1.5 \times 10^7$ cells per sample were washed in PBS and resuspended in 1 ml of hypotonic solution (20 mM Tris–HCl (pH 7.4), 10 mM NaCl and 3 mM MgCl$_2$) and incubated on ice for 5 min. After incubation, NP-40 was added to a final concentration of 0.1%; samples were mixed gently, left on ice for 5 min and centrifuged for 5 min at 500$g$. The resulting pellets were resuspended in 900 µl of B150AG buffer (10 mM Tris–HCl (pH 7.5), 2 mM MgCl$_2$, 150 mM NaCl, 0.5% Triton X-100, 50 mM L-arginine (Sigma A5006), 50 mM L-glutamine (Sigma, G1251) + protease inhibitor) and mixed by vortexing. After that, samples were centrifuged for 5 min at 4 °C at maximum speed. Supernatants were then transferred to new tubes for IP.

For each IP, 25 µl of protein G Dynabeads magnetic beads (Thermo Fisher Scientific) were washed twice with 1 mL of B150 buffer (10 mM Tris–HCl (pH 7.5), 2 mM MgCl$_2$, 150 mM NaCl and 0.5% Triton X-100). After washing, beads were resuspended in 100 µl of B150 buffer + protease inhibitor and incubated with lysates for 1 h at 4 °C. Precleared lysates were moved to fresh tubes and incubated at 4 °C overnight with 5 µg of a-Snf2h antibody. The day after, 25 µl of beads per IP were washed as described above and incubated with samples at 4 °C for 4 h. After incubation, beads were washed three times with B150 buffer, resuspended in 250 µl of B150nd buffer (10 mM Tris–HCl (pH 7.5), 2 mM MgCl$_2$ and 150 mM NaCl) and moved to clean tubes. Then beads were again washed on a magnet with 1 ml of B150nd buffer, the supernatant was removed and beads were centrifuged for 30 s at maximum speed.

Protein digestion was carried out as previously described[48]. In brief, beads were resuspended in 5 µl digestion buffer (3 M guanidinium hydrochloride, 20 mM EPPS (Hepps; pH 8.5), 10 mM chloroacetamide and 5 mM tris(2-carboxyethyl)phosphine) and digested with 1 µl of 0.2 µg µl$^{-1}$ Lys-C at room temperature for 4 h. In total, 17 µl (50 mM) HEPES (pH 8.5) were added to the beads, followed by the addition of 1 µl (0.2 µg µl$^{-1}$) trypsin. Beads were then incubated at 37 °C overnight. The day after, another 1 µl of 0.2 µg µl$^{-1}$ trypsin was added, and samples were digested for an additional 5 h. Samples were acidified by adding 1 µl of 20% trifluoroacetic acid and sonicated in an ultrasound bath. Peptides were analyzed by liquid chromatography–tandem mass spectrometry on an EASY-nLC 1000 (Thermo Fisher Scientific) using a two-column set-up. The peptides were applied onto a peptide µPAC trapping column in 0.1% formic acid and 2% acetonitrile in H$_2$O at a constant flow rate of 5 µl min$^{-1}$. Using a flow rate of 500 nl min$^{-1}$, peptides were separated at room temperature with a linear gradient of 3–6% buffer B in buffer A in 4 min followed by a linear increase from 6% to 22% in 55 min, 22% to 40% in 4 min, 40% to 80% in 1 min, and the column was finally washed for 13 min at 80% buffer B in buffer A (buffer A: 0.1%

formic acid; buffer B: 0.1% formic acid in acetonitrile) on a 50 cm µPAC column (PharmaFluidics) mounted on an EASY-Spray source (Thermo Fisher Scientific) connected to an Orbitrap Fusion LUMOS (Thermo Fisher Scientific). The data were acquired using 120,000 resolution for the peptide measurements in the Orbitrap and a top T (3 s) method with higher-energy collisional dissociation fragmentation for each precursor and fragment measurement in the ion trap according to the recommendation of the manufacturer (Thermo Fisher Scientific).

## RNA-seq

Total RNA for RNA-seq was purified using the RNeasy Mini Kit (Qiagen), and any residual genomic DNA was removed using a DNA-free DNA Removal Kit (Invitrogen). The quality of purified RNA was assessed with Agilent Bioanalyzer. Sequencing libraries were prepared from purified RNA for two or three independent replicates using the TruSeq RNA Library Prep Kit v2 (Illumina). Libraries were analyzed and quantified using Agilent Bioanalyzer and sequenced on the Illumina HiSeq 2500 or Illumina NovaSeq. Sequencing depth and number of mapped reads for each sample are reported in Supplementary Table 7.

## ATAC−seq

ATAC−seq was performed according to previously described protocols[23]. Briefly, 50,000 cells were washed with cold phosphate-buffered saline and resuspended in 50 µl of lysis buffer (10 mM Tris−HCl (pH 7.4), 10 mM NaCl, 3 mM $MgCl_2$, 0.1% NP-40, 0.1% Tween-20 and 0.01% digitonin) and incubated on ice for 3 min to extract the nuclei. After lysis, 1 ml of wash buffer (10 mM Tris−HCl (pH 7.4), 10 mM NaCl, 3 mM $MgCl_2$ and 0.1% NP-40) was added, and the tubes were inverted to mix. The nuclei were cold centrifuged at $500g$ for 10 min. The nuclei pellet was incubated in 50 µl of transposition reaction buffer (25 µl (2×) TD buffer, 2.5 µl transposase (100 nM final), 16.5 µl PBS, 0.5 µl (1%) digitonin, 0.5 µl (10%) Tween-20 and 5 µl water) for 30 min at 37 °C in an orbital shaker. The DNA was purified using the MinElute PCR Purification Kit (Qiagen). The eluted transposed DNA was submitted to PCR using Q5 High-Fidelity Polymerase (New England Biolabs). DNA was amplified with seven cycles of PCR. The libraries were analyzed and quantified using Agilent Bioanalyzer or Agilent Fragment Analyzer and sequenced on the Illumina NextSeq platform at 41 bp paired-end or NovaSeq at 50 bp paired-end. Sequencing depth and number of mapped reads for each sample are reported in Supplementary Table 7. All ATAC−seq experiments were performed in at least two independent replicates per condition.

## MNase-seq

MNase-seq was performed as previously described[23]. In brief, 1 million cells were resuspended in 1 ml of buffer 1 (0.3 M sucrose, 15 mM Tris (pH 7.5), 60 mM KCl, 15 mM NaCl, 5 mM $MgCl_2$, 2 mM EDTA, 0.5 mM DTT, 1× protease inhibitor cocktail (PIC), 0.2 mM spermine and 1 mM spermidine) with detergent (buffer 1 + 0.02% NP-40) and incubated on ice for 5 min. Nuclei were then pelleted at $300g$ for 5 min at 4 °C. Nuclei were gently resuspended in 1 ml of buffer 2 (0.3 M sucrose, 15 mM Tris (pH 7.5), 60 mM KCl, 15 mM NaCl, 5 mM $MgCl_2$, 0.5 mM DTT, 1× PIC, 0.2 mM spermine and 1 mM spermidine). Nuclei were then pelleted for 5 min at 300g at 4 °C. Pellets were resuspended in 400 µl MNase buffer (0.3 M sucrose, 50 mM Tris (pH 7.5), 4 mM $MgCl_2$, 1 mM $CaCl_2$ and 1× PIC). In total, 5 U of MNase S7 micrococcal nuclease (Roche, 10107921001) were added. Nuclei were then incubated for 30 min at 37 °C. Reaction was stopped by adding EDTA to a final concentration of 5 mM. SDS (to a final concentration of 1%) and proteinase K (200 µg ml⁻¹) were then added to the samples, followed by incubation at 55 °C for 1 h with shaking. MNase-digested DNA was purified using AMPure XP beads (Beckman Coulter, A63881). In total, 500 ng of purified DNA was then used for library preparation with the NEBNext Ultra Library Preparation Kit (New England Biolabs), using five PCR cycles. Libraries were analyzed and quantified using Agilent Bioanalyzer and sequenced on the Illumina

NextSeq 500 (41 bp paired-end) or Illumina NovaSeq. Sequencing depth and number of mapped reads for each sample are reported in Supplementary Table 7. All MNase-seq experiments were performed in at least two independent replicates per condition.

## SMF and amplicon bisulfite sequencing

SMF was carried out as previously described[49] for WT and BptfΔ cells in three independent replicates. In brief, $0.25 × 10^6$ cells per sample were lysed in 1 ml ice-cold lysis buffer (10 mM Tris (pH 7.4), 10 mM NaCl, 3 mM $MgCl_2$, 0.1 mM EDTA and 0.5% NP-40). After spinning, nuclei were first resuspended in 250 µl of ice-cold wash buffer (10 mM Tris (pH 7.4), 10 mM NaCl, 3 mM $MgCl_2$ and 0.1 mM EDTA) and then in 1× M.CviPI buffer (New England Biolabs) containing 1 mM SAM, 300 mM sucrose and 200 U of GpC methyltransferase (M.CviPI; New England Biolabs, M0227L) in a total volume of 300 µl and incubated at 37 °C for 15 min. The reaction was stopped by adding 300 µl of stop solution (20 mM Tris−HCl (pH 7.9), 600 mM NaCl, 1% SDS and 10 mM EDTA). After RNAse A and proteinase K treatments, DNA was first purified by phenol:chloroform and then further extracted with only chloroform. Finally, DNA was precipitated in isopropanol and resuspended in dd$H_2$O. For each sample, 2 µg of methylated DNA was used for bisulfite conversion using an EZ DNA methylation-gold kit (Zymo). Converted DNA was used as a substrate for PCR amplification of endogenous CTCF sites using bisulfite-compatible primers (Supplementary Table 8) and KAPA HiFi Uracil+ (Roche) ((95 °C, 4 min) ×1, (98 °C, 20 s; 60 °C, 15 s; 72 °C, 20 s) ×35, (75 °C, 5 min) ×1 and 4 °C, hold). Amplicons were purified using AMPure XP beads, pooled by sample and used for library preparation using the NEBNext Ultra Library Preparation Kit (New England Biolabs). Libraries were analyzed and quantified using Agilent Bioanalyzer and sequenced on an Illumina MiSeq (250 bp paired-end).

## CUT&RUN

Cleavage under targets and release using nuclease (CUT&RUN) was performed following the EpiCypher manufacturer's protocol (v.1.5.2) with some modifications. For all conditions, CUT&RUN was performed with both SNF2H and IgG antibodies (Supplementary Table 6) in two independent replicates. In brief, the day before the experiment, WT, Snf2hΔ and BptfΔ mES cells were seeded in six-well plates. The day after, 10 µl per sample of concanavalin A beads (concanavalin A magnetic beads; Bangs Laboratories, BP531) were washed twice with bead activation buffer (20 mM HEPES (pH 7.9), 10 mM KCl, 1 mM $CaCl_2$ and 1 mM $MnCl_2$) and resuspended in 10 µl of the same buffer. For each sample, $0.5 × 10^6$ cells were washed in PBS and centrifuged at room temperature at $600g$ for 3 min. Cells were then washed twice in 100 µl of wash buffer (20 mM HEPES (pH 7.5), 150 mM NaCl, 0.5 mM spermidine, protease inhibitor) and resuspended in 100 µl of the same buffer. Afterward, cells were aliquoted in eight-strip tubes containing 10 µl of activated beads, mixed with gentle vortexing and incubated at room temperature for 10 min. After this time, supernatant was removed and beads were gently resuspended in 50 µl of cold antibody buffer (wash buffer + 0.001% digitonin + 2 mM EDTA). In total, 0.5 µl of antibodies were added to each sample and left overnight at 4 °C. The following day, beads were washed twice using 250 µl of cold digitonin buffer (wash buffer + 0.001% digitonin) and gently resuspended in 50 µl of the same buffer. In total, 2.5 µl of CUTANA pAG-MNase (20× pAG-MNase; Epicypher, 15-1016) were added to each of the samples, which were then gently mixed and left for 10 min at room temperature. After this time, 250 µl of cold digitonin buffer was added directly to the samples. The previous step was repeated for two washes, and then samples were resuspended in 50 µl of digitonin buffer. To start the digestion, 2 µl of 50 mM $CaCl_2$ was added to the samples, which were then gently mixed and left at 4 °C for 2 h. After this time, 33 µl of stop buffer (340 mM NaCl, 20 mM EDTA, 4 mM EGTA, 50 µg ml⁻¹ RNase A and 50 µg ml⁻¹ glycogen) were added, and samples were then vortexed and incubated at 37 °C for 10 min. Samples were moved on the magnet, and

the supernatant was transferred to clean 1.5 ml tubes for nucleic acid extraction using the MinElute PCR Purification Kit (Qiagen). Purified DNA was used for library preparation using the NEBNext Ultra Library Prep Kit (Illumina) according to the manufacturer's instructions with the following Epicypher manufacturer's modifications. DNA clean-up before PCR amplification was done using 1.1× AMPure XP beads. PCR amplification parameters were adjusted to 1 cycle of 45 s at 98 °C, 14 cycles of 15 s at 98 °C followed by 10 s at 60 °C and 1 cycle of 1 min at 72 °C. DNA was again purified using 1.1× AMPure XP beads and eluted in 0.1× TE buffer. Libraries were analyzed and quantified using Agilent Bioanalyzer and sequenced on the Illumina NextSeq platform at 41 bp paired-end. Sequencing depth and number of mapped reads for each sample are reported in Supplementary Table 7.

## ChIP–seq

ChIP was carried out as previously described[23]. In brief, cells were grown to confluence and cross-linked in DMEM containing 1% formaldehyde for 10 min at room temperature. The reaction was quenched with 200 mM (final concentration) glycine, and cells were scraped off and rinsed with 10 ml of 1× PBS. Pellets were resuspended first in 10 ml of buffer 1 (10 mM Tris (pH 8.0), 10 mM EDTA, 0.5 mM EGTA and 0.25% Triton X-100) and then in 10 ml of buffer 2 (10 mM Tris (pH 8.0), 1 mM EDTA, 0.5 mM EGTA and 200 mM NaCl). Then cells were lysed in 1 ml lysis buffer (50 mM HEPES/KOH (pH 7.5), 500 mM NaCl, 1 mM EDTA, 1% Triton X-100, 0.1% DOC, 0.1% SDS, protease inhibitors) and sonicated for 20 cycles of 30 s using a Diagenode Bioruptor Pico, with 30 s breaks in between cycles. For the IP, lysate was first precleared with protein A/G magnetic Dynabeads Magnetic beads (Thermo Fisher Scientific) for 1 h at 4 °C and then incubated with 5 µg of antibody (Supplementary Table 6) overnight at 4 °C. The mixture was then incubated for 3 h at 4 °C with washed protein A/G magnetic Dynabeads Magnetic beads. Beads were washed three times with 1 ml lysis buffer, once with 1 ml DOC buffer (10 mM Tris (pH 8.0), 0.25 M LiCl, 0.5% NP-40, 0.5% deoxycholate and 1 mM EDTA), once with TE and bound chromatin was eluted in 1% SDS/0.1 M NaHCO$_3$. After RNase A treatment, proteinase K digestion was performed at 55 °C for 2 h, before reversing the cross-linking by overnight incubation at 65 °C. DNA was isolated by purification using AMPure XP beads. A sample of the input chromatin was treated in the same way to generate total input DNA. Immunoprecipitated DNA and 200 ng of input DNA were submitted to library preparation (NEBNext Ultra DNA Library Prep Kit; Illumina). In the library preparation protocol, input samples were amplified using five PCR cycles and IP samples using 12 cycles. Libraries were analyzed and quantified using Agilent Bioanalyzer and sequenced on the Illumina HiSeq 2500 or Illumina NovaSeq. Sequencing depth and number of mapped reads for each sample are reported in Supplementary Table 7. All ChIP–seq experiments were performed in at least two independent replicates per condition.

## Hi-C

Hi-C experiments were performed as previously described in two independent replicates[14]. In brief, 5 million cells per sample were fixed in DMEM containing 1% formaldehyde for 10 min at room temperature. After quenching the reaction with glycine (125 mM final concentration) and a 15 min incubation on ice, cells were pelleted and resuspended in 500 µl of freshly prepared ice-cold lysis buffer supplemented with protease inhibitors (10 mM Tris–HCl (pH 8.0), 10 mM NaCl, 0.2% NP-40 and 1× protease inhibitors). Nuclei were then resuspended in ice-cold 1× NEBuffer 2 containing 0.05% of SDS and incubated at 37 °C for 30 min, after which Triton X-100 was added to quench the SDS. Permeabilized nuclei were digested with 400 U of MboI restriction enzyme (NEB, R0147M) at 37 °C overnight. The 5′ overhangs were filled with the incorporation of biotin-14-dATPs (Thermo Fisher Scientific, 19524016) using DNA polymerase I large (Klenow) fragment (NEB, M0210L), and resulting blunt ends were ligated with T4 DNA Ligase (NEB, M0202T).

After RNAse A and proteinase K treatment and reverse cross-linking (65 °C overnight), DNA was purified using AMPure XP beads, and 2 µg of purified DNA was sheared using Covaris S220 Focused-ultrasonicator to an average size of 300–500 bp. The biotin-labeled ligation junctions were then captured using Dynabeads MyOne Streptavidin T1 beads (Thermo Fisher Scientific, 65601) in binding buffer (5 mM Tris–HCl (pH 7.5), 0.5 mM EDTA and 1 M NaCl). Finally, end-repairing, A-tailing, Illumina adapter ligation and indexing PCR amplification (six cycles) were performed on beads using NEBNext reagents and buffers before final purification with AMPure XP beads. Resulting libraries were analyzed and quantified using Agilent Bioanalyzer and sequenced on Illumina NextSeq at 41 bp paired-end.

## Co-IP MS protein enrichment analysis

Co-IP MS enrichment analysis was carried out as previously described[48]. In brief, protein identification and relative quantification were performed with MaxQuant (v.1.5.3.8) using Andromeda as the search engine[50] and label-free quantification[51,52]. The mouse subset of the UniProt (v.2019_04) combined with the contaminant database from MaxQuant was searched, and the protein and peptide false discovery rate were set to 1% and 0.1%, respectively. The following analysis was performed using R v.4.3.0. Protein intensities were first normalized to the smallest total sum of intensities across all samples, then log$_2$-transformed after dividing samples by $2^{20}$ and adding a pseudocount of five to stabilize the variance of the data. SNF2H-enriched samples were compared to datasets generated by co-IP MS in the Snf2hΔ line (that is, mock IP), and significance estimates were determined using limma[53] (v.3.56.2). Proteins with <0.01 adjusted $P$ value were considered significantly enriched.

## RNA-seq data analysis

RNA-seq reads were aligned to the mouse genome (BSgenome. Mmusculus.UCSC.mm10 v.1.4.0). Promoters were defined as ±1,000 nt around the TSS of each transcript in the University of California Santa Cruz (UCSC) Known Genes database, which was accessed via the Bioconductor package TxDb.Mmusculus.UCSC.mm10.knownGene v.3.10.0. Reads were aligned using the qAlign function from the QuasR package (v.1.40.1), with parameters 'splicedAlignment = TRUE' and 'aligner = 'Rhisat2'. Differential expression analysis was performed using gene-level quantifications and the quasi-likelihood method (glmQLFit and glmQLFTest functions) with default parameters using the edgeR package (v.3.40.2). First, weakly expressed or nondetected genes were filtered out using the filterByExpr function, and then a model was fitted of the form -batch + genotype (where batch is a factor with levels corresponding to the batch of RNA-seq experiment associated with the sample, and genotype is a factor with levels corresponding to the genotype of the cell line). DEGs for each knockout–WT pair were selected among those having at least twofold change (FC; absolute log$_2$(FC) > 1) in either direction and a false discovery rate smaller than 0.01, as calculated by edgeR.

The same DEGs were clustered using $k$-means ($k$-means function from statistics) with $k = 4$. The number of clusters, $k$, was determined by performing $k$-means clustering over a range of $k$ of 2–30 and selecting a value over which reduction in the total within-cluster sum of squares appeared less significant (elbow method). DEGs were displayed using ComplexHeatmap (v.2.12.0). For all plots, the mean signal from at least two independent replicates is reported unless otherwise specified in the figure legend.

For Gene Ontology analysis, enriched 'Biological Process' terms in both upregulated and downregulated genes for each genotype were searched using the enrichGO function in the clusterProfiler package (v.4.8.1), using all expressed genes as background and with a $P$ value cut-off of 0.01. Promoter regions of upregulated and downregulated genes from each deletion line were annotated by chromatin states defined by distinct combinations of histone modifications,

as available from ref. 54 (chromHMM[18],[19] maps for mESC ENCODE[55] datasets were downloaded from https://github.com/guifengwei/ChromHMM_mESC_mm10). To streamline, StrongEnhancer (state S8) was merged into Enhancer (S4), and Insulator, TranscriptionElongation, TranscriptionTransition, Heterochromatin and WeakEnhancer states were labeled as other. For promoters overlapping with multiple chromatin states, assignment was prioritized in the following order: ActivePromoter, BivalentChromatin, Enhancer, RepressedChromatin, other and intergenic.

### ATAC−seq data analysis

ATAC−seq reads were trimmed using cutadapt v2.5 with parameters, -a CTGTCTCTTATACACA -A CTGTCTCTTATACACA -m 5 -overlap = 1, and mapped to the mouse genome (BSgenome.Mmusculus.UCSC. mm10 v.1.4.0) using the qAlign function in QuasR (v1.40.1) with default parameters, which uses bowtie for short read alignments. ATAC−seq peaks were called using MACS2 (v.2.2.7.1) with parameters --nomodel --shift -100 --extsize 200 --keep-dup all -g mm --qvalue = 1e−2. For comparative analysis, a unique peak set was created with all genomic regions that were called as a peak in at least two replicates of at least one sample.

Differentially accessible regions were called using read counts on peaks and the quasi-likelihood method (glmQLFit and glmQLFTest functions) with default parameters using the edgeR package (v.3.40.2). A model was fitted of the form -batch + genotype (where batch is a factor with levels corresponding to the batch of ATAC−seq experiment associated with the sample, and genotype is a factor with levels corresponding to the genotype of the cell line). Differentially accessible peaks were clustered using $k$-means with $k$ = 4 (using the $k$-means function from statistics). $\log_2$(FCs) in ATAC signal at these differentially accessible peaks were displayed using ComplexHeatmap (v.2.12.0).

For motif analysis, enrichment for each of the vertebrate TF motifs contained in the JASPAR2022 database[56] was calculated using the calcBinnedMotifEnrR in the monaLisa package (v.1.6.0)[57]. For visualization, motifs that had a $\log_2$ fold enrichment of > 1.5 and $−\log_{10}$-adjusted $P$ value > 100 in at least one bin were selected.

ATAC−seq metaprofiles around bound CTCF sites were generated using the qProfile function from QuasR to get read counts in a 2-kb window anchored by the oriented CTCF-binding motif, normalized by sequencing depth. Profiles were then smoothed with a running mean of 21 bp and multiplied by 100. For all plots, the mean signal from at least two independent replicates is reported unless otherwise specified in the figure legend.

Strong DHSs were defined as previously described[14].

### MNase-seq data analysis

MNase-seq reads were trimmed using cutadapt v.2.5 with parameters -a AGATCGGAAGAGCACACGTCTGAACTCCAGTCA -A AGATCGGAAGAGCGTCGTGTAGGGAAAGAGTGT -m 5 --overlap=1, and mapped to the mouse genome (BSgenome.Mmusculus.UCSC.mm10 v.1.4.0) using the qAlign function in QuasR (v.1.40.1) with default parameters, which uses bowtie for short read alignment. MNase-seq metaprofiles around sites of interest (TSSs, DHSs or TF-binding sites) were generated using the qProfile function from QuasR with the parameter shift = 'halfInsert', to get MNase fragment midpoint counts in a 2-kb window anchored to the TF-binding motif. Raw counts were normalized by dividing through the median of each profile and multiplying by the median of all sample medians. Profiles were then smoothed with a running mean of 21 bp. Similarly, heatmaps of MNase-seq fragment midpoints were generated by normalizing the profiles to sequencing depth.

NRLs were calculated using a Phasogram-based approach described in refs. 14,58 implemented using the calcPhasogram and estimateNRL functions in the swissknife R package (https://fmicompbio.r-universe.dev/swissknife v.0.40) with default parameters.

For plots showing MNase signal as a function of fragment length, we counted reads in a 2 kb window centered on bound CTCF motifs ('ChIP−seq data analysis') and divided based on the length of the sequenced fragment. Data were standardized using the scale function in R.

### CUT&RUN data analysis

CUT&RUN reads were trimmed using cutadapt v.2.5 with parameters -a AGATCGGAAGAGCACACGTCTGAACTCCAGTCA -A AGATCGGAAGAGCGTCGTGTAGGGAAAGAGTGT -m10 -overlap = 1, and mapped to the mouse genome (BSgenome.Mmusculus.UCSC.mm10 v.1.4.0) using the qAlign function in QuasR (v.1.40.1) with default parameters. To account for differences in library size, the number of total reads mapped for each sample was scaled down to the sample with the lowest number of mapped reads. Average metaplots and single locus heatmaps were generated using the qProfile function in QuasR with default parameters; profiles were calculated over 2 kb windows centered on either CTCF-bound motifs or DHS center (see below) and smoothed over 51 bp. For the average metaplots, the signal was divided by the total number of genomic regions considered. For the single locus heatmaps, CTCF regions were sorted by SNF2H signal in WT. For the boxplots (Extended Data Fig. 5e,g), reads were counted over 250 bp windows centered on the region of interest using the QuasR function qCount, whereby reads were shifted by half the fragment length and each fragment was counted once. $\log_2$ read counts were calculated as $\log_2(n + 8)$, in which $n$ is the library-size normalized count and 8 is the pseudocount, used to decrease noise levels at low read counts in any comparison. Enrichment over controls (IgG) was calculated by subtracting the $\log_2$ read counts of the control from the $\log_2$ read counts of the corresponding sample. All plots were generated using the mean values from two independent replicates.

DHSs and CTCF-bound sites were defined as described in the above sections.

### ChIP−seq data analysis

ChIP−seq reads were trimmed using cutadapt v.2.5 with parameters -a AGATCGGAAGAGCACACGTCTGAACTCCAGTCA -m 5 −overlap=1, and mapped to the mouse genome (BSgenome.Mmusculus.UCSC. mm10 v.1.4.0) using the qAlign function in QuasR (v.1.40.1) with default parameters, which uses bowtie for short read alignment. ChIP enrichment between immunoprecipitated and input samples was calculated using the following equation:

$$e_i = \log_2 \frac{n_i/N_i \times \text{median}(N) + 8}{n_j/N_j \times \text{median}(N) + 8},$$

where $e_i$ is the ChIP enrichment of a region in sample $i$; $n_i$ and $n_j$ are the number of alignments in the immunoprecipitated sample $i$ and the corresponding input sample $j$; $N_i$ and $N_j$ are the library sizes (total number of alignments) in samples $i$ and $j$; median ($N$) is the median library size over all samples. Changes in ChIP enrichment between two immunoprecipitated samples were calculated using the same formula.

For genome-wide site predictions of CTCF, the motif MA0139.1 from the JASPAR2022 Bioconductor package v.0.99.7 was used. Bound CTCF sites were defined as motifs that have a $\log_2$(enrichment) (IP over input in a 251-bp window centered on the motif) of at least 1.0 (twofold). CTCF motifs were clustered using $k$-means with $k$ = 5 (using the $k$-means function from statistics), using changes in ChIP−seq and ATAC−seq enrichment signal in BptfΔ cells compared to WT controls. ChIP−seq metaprofiles were generated using the qProfile function from QuasR (v.1.40.1) to get read counts in a 2-kb window anchored by the TF-binding motif without applying shifting. Counts were normalized by sequencing depth, divided by the total number of sites and multiplied by 1,000. Profiles were then smoothed with a running mean of 21 bp.

For the heatmap with average ChIP–seq profiles around per-cluster ATAC–seq peaks, ChIP–seq read counts in 2-kb windows centered on ATAC–seq peak midpoints were obtained using the qProfile function from QuasR pooling all samples measuring the same ChIP–seq target and normalizing them to RPKM. Normalized values were averaged across ATAC–seq peaks in each cluster and smoothed using a running mean of 45 bp. For better comparability between ChIP–seq targets, average cluster profiles from each target were further normalized by dividing through their maximum value or through 1.5 if the maximum was <1.5. CTCF sites were similarly annotated using the mESC chrom-HMM map, as explained above. To streamline the annotations, states were combined as Strong/Weak/Enhancers (S4/S8/S11), Heterochromatin/Repressed (S3/S5) and TranscriptionTransition/Elongation (S9/S10). For CTCF sites overlapping with multiple chromatin states, assignment was prioritized in the following order: ActivePromoter, BivalentChromatin, Enhancer, RepressedChromatin, Insulator and Intergenic. For all samples generated in this study, the mean signal from at least two independent replicates is reported.

### SMF analysis

Reads were trimmed using Trimmomatic[59] (v.0.32) in paired-end mode using the ILLUMINACLIP option. Trimmed reads were mapped to the mouse genome (BSgenome.Mmusculus.UCSC.mm10 v.1.4.0) using the qAlign function from the QuasR package (v.1.40.1) with parameters for bisulfite data. DNA methylation was quantified for all Cs using the qMeth function and then separated into Cs in the CpG or GpC context, removing GCG and CCG sequence contexts as these cannot be distinguished between endogenous methylation and SMF methylation. Plots of SMF data report the mean signal for three independent replicates of $(1 - GpC$ methylation) to visualize the footprint.

### Deep learning model

A CNN was trained on one-hot-encoded 150-bp-long DNA sequence(s) centered at CTCF-bound sites ($n = 47,630$) as input to predict the change in CTCF binding in BptfΔ cells compared to WT as measured by ChIP–seq. The architecture of CNN was adapted from Basset[60] and further modified based on the DeepSTARR design in ref. 27. The CNN in our study starts with four sequential convolutional layers (1D, filters = 128, 128, 128, 64; size = 5, 3, 5, 3) each followed by ReLU activation and max-pooling (size = 2). The output of the convolutional layers was fed into two fully connected layers with ReLU activation having 128 and 64 neurons, respectively. Dropout of 0.4 was applied after each fully connected layer. The final layer was used to predict the CTCF ChIP–seq changes in BptfΔ cells compared to WT, using a linear activation function. The model was implemented in the Keras framework[61] using the Keras R package (v.2.2.5.0), with TensorFlow[62] (v.2.0.0) backend. The training was performed using a mean-squared-error loss function and the Adam optimizer[63] with a batch size of 64 and monitored for early stopping based on validation loss (20% of the training set) with patience of 15 epochs. CTCF sites from chromosomes 16, 17, 18 and 19 were excluded from the training ($n = 32,988$) and validation ($n = 8,248$) sets and kept as the test set ($n = 6,394$) for model evaluation. For model interpretability, the DeepExplainer implementation[64] from the SHAP library[65,66] was used to calculate contribution scores for every nucleotide in the provided sequences around bound CTCF sites. As reference sequence for DeepExplainer, 100 dinucleotide-shuffled versions were generated for each CTCF site. To summarize the contribution of each nucleotide at each position across all input sequences, average contribution scores per position were computed for each of the four bases by taking the average of the contribution scores of the nucleotides present in the input sequence. The resulting contribution weight matrix (as introduced in ref. 67) was visualized using ggseqlogo[68] (v.1.0). TFBSTools[69] was used to identify the position of the M2 motif as defined in ref. 29 (downloaded from CTCFBSDB 2.0 (ref. 70)).

### Hi-C data analysis

Sequencing depth and number of mapped reads for each sample are reported in Supplementary Table 7. Hi-C analysis was performed using Python (v.3.9.7). Data were processed using the distiller-nf pipeline (https://github.com/open2c/distiller-nf). Briefly, reads were mapped using BWA (v.0.7.17) with default parameters 'bwa mem -SP5M'. Hi-C pairs contacts were extracted from mapped files and processed using pairtools[71] (v.1.0.2), specifically with the parse (with --add-columns mapq --walks-policy all additional arguments), sort and dedup (with max_mismatch_bp: 1 setting). Deduplicated pairs were then filtered to retain pairs with both sides mapping with high confidence (mapq ≥30) and converted to cooler format using cooler[72] (v.0.9.0) using the 'cload pairs' function with default parameters. At this stage, two replicates were merged. Matrices for individual samples and merged datasets were then zoomified to produce multiresolution cooler (.mcool) files, and these were balanced with the −cis-only flag. For all downstream analyses, when not specified otherwise, merged datasets were used. Further analysis was performed using the Quaich pipeline (https://github.com/open2c/quaich). We included the Hi-C dataset generated here, the SNF2H knockout dataset from our earlier work[14], the CTCF-AID depletion dataset from ref. 30 for analysis, together with the mESCs dataset from ref. 46 as a deep reference dataset for feature annotation. We used cooltools[73] (v.0.6.1) to calculate genome-wide insulation profiles at 10 kb resolution with 100 kb window size and used the (default) Li thresholded boundaries in the data from ref. 46 as TAD boundaries. To create a TAD annotation, we combined neighboring boundaries (using bioframe) and removed putative TADs longer than 1.5 Mb. Loops were called at 5 kb, 10 kb and 24 kb resolutions using mustache with arguments -d 10000000 -pt 0.05 -st 0.8, and results from different resolutions were merged (any dots within a 20 kb radius of each other were considered the same, and the one called at the highest resolution was retained). All pileups were created from 10 kb resolution data using coolpup.py[74] (v.1.1.0) with expected normalization and 300 kb pad for local pileups (around TAD boundaries, CTCF sites) or 100 kb for distal pileups (around loops). When insulation strength is reported in local pileups, it is calculated by dividing the mean of all values in the upper left and lower right quadrants over the mean of all values in the upper right and lower left quadrants (a ratio of contacts not crossing the central bin over the contacts encompassing the central bin), ignoring the first two diagonals. For compartment analysis, we used cooltools eigs-cis for eigenvector decomposition of intrachromosomal matrices at 100 kb resolution and used GC content calculated using cooltools genome gc as a phasing track.

### Statistics and reproducibility

No statistical method was used to predetermine the sample size. No data were excluded from the analyses. The experiments were not randomized, and the investigators were not blinded to allocation during experiments and outcome assessment.

### Reporting summary

Further information on research design is available in the Nature Portfolio Reporting Summary linked to this article.

## Data availability

Next-generation sequencing data generated in this study are available at Gene Expression Omnibus (GEO; https://www.ncbi.nlm.nih.gov/geo/) with accessions GSE234295 and GSE250229. The following public datasets were obtained from GEO: histone marks ChIP–seq: H3K27me3: (GSE30203, samples GSM747539 to GSM747541 (ref. 75)), H3K4me1: (GSE30203, sample GSM747542 (ref. 75)), H3K27ac: (GSE67867, samples GSM1891651 and GSM1891652 (ref. 76)), H3K36me3: (GSE33252, samples GSM801982 and GSM801983 (ref. 77)); Hi-C CTCF-AID-UNT and CTCF-AUX-48h (GSE98671, samples GSM2644945 to GSM2644948 (ref. 30)); ChIP–seq CTCF-AID-UNT and CTCF-AUX-48h (GSE98671,

samples GSM2609185 and GSM2609186 (ref. 30)); SNF2H knockout (GSE112136 Hi-C samples GSM3331341 to GSM3331344, MNase samples GSM3058339 to GSM3058342, RNA-seq samples GSM3058347 to GSM3058359, ChIP–seq samples GSM3058327 and GSM3058328 (ref. 14)). The UCSC annotation of known genes for mm10 was obtained through the Bioconductor annotation package TxDb.Mmusculus.UCSC.mm10.knownGene (https://doi.org/10.18129/B9.bioc.TxDb.Mmusculus.UCSC.mm10.knownGene). The Jaspar2022 (ref. 56) motif database used in this study can be accessed online (https://jaspar2022.genereg.net/). The MS proteomics data generated in this study have been deposited to the ProteomeXchange Consortium via the PRIDE[78] partner repository with the dataset identifier PXD042945. Source data for blots in Fig. 1b and Extended Data Figs. 1b, 2a and 10f are provided with this paper.

## Code availability

This study does not use custom codes. The analyses performed using publicly available packages are detailed in the Methods section.

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

## Acknowledgements

We thank members of the D.S. Laboratory for their critical input on study and manuscript. We thank L. Isbel for help with CUT&RUN experiments, R. Grand and M. Pregnolato for help with SMF experiments and M. Stadler for advice on data analysis. We thank S. Smallwood and the functional genomics platform of the FMI for next-generation sequencing support and D. Hess and the proteomics and protein analysis platform of the FMI for MS support. D.S. and L.G. acknowledge support from the Novartis Research Foundation, the Swiss National Science Foundation (310030B_176394 to D.S. and 310030_192642 to L.G.) and the European Research Council under the European Union's (EU) Horizon 2020 research and innovation program grant agreements (ReadMe-667951 and DNAaccess-884664 to D.S., 759366 'BioMeTre' to L.G.). M. Iurlaro was supported by a European Molecular Biology Organization advanced fellowship (ALTF 611-2019). The funders had no role in study design, data collection and analysis, decision to publish or preparation of the manuscript.

## Author contributions

M. Iurlaro, F.M. and D.S. conceived the study and planned the experiments. M. Iurlaro derived accessory subunit cell lines and performed the experiments and comprehensive data analysis. F.M. performed remodeler localization experiments and IP-MS experiments and analyzed resulting data; derived SNF2L knockout cell lines; and performed the related experiments and data analysis. I.M.F. performed analysis of Hi-C data. C.W. performed co-IP and western blots experiments. M. Iskar performed deep learning modeling and chromHMM analysis. L.B. supervised and instructed on data analysis and analyzed external RNA datasets for revisions. L.G. supervised the Hi-C data analysis and interpretation. M. Iurlaro, F.M. and D.S. interpreted the results and wrote the paper with input from all authors.

## Competing interests

The authors declare no competing interests.

## Additional information

**Extended data** is available for this paper at https://doi.org/10.1038/s41588-024-01767-x.

**Correspondence and requests for materials** should be addressed to Dirk Schübeler.

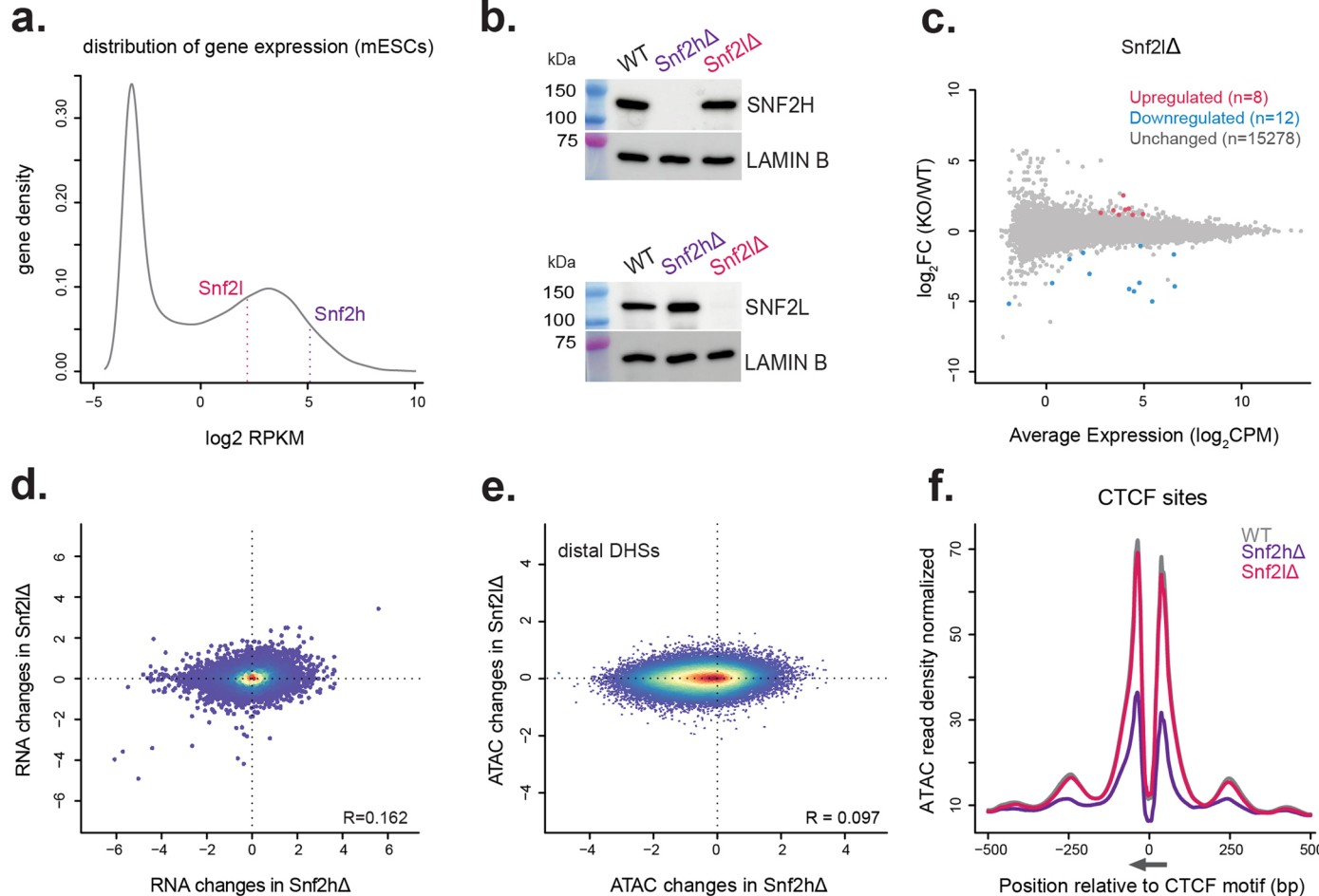

**Extended Data Fig. 1 | In contrast to SNF2H, deletion of SNF2L causes minor effects at the level of gene expression and chromatin accessibility in mES cells. a**. Gene expression (log₂RPKM) distribution in WT mES cells. Expression level of *Snf2l* and *Snf2h* are highlighted. **b**. Western blot detection of SNF2H and SNF2L (upper and lower blot, respectively) protein levels in WT and deletion lines. Blots are representative of two independent experiments. **c**. RNA changes in SNF2L deletion line are shown as MA plot. Differentially expressed genes (Methods) are reported. **d**. RNA changes in SNF2L deleted line (y axis) vs SNF2H deleted line (x axis) are shown as density scatter plot. **e**. ATAC changes in SNF2L deleted line (y axis) vs SNF2H deletion line (x axis) are shown as density scatter plot. **f**. Average ATAC signal at CTCF-bound sites in WT, Snf2lΔ and Snf2hΔ cells. Canonical motif orientation (5′ to 3′) indicated by the arrow.

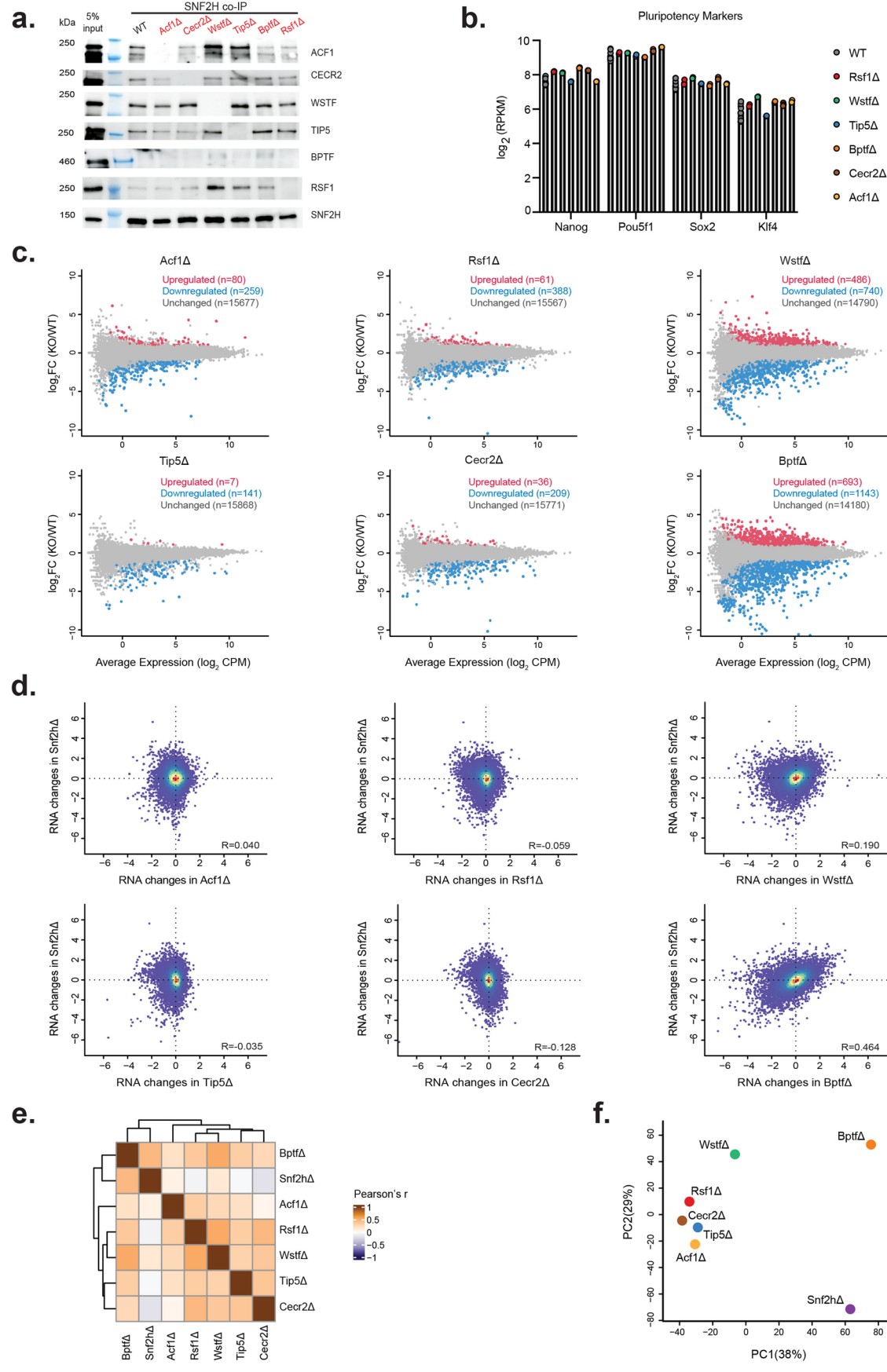

**Extended Data Fig. 2 | See next page for caption.**

**Article** https://doi.org/10.1038/s41588-024-01767-x

**Extended Data Fig. 2 | ISWI accessory subunit deletions are subcomplex-specific, and their transcriptional response shows a similarity between loss of BPTF and loss of SNF2H. a.** SNF2H co-immunoprecipitations followed by western blot against ISWI subunits in WT and ISWI deletion lines as indicated. Blots for detection of BPTF and SNF2H are representative of at least two experiments. Blots for the detection of all other proteins have not been repeated. **b.** Expression of several pluripotency markers (log$_2$ RPKM) in WT and ISWI deletion lines. **c.** RNA changes in the generated deletion lines are shown as MA plots. Differentially expressed genes are shown as blue (downregulated) and red (upregulated) dots. **d.** Quantitative comparison of RNA changes (log$_2$FC) upon deletion of accessory subunits (x-axis) versus SNF2H deletion (y-axis). R: Pearson's correlation coefficient. **e.** Heatmap of Pearson's correlation of transcriptional changes induced by each deletion. Correlation was calculated on log$_2$ fold change data of genes called as differentially expressed in at least one contrast. **f.** Principal component analysis of transcriptional changes induced by each ISWI deletion. PCA was performed on log$_2$ fold change data of genes called as differentially expressed in at least one contrast.

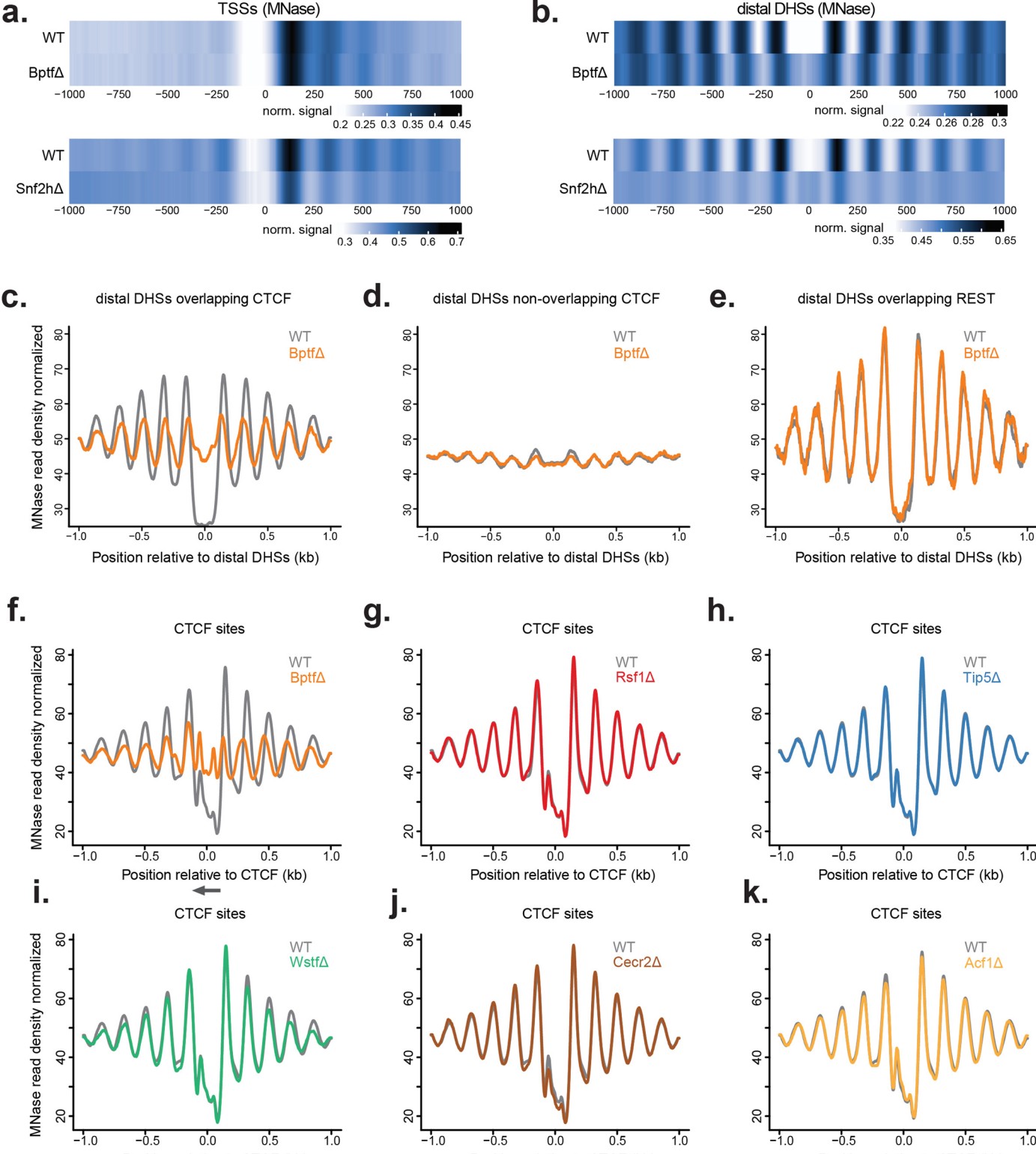

**Extended Data Fig. 3 | Nucleosome profiling reveals a BPTF-specific response at CTCF sites. a,b.** Average nucleosomal profiles at transcription start sites (TSSs, **a**) or distal DNaseI hypersensitive sites (distal DHSs, **b**) shown as heatmap for BptfΔ and Snf2hΔ cells with respective controls (Snf2hΔ and associated WT data taken from ref. 14). **c.** MNase average signal at distal DNaseI hypersensitive sites bound by CTCF in WT and BptfΔ cells. **d.** Same analysis (as in **c**) for distal DNaseI hypersensitive sites not bound by CTCF **e.** Same analysis (as in **c** and **d**) at distal DNaseI hypersensitive sites bound by REST. **f–k.** MNase average signal at CTCF sites in WT and all deletion lines. Canonical motif orientation (5′ to 3′) indicated by the arrow.

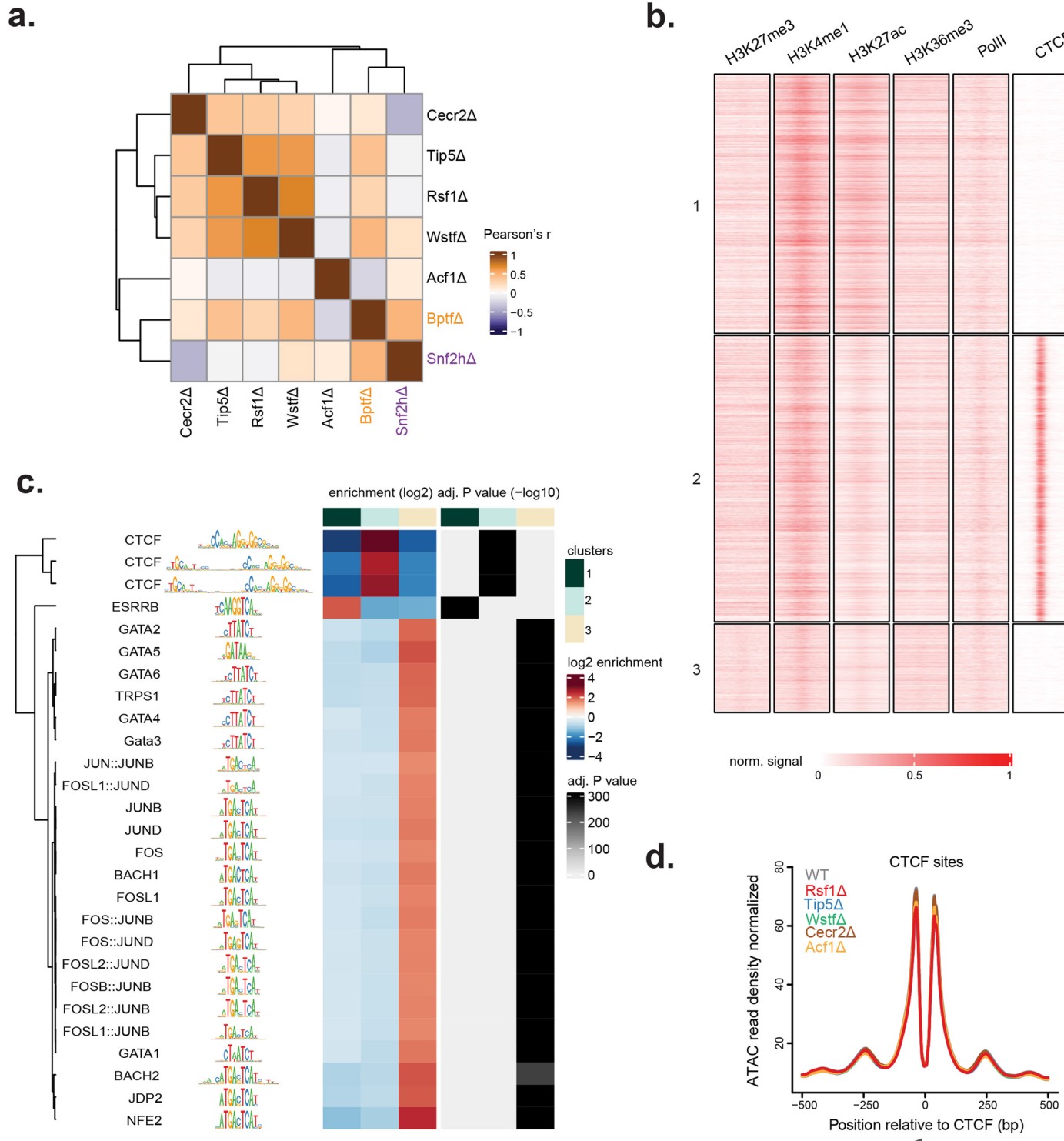

**Extended Data Fig. 4 | BPTF deletion displays loss of chromatin accessibility specifically at CTCF sites. a**. Heatmap showing Pearson's correlations of chromatin accessibility changes induced by each deletion. Correlation was calculated on log$_2$ fold change ATAC-seq signal on peaks called as differentially accessible in at least one contrast. **b**. Enrichment of chromatin marks and chromatin-associated factors in clusters with differential accessibility response (regions and clusters as in Fig. 2d, cluster number reported on the left). **c**. Motif enrichment analysis over the same clusters (as in Fig. 2d). Adjusted p-value calculated through a one-sided Fisher's exact test with Benjamini–Hochberg multiple testing correction. **d**. Average ATAC-seq signal at bound CTCF sites in WT and individual deletion lines. Canonical motif orientation (5′ to 3′) indicated by the arrow.

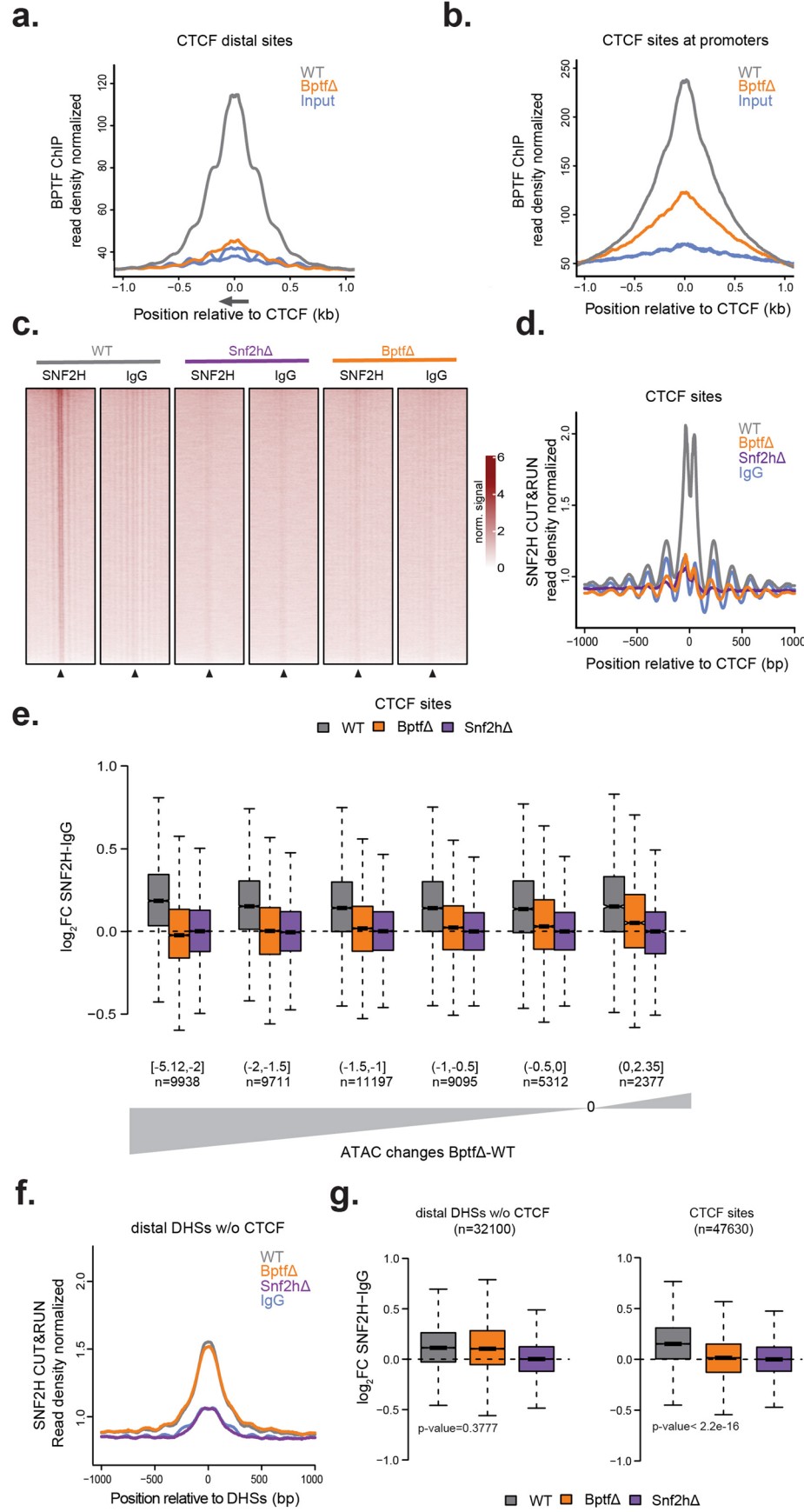

**Extended Data Fig. 5 | See next page for caption.**

**Extended Data Fig. 5 | Absence of BPTF affects SNF2H localization at CTCF sites. a**. Detection of BPTF by ChIP–seq at distal bound CTCF sites shown as average signal for WT (gray) and BptfΔ (orange). Input signal is shown in blue. Canonical motif orientation 5′–3′ indicated by the arrow. **b**. Same analysis as in **a** for bound CTCF sites within ±500 bp from annotated promoters. **c**. CUT&RUN signal for SNF2H in WT, Snf2hΔ and BptfΔ cells, shown as alignment densities centered on bound CTCF motifs (black arrowhead). IgG signal is shown for each background as negative control. **d**. SNF2H CUT&RUN average signal at bound CTCF sites for WT, BptfΔ, Snf2hΔ and IgG. **e**. Boxplot (as in Fig. 4a) showing SNF2H CUT&RUN signal (log₂ fold change over IgG control) for WT (gray), BptfΔ (orange), Snf2hΔ (purple) at bound CTCF sites grouped by ATAC changes in BptfΔ. Groups are displayed from left to right starting with regions with stronger ATAC loss on the left (n= number of sites in each group, range of ATAC changes in each group is reported in the x-axis). **f**. Detection of SNF2H by CUT&RUN at distal DNaseI hypersensitive sites that do not overlap with CTCF motifs shown as average signal for WT (gray), BptfΔ (orange), Snf2hΔ (purple). IgG signal is shown in blue as negative control. CTCF sites are excluded to illustrate consistent binding of SNF2H in WT and BptfΔ lines outside of CTCF-bound sites. **g**. Boxplots (as in Fig. 4a) showing SNF2H CUT&RUN signal (log₂ fold change over IgG control) for WT (gray), BptfΔ (orange), Snf2hΔ (purple) at distal DNaseI hypersensitive regions not overlapping with CTCF sites (left) and at bound CTCF sites (right), illustrating specific reduction in SNF2H signal over CTCF sites in the BptfΔ line. Significance between WT and BptfΔ conditions was calculated by a one-sided Wilcoxon signed-rank test.

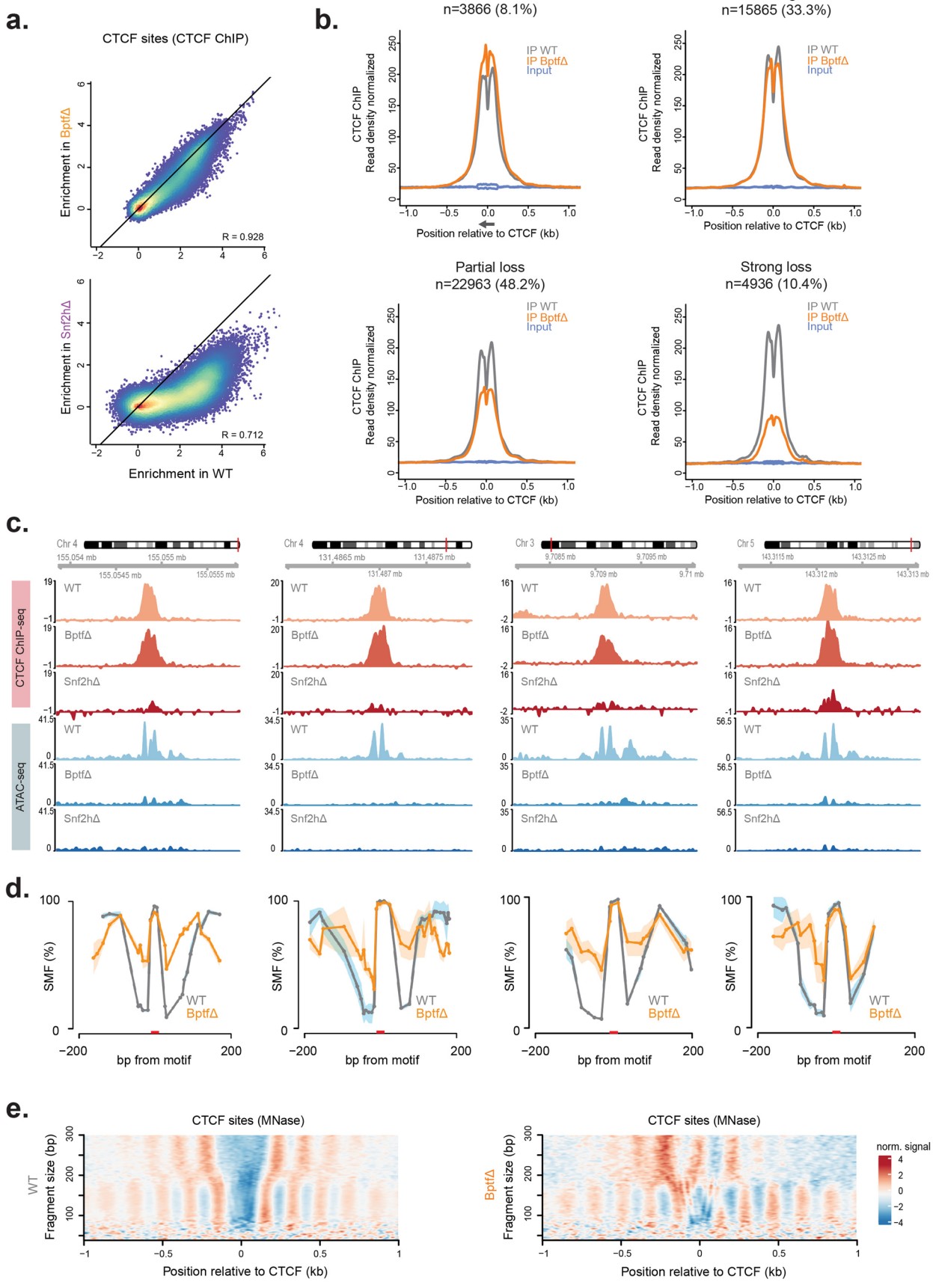

**Extended Data Fig. 6 | See next page for caption.**

**Extended Data Fig. 6 | CTCF binding largely persists in absence of BPTF but coincides with changes in nucleosome organization. a**. Quantitative comparison of CTCF binding (log$_2$ enrichment measured in IP/input) in WT cells (x-axis) vs BptfΔ (up) or Snf2hΔ (down) (y-axis), illustrating persistent CTCF binding in BptfΔ versus the strong reduction in Snf2hΔ. R: Pearson's correlation coefficient. **b**. Average CTCF ChIP−seq signal in WT and BPTF-depleted cells (inputs reported as control). CTCF sites have been divided based on their changes in binding upon BPTF depletion—in sites that gain binding (defined as regions with a log$_2$FC over WT equal to or higher than 0.25), unchanged sites (log$_2$FC over WT lower than 0.25 and higher than −0.25), sites with partial loss of binding (log$_2$FC over WT equal to or lower than −0.25 and higher than −1) and sites with strong loss of binding (log$_2$FC over WT lower than −1). Canonical motif orientation 5′−3′ indicated by the arrow. **c**. Single loci representative of regions with strong accessibility loss but unchanged CTCF binding upon BPTF deletion. ChIP signal is shown in shades of red for WT, BptfΔ and Snf2hΔ. ATAC signal is reported in shades of blue for the same lines. **d**. Single-molecule footprinting signal in WT and BPTF-depleted cells for the same sites shown in **c**. Shaded line represents standard deviation, red square indicates CTCF motif. **e**. V-plots representing standardized MNase data are shown as a function of fragment size on the y-axis and fragment midpoint position on the x-axis at bound CTCF sites in WT (left) and BptfΔ (right), highlighting relative accumulation of longer MNase fragments (>200 bp) spanning CTCF-bound sites upon deletion of BPTF.

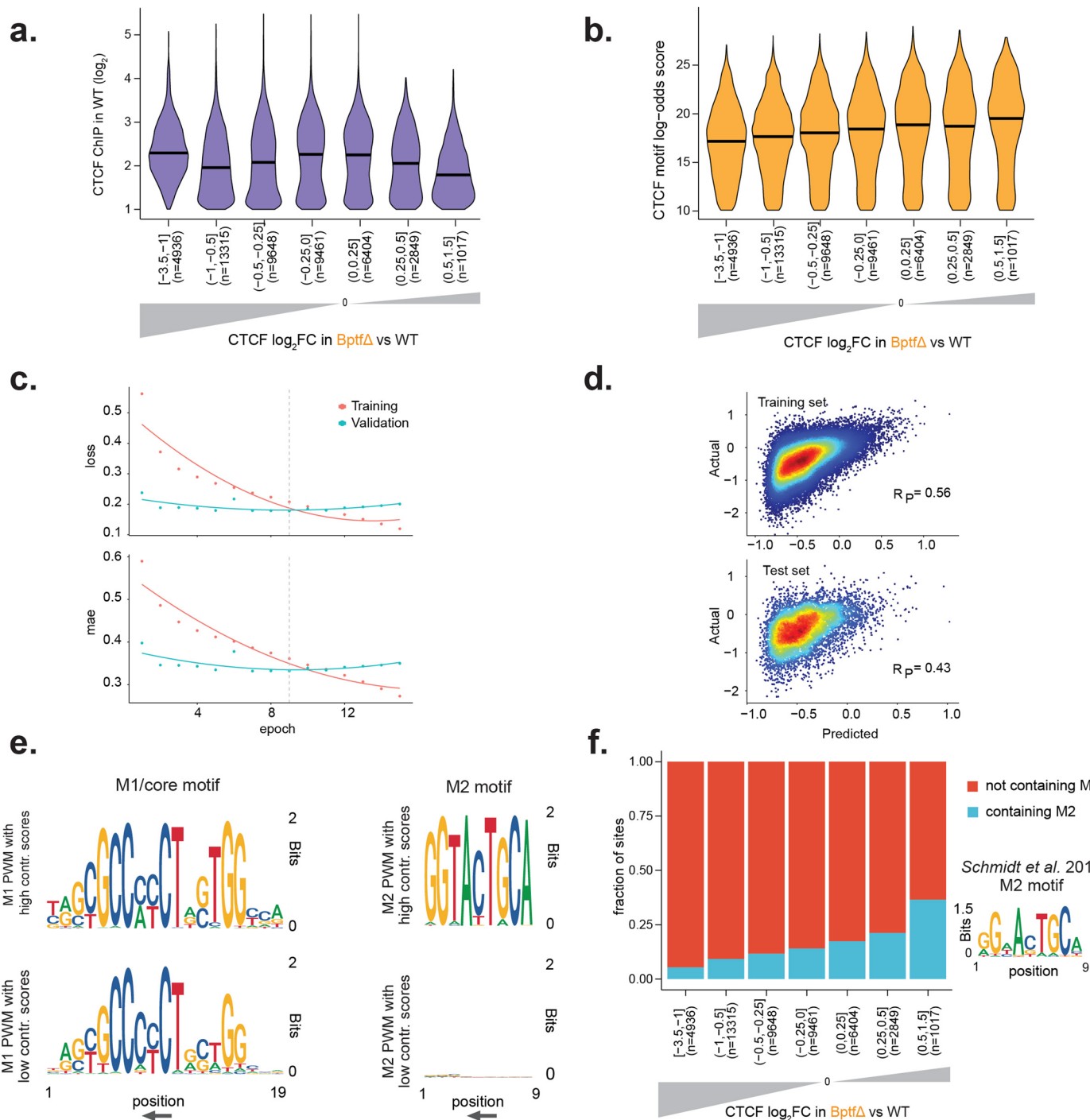

**Extended Data Fig. 7 | Deep learning identifies CTCF motif features enriched at sites of persistent binding in absence of NURF. a.** CTCF sites were grouped based on changes in CTCF binding in BptfΔ versus WT. For each group, the binding strength in WT cells is shown as violin plots with median (black line). n = number of CTCF sites within each group. Range of CTCF binding changes in each group is reported in the x-axis. **b.** Same groups (as in **a**) but now showing CTCF motif score (canonical motif M1 log-odds score) as violin plots with median (black line). Illustrating a trend for higher motif scores in sites with persistent CTCF binding in absence of BPTF. **c.** Plots of loss (mean squared error, top) and the mean absolute error (bottom) metrics at each training epoch step for the training (red line) and validation sets (blue line). The dotted line indicates the selected epoch with the minimum validation loss. **d.** Scatter plot showing the observed vs predicted CTCF ChIP–seq log₂ fold change in BptfΔ compared to WT for the training (top) and the test set from held-out chromosomes (bottom). $R_p$ indicates the Pearson correlation coefficient. **e.** Position weight matrix logos were generated in bits for the CTCF sites with the highest (n = 1000, top) and lowest (n = 1000, bottom) contribution scores calculated from the deep learning model. Sequence logos were created independently for M1 (left) and M2 (right) motifs. Canonical motif orientation 5′–3′ indicated by the arrow. **f.** Fraction of CTCF sites containing an M2 motif (as defined in ref. [29]) and grouped (as in **a** and **b** by CTCF changes in BptfΔ), illustrating the increased presence of M2 at sites with persistent CTCF binding in absence of BPTF.

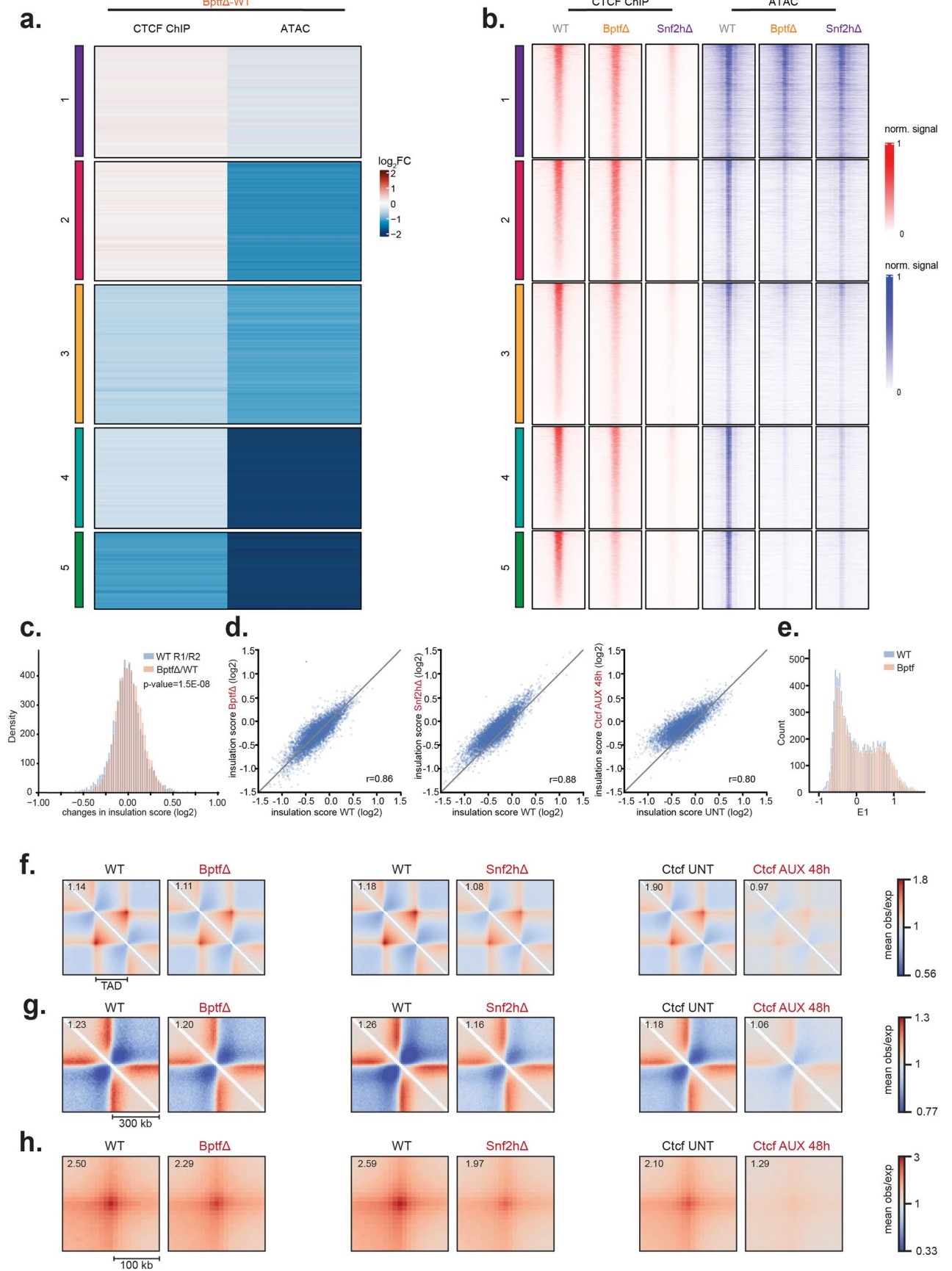

**Extended Data Fig. 8 | See next page for caption.**

**Extended Data Fig. 8 | Comparison of 3D genome changes upon BPTF, SNF2H or CTCF depletion. a**. Heatmap showing log$_2$ fold-changes in CTCF binding (ChIP–seq) and accessibility (ATAC-seq) upon BPTF deletion at clustered bound CTCF sites (as in Fig. 4a). Cluster numbers reported on the left. **b**. CTCF binding (ChIP–seq) and chromatin accessibility (ATAC-seq) signal at clustered bound CTCF sites (as in **a**), in WT, BptfΔ and Snf2hΔ cells. Cluster numbers are reported on the left. **c**. Changes in insulation score at TAD boundaries (boundaries identified in ref. 46 mESCs dataset) in BptfΔ vs WT (orange). Changes between replicates in WT condition are reported as control (blue). Two-sided Wilcoxon test p-value is reported. **d**. Scatter plots reporting insulation score at TAD boundaries in controls (x axis) and BptfΔ, Snf2hΔ and CTCF depleted cells (y axis). **e**. Compartment signal (first eigenvector values) for WT and BptfΔ cells. **f**. Mean observed/expected contact frequency measured using Hi-C at TADs[34] in BptfΔ, Snf2hΔ and CTCF depleted cells (48 h) and their respective controls. **g**. Same as **f** at TAD boundaries. **h**. Same as **f** and **g** at Hi-C loops.

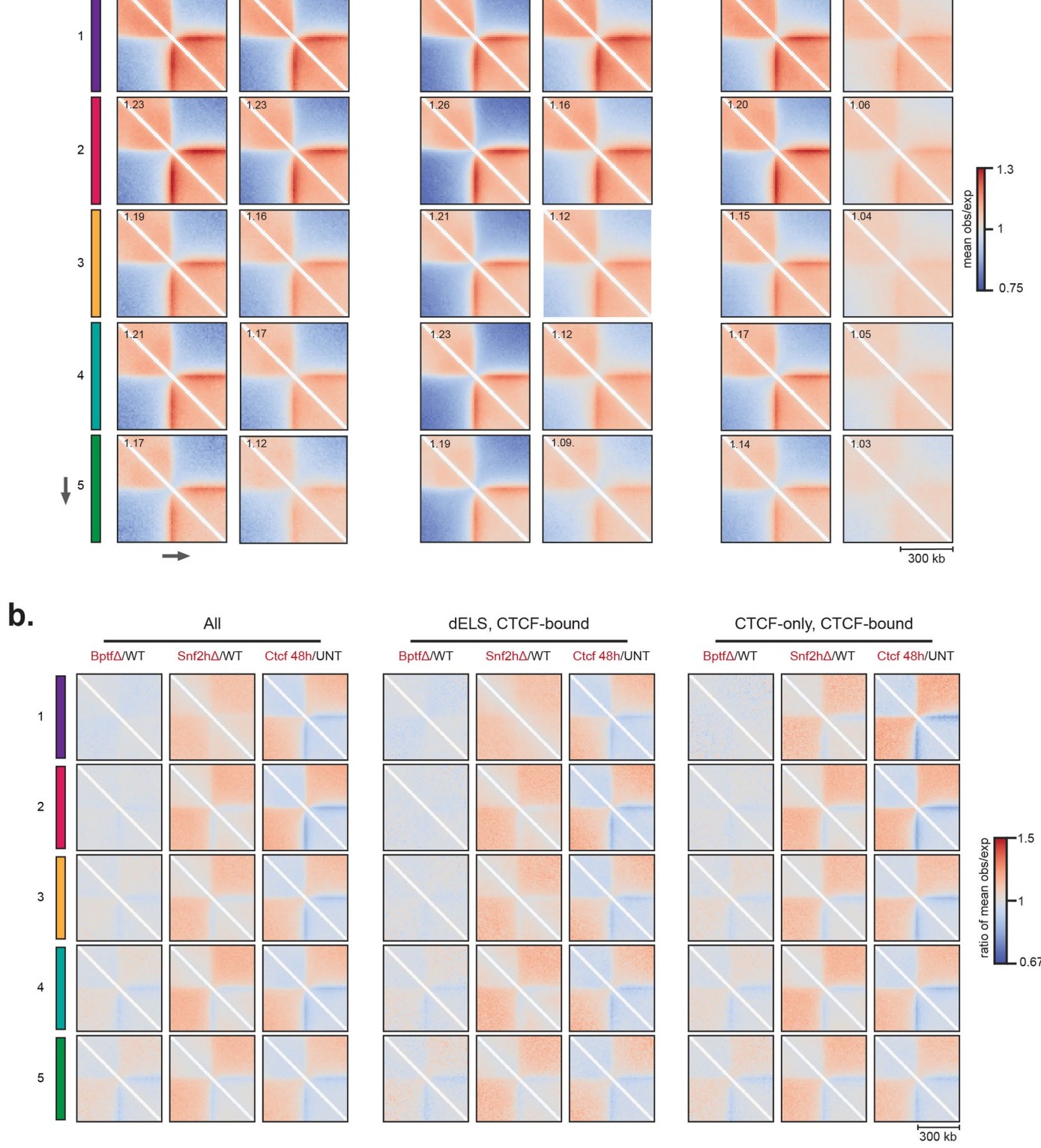

**Extended Data Fig. 9 | NURF-dependent accessibility and binding loss relate to loss of long-range chromatin interactions. a.** Mean observed/expected contact frequency measured using Hi-C at CTCF sites split by the same clusters as in Fig. 4a (cluster number on the left) in BptfΔ, Snf2hΔ and CTCF depleted cells (48 h) and their respective controls, at 10 kb resolution. **b.** Changes in observed/expected contact frequency measured by Hi-C at CTCF sites split by clusters as in **a** (cluster number on the left). CTCF sites are split into sites overlapping all candidate cis-regulatory elements (cCREs defined as in ref. 32) ('All', first panel from the left), CTCF sites overlapping cCREs with distal enhancer-like features ('dELS,CTCF-bound', second panel from the left), CTCF sites overlapping cCREs with only CTCF bound ('CTCF-only, CTCF-bound', third panel from the left).

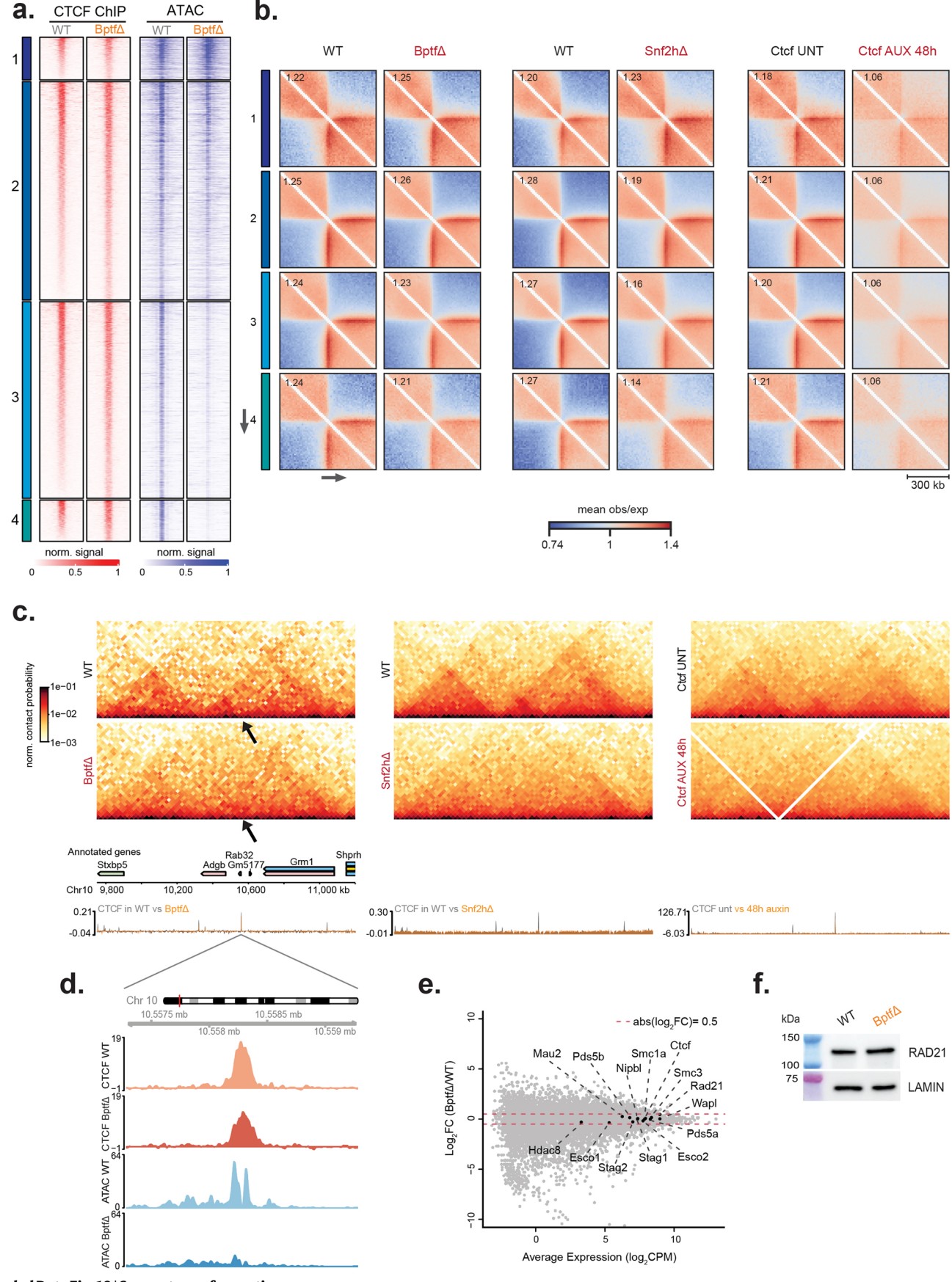

**Extended Data Fig. 10 | See next page for caption.**

**Extended Data Fig. 10 | NURF-mediated accessibility loss relates to loss of long-range chromatin interactions. a**. CTCF binding (ChIP–seq) and chromatin accessibility (ATAC-seq) signal in WT and BptfΔ cells within the four groups defined (as in Fig. 5a). Group number reported on the left. **b**. Mean observed/ expected contact frequency measured using Hi-C at CTCF split into the same groups as in Fig. 5a (group number on the left) in BptfΔ, Snf2hΔ and CTCF depleted cells (48 h) and their respective controls, at 10 kb resolution. Canonical motif orientation 5′–3′ indicated by the arrow. **c**. Hi-C heatmap at a representative locus illustrating loss of 3D contacts in BptfΔ, Snf2hΔ and CTCF depleted cells (48 h) in comparison to their controls, at 25 kb resolution. The black arrows

indicate a CTCF site showing loss of accessibility and 3D contacts despite persistent binding. Gene annotations and CTCF ChIP–seq tracks are shown below for reference; for the Snf2hΔ and CTCF depletion conditions, single replicates of ChIP–seq data are shown (data from refs. 14,30). **d**. CTCF ChIP and accessibility in the 2 kb region flanking the CTCF site highlighted in **c**. **e**. MA plot reporting variation in RNA in BptfΔ (y axis) vs average read counts (x axis). *Ctcf*, *Rad21* and cohesin complex components and regulators are highlighted to show their RNA variations upon BPTF depletion. **f**. Protein levels of RAD21 quantified by western blotting in WT and BptfΔ cells. Blot is representative of three experiments.

# Reporting Summary

## Statistics

For all statistical analyses, confirm that the following items are present in the figure legend, table legend, main text, or Methods section.

| n/a | Confirmed | |
|---|---|---|
| ☐ | ☒ | The exact sample size (*n*) for each experimental group/condition, given as a discrete number and unit of measurement |
| ☐ | ☒ | A statement on whether measurements were taken from distinct samples or whether the same sample was measured repeatedly |
| ☐ | ☒ | The statistical test(s) used AND whether they are one- or two-sided *Only common tests should be described solely by name; describe more complex techniques in the Methods section.* |
| ☐ | ☒ | A description of all covariates tested |
| ☐ | ☒ | A description of any assumptions or corrections, such as tests of normality and adjustment for multiple comparisons |
| ☐ | ☒ | A full description of the statistical parameters including central tendency (e.g. means) or other basic estimates (e.g. regression coefficient) AND variation (e.g. standard deviation) or associated estimates of uncertainty (e.g. confidence intervals) |
| ☐ | ☒ | For null hypothesis testing, the test statistic (e.g. *F*, *t*, *r*) with confidence intervals, effect sizes, degrees of freedom and *P* value noted *Give P values as exact values whenever suitable.* |
| ☒ | ☐ | For Bayesian analysis, information on the choice of priors and Markov chain Monte Carlo settings |
| ☐ | ☒ | For hierarchical and complex designs, identification of the appropriate level for tests and full reporting of outcomes |
| ☐ | ☒ | Estimates of effect sizes (e.g. Cohen's *d*, Pearson's *r*), indicating how they were calculated |

*Our web collection on statistics for biologists contains articles on many of the points above.*

## Software and code

Policy information about availability of computer code

| | |
|---|---|
| Data collection | Software used for data collection: For ChIPseq, Illumina RTA 1.18.64 and bcl2fastq2 v2.17 was used for basecalling and demultiplexing. For RNAseq, Illumina RTA 1.18.64 and bcl2fastq2 v2.17 was used for basecalling and demultiplexing samples generated by Illumina HiSeq sequencing. Illumina RTA 3.4.4 and bcl2fastq2 v2.20 was used for basecalling and demultiplexing samples generated by Illumina NovaSeq sequencing. For ATACseq, Illumina RTA 2.4.11 and bcl2fastq2 v2.17 was used for basecalling and demultiplexing samples generated by Illumina NextSeq. Illumina RTA 3.4.4 and bcl2fastq2 v2.20 was used for basecalling and demultiplexing samples generated by Illumina NovaSeq.  For MNaseseq, Illumina RTA 2.4.11 and bcl2fastq2 v2.17 was used for basecalling and demultiplexing samples generated by Illumina NextSeq. Illumina RTA 3.4.4 and bcl2fastq2 v2.20 was used for basecalling and demultiplexing samples generated by Illumina NovaSeq sequencing. For CUT&RUN, Illumina RTA 2.4.11 and bcl2fastq2 v2.17 was used for basecalling and demultiplexing. Protein identification and relative quantification was performed with MaxQuant v.1.5.3.8. |
| Data analysis | When required adaptor removal for genomic experiments was performed using cutadapt_2.5. Data analysis for genomic experiments was performed using R 4.3.0 and R/Bioconductor packages: BSgenome.Mmusculus.UCSC.mm10_1.4.0, TxDb.Mmusculus.UCSC.mm10.knownGene 3.10.0, QuasR_1.40.1, RBowtie_1.40.0, ComplexHeatmap_2.12.0., clusterProfiler 4.8.1, edgeR 3.40.2, MACS2_2.2.7.1,  swissknife 0.40, ggseqlogo_1.0, Keras_2.2.5.0,TensorFlow_ 2.0.0, JASPAR2022 0.99.7, monaLisa 1.6.0. For proteomics experiments protein identification and relative quantification was performed using MaxQuant_1.5.3.8, limma_3.56.2 was used for statistics. For Hi-C experiments, data analysis was peformed using using Python 3.9.7 and packages: BWA 0.7.17, cooler 0.9.0, cooltools 0.6.1, mustache 1.3.2, coolpup.py 1.1.0. |

For manuscripts utilizing custom algorithms or software that are central to the research but not yet described in published literature, software must be made available to editors and reviewers. We strongly encourage code deposition in a community repository (e.g. GitHub). See the Nature Portfolio guidelines for submitting code & software for further information.

## Data

Policy information about availability of data

All manuscripts must include a data availability statement. This statement should provide the following information, where applicable:

- Accession codes, unique identifiers, or web links for publicly available datasets
- A description of any restrictions on data availability
- For clinical datasets or third party data, please ensure that the statement adheres to our policy

Next-generation sequencing data generated in this study are available at Gene Expression Omnibus (GEO, https://www.ncbi.nlm.nih.gov/geo/) with accession no. GSE234295 and GSE250229. The following public datasets were obtained from GEO: histone marks ChIP–seq: H3K27me3: (GSE30203, samples GSM747539 to GSM747541) (ref. 75), H3K4me1: (GSE30203, sample GSM747542) (ref.75), H3K27ac: (GSE67867, samples GSM1891651 and GSM1891652) (ref.76), H3K36me3: (GSE33252, samples GSM801982 and GSM801983) (ref. 77); Hi-C CTCF-AID-UNT and CTCF-AUX-48h (GSE98671, samples GSM2644945 to GSM2644948) (ref.30); ChIP-seq CTCF-AID-UNT and CTCF-AUX-48h (GSE98671, samples GSM2609185 and GSM2609186) (ref. 30); SNF2H knockout: (GSE112136 Hi-C samples GSM3331341 to GSM3331344, MNase samples GSM3058339 to GSM3058342, RNA-seq samples GSM3058347 to GSM3058359, ChIP-seq samples GSM3058327 and GSM3058328) (ref.14). The UCSC annotation of known genes for mm10 was obtained through the Bioconductor annotation package TxDb.Mmusculus.UCSC.mm10.knownGene (10.18129/B9.bioc.TxDb.Mmusculus.UCSC.mm10.knownGene). The Jaspar2022 (ref.56) motif database used in this study can be accessed online (https://jaspar2022.genereg.net/).
The mass spectrometry proteomics data generated in this study have been deposited to the ProteomeXchange Consortium via the PRIDE78 partner repository with the dataset identifier PXD042945.
Source data are provided with this paper.

## Research involving human participants, their data, or biological material

Policy information about studies with human participants or human data. See also policy information about sex, gender (identity/presentation), and sexual orientation and race, ethnicity and racism.

| | |
|---|---|
| Reporting on sex and gender | n/a |
| Reporting on race, ethnicity, or other socially relevant groupings | n/a |
| Population characteristics | n/a |
| Recruitment | n/a |
| Ethics oversight | n/a |

Note that full information on the approval of the study protocol must also be provided in the manuscript.

# Field-specific reporting

Please select the one below that is the best fit for your research. If you are not sure, read the appropriate sections before making your selection.

☒ Life sciences      ☐ Behavioural & social sciences      ☐ Ecological, evolutionary & environmental sciences

For a reference copy of the document with all sections, see nature.com/documents/nr-reporting-summary-flat.pdf

# Life sciences study design

All studies must disclose on these points even when the disclosure is negative.

| | |
|---|---|
| Sample size | No statistical method was used to pre-determine sample size. All experiment were performed in at least 2 independent replicates in line with accepted practice in the genomics field. |
| Data exclusions | No data was excluded. |
| Replication | All experiments were performed in 2-4 biological replicates as indicated in the manuscript. All attempts at replication were successful. |
| Randomization | The study conditions did not require randomization as this study design is not affected by the order of recording for experimental data. |
| Blinding | The study conditions did not require blinding as this study design is not affected by the observer carrying out the experiments. |

# Reporting for specific materials, systems and methods

We require information from authors about some types of materials, experimental systems and methods used in many studies. Here, indicate whether each material, system or method listed is relevant to your study. If you are not sure if a list item applies to your research, read the appropriate section before selecting a response.

## Materials & experimental systems

| n/a | Involved in the study |
|-----|-----------------------|
| ☐ | ☒ Antibodies |
| ☐ | ☒ Eukaryotic cell lines |
| ☒ | ☐ Palaeontology and archaeology |
| ☒ | ☐ Animals and other organisms |
| ☒ | ☐ Clinical data |
| ☒ | ☐ Dual use research of concern |
| ☒ | ☐ Plants |

## Methods

| n/a | Involved in the study |
|-----|-----------------------|
| ☐ | ☒ ChIP-seq |
| ☒ | ☐ Flow cytometry |
| ☒ | ☐ MRI-based neuroimaging |

## Antibodies

| | |
|---|---|
| Antibodies used | Snf2h-ab72499-Abcam, Snf2h-ab3749-Abcam, Snf2l-D4Q7V-CellSignaling, CTCF-C-20X-Santa Cruz, CTCF-ab128873-Abcam, Rad21-ab992-Abcam, Rad21-154769-Abcam, Wapl-16370-1-AP-Proteintech, IgG-M7023_Sigma Aldrich, Acf1-A301-318A-Bethyl Laboratories, Cecr2-LSC-496852-LSBio, Wstf-ab51256-Abcam, Tip5-C15310090-Diagenode, Bptf-ABE24-Millipore, Rsf1-ab109002-Abcam, LaminB-ab16048-Abcam. |
| Validation | all antibodies were validated by the manufacturer. |

ab72499  https://www.abcam.com/products/primary-antibodies/snf2h-antibody-ab72499.html
D4Q7V https://www.cellsignal.com/products/primary-antibodies/smarca1-d4q7v-rabbit-mab/12483
ab3749 https://www.abcam.com/products/primary-antibodies/snf2h-antibody-chip-grade-ab3749.html
C-20X  https://www.scbt.com/p/ctcf-antibody-c-20
ab128873  https://www.abcam.com/products/primary-antibodies/ctcf-antibody-epr7314b-chip-grade-ab128873.html
ab992  https://www.abcam.com/products/primary-antibodies/rad21-antibody-ab992.html
ab154769 https://www.abcam.com/products/primary-antibodies/rad21-antibody-ab154769.html
16370-1-AP https://www.thermofisher.com/antibody/product/WAPL-WAPAL-Antibody-Polyclonal/16370-1-AP
M7023 https://www.sigmaaldrich.com/CH/en/product/sigma/m7023
A301-318A  https://www.thermofisher.com/antibody/product/ACF1-BAZ1A-Antibody-Polyclonal/A301-318A
LSC-496852 https://www.lsbio.com/antibodies/cecr2-antibody-wb-western-ls-c496852/510169
ab51256 https://www.abcam.com/products/primary-antibodies/wstf-antibody-ep1704y-ab51256.html
C15310090 https://www.diagenode.com/en/p/tip-5-polyclonal-antibody-classic-100-ul
ABE24  https://www.merckmillipore.com/CH/de/product/Anti-BPTF-Antibody,MM_NF-ABE24
ab109002  https://www.abcam.com/products/primary-antibodies/rsf1-antibody-epr37492-ab109002.html
ab16048 https://www.abcam.com/products/primary-antibodies/lamin-b1-antibody-nuclear-envelope-marker-ab16048.html

## Eukaryotic cell lines

Policy information about cell lines and Sex and Gender in Research

| | |
|---|---|
| Cell line source(s) | Mouse ES cell line of  129S6/SvEvTac background was originally obtained from Miriam Bibel and is described in PMID 17546008. Snf2hΔ line was generated in this background and is described in PMID 30996347. The new lines generated in this study namely Acf1Δ, Rsf1Δ, Tip5Δ, Cecr2Δ, WstfΔ,BptfΔ and Snf2lΔ were generated in the same background and are available upon request. |
| Authentication | Genotype of cell lines was tested at the level of DNA sequence and protein. |
| Mycoplasma contamination | Cell lines tested negative for Mycoplasm. |
| Commonly misidentified lines (See ICLAC register) | No commonly misidentified cell lines were used. |

## Plants

| | |
|---|---|
| Seed stocks | / |
| Novel plant genotypes | / |
| Authentication | / |

# ChIP-seq

## Data deposition

☒ Confirm that both raw and final processed data have been deposited in a public database such as GEO.

☐ Confirm that you have deposited or provided access to graph files (e.g. BED files) for the called peaks.

**Data access links**
*May remain private before publication.*

https://www.ncbi.nlm.nih.gov/geo/query/acc.cgi?acc=GSE234295 (SuperSeries - all data)
https://www.ncbi.nlm.nih.gov/geo/query/acc.cgi?acc=GSE234273 (ChIP-seq data)

https://www.ncbi.nlm.nih.gov/geo/query/acc.cgi?acc=GSE250229 (SuperSeries all data)
https://www.ncbi.nlm.nih.gov/geo/query/acc.cgi?acc=GSE250226 (ChIP-seq)

**Files in database submission**

For each ChIP-seq sample, the GEO entry contains two files: the rawdata (fastq format) and a file with alignment density per 100 bp in the mouse mm10 genome (wig file).

fastq files:
CTCF_input_BPTFko_rep1_1.fastq.gz
CTCF_input_BPTFko_rep2_1.fastq.gz
CTCF_input_BPTFko_rep3_1.fastq.gz
CTCF_input_BPTFko_rep3_2.fastq.gz
CTCF_input_BPTFko_rep4_1.fastq.gz
CTCF_input_BPTFko_rep4_2.fastq.gz
CTCF_input_wt_rep1_1.fastq.gz
CTCF_input_wt_rep2_1.fastq.gz
CTCF_input_wt_rep3_1.fastq.gz
CTCF_input_wt_rep3_2.fastq.gz
CTCF_input_wt_rep4_1.fastq.gz
CTCF_input_wt_rep4_2.fastq.gz
CTCF_IP_BPTFko_rep1_1.fastq.gz
CTCF_IP_BPTFko_rep2_1.fastq.gz
CTCF_IP_BPTFko_rep3_1.fastq.gz
CTCF_IP_BPTFko_rep3_2.fastq.gz
CTCF_IP_BPTFko_rep4_1.fastq.gz
CTCF_IP_BPTFko_rep4_2.fastq.gz
CTCF_IP_wt_rep1_1.fastq.gz
CTCF_IP_wt_rep2_1.fastq.gz
CTCF_IP_wt_rep3_1.fastq.gz
CTCF_IP_wt_rep3_2.fastq.gz
CTCF_IP_wt_rep4_1.fastq.gz
CTCF_IP_wt_rep4_2.fastq.gz
Rad21_input_BPTFko_rep1_1.fastq.gz
Rad21_input_BPTFko_rep1_2.fastq.gz
Rad21_input_BPTFko_rep2_1.fastq.gz
Rad21_input_BPTFko_rep2_2.fastq.gz
Rad21_input_BPTFko_rep3_1.fastq.gz
Rad21_input_BPTFko_rep3_2.fastq.gz
Rad21_input_wt_rep1_1.fastq.gz
Rad21_input_wt_rep1_2.fastq.gz
Rad21_input_wt_rep2_1.fastq.gz
Rad21_input_wt_rep2_2.fastq.gz
Rad21_input_wt_rep3_1.fastq.gz
Rad21_input_wt_rep3_2.fastq.gz
Rad21_IP_BPTFko_rep1_1.fastq.gz
Rad21_IP_BPTFko_rep1_2.fastq.gz
Rad21_IP_BPTFko_rep2_1.fastq.gz
Rad21_IP_BPTFko_rep2_2.fastq.gz
Rad21_IP_BPTFko_rep3_1.fastq.gz
Rad21_IP_BPTFko_rep3_2.fastq.gz
Rad21_IP_wt_rep1_1.fastq.gz
Rad21_IP_wt_rep1_2.fastq.gz
Rad21_IP_wt_rep2_1.fastq.gz
Rad21_IP_wt_rep2_2.fastq.gz
Rad21_IP_wt_rep3_1.fastq.gz
Rad21_IP_wt_rep3_2.fastq.gz
Wapl_IP_BPTFko_rep1_1.fastq.gz
Wapl_IP_BPTFko_rep1_2.fastq.gz
Wapl_IP_BPTFko_rep2_1.fastq.gz
Wapl_IP_BPTFko_rep2_2.fastq.gz
Wapl_IP_wt_rep1_1.fastq.gz
Wapl_IP_wt_rep1_2.fastq.gz
Wapl_IP_wt_rep2_1.fastq.gz

Wapl_IP_wt_rep2_2.fastq.gz
BPTF_IP_wt_rep1_1_read1.fastq.gz
BPTF_IP_wt_rep2_1_read1.fastq.gz
BPTF_IP_bptf_ko_rep1_1_read1.fastq.gz
BPTF_IP_bptf_ko_rep2_1_read1.fastq.gz
BPTF_input_wt_rep1_1_read1.fastq.gz
BPTF_input_wt_rep2_1_read1.fastq.gz
BPTF_input_BPTFko_rep1_1_read1.fastq.gz
BPTF_input_BPTFko_rep2_1_read1.fastq.gz
BPTF_IP_wt_rep1_1_read2.fastq.gz
BPTF_IP_wt_rep2_1_read2.fastq.gz
BPTF_IP_bptf_ko_rep1_1_read2.fastq.gz
BPTF_IP_bptf_ko_rep2_1_read2.fastq.gz
BPTF_input_wt_rep1_1_read2.fastq.gz
BPTF_input_wt_rep2_1_read2.fastq.gz
BPTF_input_BPTFko_rep1_1_read2.fastq.gz
BPTF_input_BPTFko_rep2_1_read2.fastq.gz

Wig files:
CTCF_input_BPTFko_rep1.wig.gz
CTCF_input_BPTFko_rep2.wig.gz
CTCF_input_BPTFko_rep3.wig.gz
CTCF_input_BPTFko_rep4.wig.gz
CTCF_input_wt_rep1.wig.gz
CTCF_input_wt_rep2.wig.gz
CTCF_input_wt_rep3.wig.gz
CTCF_input_wt_rep4.wig.gz
CTCF_IP_BPTFko_rep1.wig.gz
CTCF_IP_BPTFko_rep2.wig.gz
CTCF_IP_BPTFko_rep3.wig.gz
CTCF_IP_BPTFko_rep4.wig.gz
CTCF_IP_wt_rep1.wig.gz
CTCF_IP_wt_rep2.wig.gz
CTCF_IP_wt_rep3.wig.gz
CTCF_IP_wt_rep4.wig.gz
Rad21_input_BPTFko_rep1.wig.gz
Rad21_input_BPTFko_rep2.wig.gz
Rad21_input_BPTFko_rep3.wig.gz
Rad21_input_wt_rep1.wig.gz
Rad21_input_wt_rep2.wig.gz
Rad21_input_wt_rep3.wig.gz
Rad21_IP_BPTFko_rep1.wig.gz
Rad21_IP_BPTFko_rep2.wig.gz
Rad21_IP_BPTFko_rep3.wig.gz
Rad21_IP_wt_rep1.wig.gz
Rad21_IP_wt_rep2.wig.gz
Rad21_IP_wt_rep3.wig.gz
Wapl_IP_BPTFko_rep1.wig.gz
Wapl_IP_BPTFko_rep2.wig.gz
Wapl_IP_wt_rep1.wig.gz
Wapl_IP_wt_rep2.wig.gz
BPTF_IP_wt_rep1.wig.gz
BPTF_IP_wt_rep2.wig.gz
BPTF_IP_bptf_ko_rep1.wig.gz
BPTF_IP_bptf_ko_rep2.wig.gz
BPTF_input_wt_rep1.wig.gz
BPTF_input_wt_rep2.wig.gz
BPTF_input_bptf_ko_rep1.wig.gz
BPTF_input_bptf_ko_rep2.wig.gz

Genome browser session
(e.g. UCSC)

The following files can be uploaded to the UCSC genome browser by pasting the following URLs into "Paste URLs or data" in "add custom tracks".
http://www.fmi.ch/groupdata/gschub/BPTF/CTCF_input_BPTFko_rep1.bw
http://www.fmi.ch/groupdata/gschub/BPTF/CTCF_input_BPTFko_rep2.bw
http://www.fmi.ch/groupdata/gschub/BPTF/CTCF_input_BPTFko_rep3.bw
http://www.fmi.ch/groupdata/gschub/BPTF/CTCF_input_BPTFko_rep4.bw
http://www.fmi.ch/groupdata/gschub/BPTF/CTCF_input_wt_rep1.bw
http://www.fmi.ch/groupdata/gschub/BPTF/CTCF_input_wt_rep2.bw
http://www.fmi.ch/groupdata/gschub/BPTF/CTCF_input_wt_rep3.bw
http://www.fmi.ch/groupdata/gschub/BPTF/CTCF_input_wt_rep4.bw
http://www.fmi.ch/groupdata/gschub/BPTF/CTCF_IP_BPTFko_rep1.bw
http://www.fmi.ch/groupdata/gschub/BPTF/CTCF_IP_BPTFko_rep2.bw
http://www.fmi.ch/groupdata/gschub/BPTF/CTCF_IP_BPTFko_rep3.bw
http://www.fmi.ch/groupdata/gschub/BPTF/CTCF_IP_BPTFko_rep4.bw
http://www.fmi.ch/groupdata/gschub/BPTF/CTCF_IP_wt_rep1.bw
http://www.fmi.ch/groupdata/gschub/BPTF/CTCF_IP_wt_rep2.bw
http://www.fmi.ch/groupdata/gschub/BPTF/CTCF_IP_wt_rep3.bw
http://www.fmi.ch/groupdata/gschub/BPTF/CTCF_IP_wt_rep4.bw
http://www.fmi.ch/groupdata/gschub/BPTF/Rad21_input_BPTFko_rep1.bw
http://www.fmi.ch/groupdata/gschub/BPTF/Rad21_input_BPTFko_rep2.bw
http://www.fmi.ch/groupdata/gschub/BPTF/Rad21_input_BPTFko_rep3.bw
http://www.fmi.ch/groupdata/gschub/BPTF/Rad21_input_wt_rep1.bw
http://www.fmi.ch/groupdata/gschub/BPTF/Rad21_input_wt_rep2.bw
http://www.fmi.ch/groupdata/gschub/BPTF/Rad21_input_wt_rep3.bw
http://www.fmi.ch/groupdata/gschub/BPTF/Rad21_IP_BPTFko_rep1.bw
http://www.fmi.ch/groupdata/gschub/BPTF/Rad21_IP_BPTFko_rep2.bw
http://www.fmi.ch/groupdata/gschub/BPTF/Rad21_IP_BPTFko_rep3.bw
http://www.fmi.ch/groupdata/gschub/BPTF/Rad21_IP_wt_rep1.bw
http://www.fmi.ch/groupdata/gschub/BPTF/Rad21_IP_wt_rep2.bw
http://www.fmi.ch/groupdata/gschub/BPTF/Rad21_IP_wt_rep3.bw
http://www.fmi.ch/groupdata/gschub/BPTF/Wapl_IP_BPTFko_rep1.bw
http://www.fmi.ch/groupdata/gschub/BPTF/Wapl_IP_BPTFko_rep2.bw
http://www.fmi.ch/groupdata/gschub/BPTF/Wapl_IP_wt_rep1.bw
http://www.fmi.ch/groupdata/gschub/BPTF/Wapl_IP_wt_rep2.bw
http://www.fmi.ch/groupdata/gschub/BPTF/BPTF_IP_wt_rep1.bw
http://www.fmi.ch/groupdata/gschub/BPTF/BPTF_IP_wt_rep2.bw
http://www.fmi.ch/groupdata/gschub/BPTF/BPTF_IP_bptf_ko_rep1.bw
http://www.fmi.ch/groupdata/gschub/BPTF/BPTF_IP_bptf_ko_rep2.bw
http://www.fmi.ch/groupdata/gschub/BPTF/BPTF_input_wt_rep1.bw
http://www.fmi.ch/groupdata/gschub/BPTF/BPTF_input_wt_rep2.bw
http://www.fmi.ch/groupdata/gschub/BPTF/BPTF_input_bptf_ko_rep1.bw
http://www.fmi.ch/groupdata/gschub/BPTF/BPTF_input_bptf_ko_rep2.bw

## Methodology

Replicates | From two to four biological replicates were performed.

Sequencing depth
```
Sample_Name Total_reads Mapped_reads
CTCF_IP_wt_rep1       35192624 32224764
CTCF_IP_wt_rep2       39928904 36950512
CTCF_IP_BPTFko_rep1   39502118 36482367
CTCF_IP_BPTFko_rep2   38256394 35329755
CTCF_input_wt_rep1    34740264 32609634
CTCF_input_wt_rep2    19397729 18238325
CTCF_input_BPTFko_rep1 35408279 33264992
CTCF_input_BPTFko_rep2 20609623 19337074
CTCF_IP_wt_rep3       41275856 38403430
CTCF_IP_wt_rep4       37015816 34646028
CTCF_IP_BPTFko_rep3   40169605 36915807
CTCF_IP_BPTFko_rep4   38298894 35359970
CTCF_input_wt_rep3    29157171 27482691
CTCF_input_wt_rep4    33984356 32132314
CTCF_input_BPTFko_rep3 26673398 25246762
CTCF_input_BPTFko_rep4 32079806 30329742
Rad21_IP_wt_rep1      42831513 40228789
Rad21_IP_wt_rep2      39772331 37385655
Rad21_IP_wt_rep3      42121353 39523083
Rad21_IP_BPTFko_rep1  36729473 34371224
Rad21_IP_BPTFko_rep2  39770249 37327224
Rad21_IP_BPTFko_rep3  40226022 37740551
Rad21_input_wt_rep1   32985598 31124741
Rad21_input_wt_rep2   32082857 30210595
Rad21_input_wt_rep3   34328803 32405497
Rad21_input_BPTFko_rep1 34306857 32357374
```

Rad21_input_BPTFko_rep2 37288033 35147322
Rad21_input_BPTFko_rep3 35198271 33103882
Wapl_IP_wt_rep1        59577081 56016345
Wapl_IP_wt_rep2        35996719 33389039
Wapl_IP_BPTFko_rep1    51526126 48419132
Wapl_IP_BPTFko_rep2    42275419 39736154
Bptf_IP_wt_rep1 128961468 92409710
Bptf_IP_wt_rep2  113898972 81045910
Bptf_IP_BPTFko_rep1 155104492 111819052
Bptf_IP_BPTFko_rep2 120492214 90475200
Bptf_input_wt_rep1 82182150 69150740
Bptf_input_wt_rep2 88491534 73008342
Bptf_input_BPTFko_rep1 76505840 63308318
Bptf_input_BPTFko_rep2 74983422 63334034

**Antibodies**

CTCF-C-20X-Santa Cruz, Rad21-ab992-Abcam, Wapl-16370-1-AP-Proteintech, Bptf- ABE24-Millipore.

**Peak calling parameters**

*Specify the command line program and parameters used for read mapping and peak calling, including the ChIP, control and index files used.*

**Data quality**

ChIP-seq sample quality was assessed using the following criteria:
- technical quality (sufficient sequencing depth and unique-hit mapping rates)
- reproducibility between biological replicate samples

**Software**

Analysis was performed using R 4.3.0 and R/Bioconductor packages: QuasR 1.40.1.

