## [Peer Review File · Nature Genetics]

Peer Review Information

Manuscript Title: Systematic assessment of ISWI subunits reveals that NURF creates local accessibility for CTCF

Corresponding author name(s): Systematic assessment of ISWI subunits reveals that NURF creates local accessibility for CTCF

Reviewer Comments & Decisions:

Decision Letter, initial version:

11th Aug 2023

Dear Dirk,

hope this email finds you well.

Your Article, "Systematic assessment of ISWI subunits reveals that NURF creates local accessibility for proper CTCF function" has now been seen by 3 referees. You will see from their comments below that while they find your work of interest, some important points are raised. We are interested in the possibility of publishing your study in Nature Genetics, but would like to consider your response to these concerns in the form of a revised manuscript before we make a final decision on publication.

To guide the scope of the revisions, the editors discuss the referee reports in detail within the team, including with the chief editor, with a view to identifying key priorities that should be addressed in revision and sometimes overruling referee requests that are deemed beyond the scope of the current study. In this case, we would like you to address Reviewers' comments in full. We are committed to providing a fair and constructive peer-review process. Do not hesitate to contact us if there are specific requests from the reviewers that you believe are technically impossible or unlikely to yield a meaningful outcome.

We therefore invite you to revise your manuscript taking into account all reviewer and editor comments. Please highlight all changes in the manuscript text file. At this stage we will need you to upload a copy of the manuscript in MS Word .docx or similar editable format.

*1) Include a "Response to referees" document detailing, point-by-point, how you addressed each referee comment. If no action was taken to address a point, you must provide a compelling argument.

This response will be sent back to the referees along with the revised manuscript.

*2) If you have not done so already please begin to revise your manuscript so that it conforms to our Article format instructions, available here.

*3) Include a revised version of any required Reporting Summary:

Please be aware of our guidelines on digital image standards.

[redacted]

We hope to receive your revised manuscript within four to eight weeks. If you cannot send it within this time, please let us know.

Sincerely,
Chiara

Chiara Anania, PhD
Associate Editor
Nature Genetics
<https://orcid.org/0000-0003-1549-4157>

Referee expertise:

Referee #1: Higher order chromatin structure, Gene expression, Epigenetics

Referee #2: Molecular Biology, Genomics, Bioinformatics

Referee #3:

Reviewers' Comments:

Reviewer #1:

Remarks to the Author:

The overarching observation by Iurlaro and colleagues is that depletion of the NURF component BPTF reduces chromatin accessibility at a subset of CTCF sites with little effect on CTCF chromatin occupancy, but impairing CTCF's architectural function. The authors conclude that CTCF function thus depends in part on its surrounding chromatin environment.

The authors start by investigating the role of individual ISWI subunits at the level of transcription, chromatin accessibility and CTCF factor binding in mouse embryonic stem cells. This follows their previous findings (PMID: 30996347) that the ISWI ATPase SNF2H has a role in nucleosomal phasing, is essential for normal binding of the transcription factor CTCF, and that reduction of CTCF binding (upon SNF2H KO) leads to reduced insulation of topological associated domains (TADs).

While most of the ISWI components tested have relatively little effect on the features mentioned above, the NURF-specific subunit BPTF is identified as a cofactor of SNF2H that influences nucleosome occupancy predominantly at distal DNase hypersensitive sites (DHS) but not at TSS and concentrated at CTCF bound sites. Importantly, BPTF Δ reduces chromatin accessibility around CTCF bound sites but in contrast to SNF2H Δ has little impact on CTCF binding. The authors exploit this finding to ask whether such chromatin changes influence CTCF function. Hi-C analysis suggests that the CTCF insulation function is impaired in a manner commensurate with changes in accessibility.

While the data presented are intriguing and generally well presented, the effect sizes are quite modest. Importantly, the results allow for alternative conclusions: Firstly, the subtle loss in "insulation" at CTCF sites in BPTF Δ cells may be technical in nature, such as, for example, restriction enzyme accessibility bias or ligation bias changes at these regions in BPTF Δ cells. Secondly, a very possible alternative explanation is that CTCF-adjacent BPTF-dependent transcription factors account for the reduction in looped contacts and changes in cohesin recruitment in BPTF Δ cells. The nucleosome changes in BPTF Δ cells occur at enhancers rich in TF binding. In other words, CTCF/nucleosome position-independent functions may account for insulation and chromosomal contacts at CTCF sites.

1. Fig.2b,c. Please show Snf2h Δ in the same panel for comparison.
2. In order to interpret the data, such as the selectivity of BPTF for DHS vs TSS, BPTF ChIP-seq or CUT&RUN data should be included.
3. Line 157: "Since a large fraction of distal DHSs in mammalian genomes are sites of CTCF binding". Please explain how this was assessed. What fraction of DHS are bound by CTCF and what size window was applied?
4. The sub-header "NURF-component BPTF mediates accessibility upon CTCF binding" is unsupported.

To demonstrate BPTF dependence on CTCF requires degradation of CTCF or editing CTCF binding sites. CTCF degron cell lines are in circulation and this is a doable experiment.

5. Please simplify for the reader. How many cluster 2 regions are there and how many have CTCF sites and within what radius? How many CTCF sites are unchanged upon BPTFΔ and Snf2hΔ? How does this compare to other transcription factors?

6. Fig.3: please show whether CTCF binding to chromatin is also affected at sites not co-bound by SNF2H.

7. This reviewer struggles with the interpretation of the footprinting experiments. The window shown is only 400 base pairs which corresponds to maximally 2 nucleosomes, perhaps less depending on the size of the nucleosome-free region. What is being measured here? What is the definition of the “flanking region”? Can this analysis include a broader window?

8. A major open question is whether loss of accessibility near CTCF sites in BPTFΔ cells is due to loss of nearby transcription factor binding. A motif search around relevant CTCF sites might be an entry point to address this issue.

9. Please clarify and reconcile these two findings: SNF2H binding at distal CTCF sites is BPTF dependent, and BPTFΔ has little impact on CTCF binding while SNF2HΔ does.

10. Fig.4c: The Hi-C data are described insufficiently. What happened to compartment boundaries, TADs, loops etc in BPTFΔ cells? How was a “reduction in long-range interactions” defined? It would be helpful to stratify the CTCF anchors against all fragments, CTCF with other CTCF sites, CTCF with TSS.

11. Insulation changes seem modest at best. In fact, in extended data Fig.8d it appears that insulation is stronger for group 1 CTCF sites in BPTFΔ cells. It would be helpful to show some examples on the Hi-C heat maps.

12. Also, changes in insulation scores can be driven by changes in intra- or inter-TAD interactions. For example, a reduction in enhancer-gene contacts in gene rich TADs can manifest in reduced insulation. Fewer intra-TAD interactions in BPTFΔ cells are not a reflection of CTCF insulation function.

13. Related to point 12: The CTCF sites in question are at distal enhancers. Are these enhancers at TAD boundaries? Are their target genes weakened in BPTFΔ cells?

14. Reduction in insulation is expected to result in inter-TAD contacts and perhaps illegitimate enhancer-promoter contacts. Can gene expression changes in BPTFΔ cells be accounted for in this manner?

15. Extended data Fig.8c,d: Why are the pile ups asymmetrical? What are the stripes at the bottom? Why are there no stripes at the top?

16. Line 319: “BPTF-mediated accessibility around CTCF sites is required for efficient insulation”. This sentence implies causation but is in fact a correlation, and accessibility may not be “required”. The effect on chromatin contacts or insulation may be mediated by TFs that contribute to contacts directly or indirectly via cohesin loading.

Minor:

1. Extended Data Fig. 4e could be made a main figure because it displays data central to this report.

2. Extended Data Fig. 6a: “log2 fold-change IP/input”. What “change”?

3. Why are cohesin and CTCF ChIP-seq peaks doublets? This is unusual.

4. Line 243: replace “loose” with “lose”

5. Break up extended data Fig.4b into individual panels.

6. Lines 293 and 368: “Affinity” to the motif has not been measured in this study and other mechanisms may account for the persistent CTCF binding in BPTFΔ cells. “CTCF affinity for its motif seems to be crucial in retaining binding in absence of NURF” is an overstatement.

7. I have difficulty in interpreting this sentence: "However, we are not aware of an example where absence of a cofactor causes loss of chromatin opening yet, largely persistent TF binding as we show here for CTCF in absence of BPTF". There are many cases of TFs binding to regions of closed chromatin with no ATAC signal or any other indications of chromatin remodelling co-factors.

Reviewer #2:

Remarks to the Author:

In this manuscript, Iurlaro/Masoni et al investigate the function of the ISWI subunits RSF1, ACF1, WSTF, TIP5, CECR2 and BPTF, by making individual loss-of-function mutations in mESCs. Although most deletions have only moderate effects on nucleosomal structure and gene expression, the authors identify strong changes in chromatin accessibility around CTCF sites after mutating the BPTF subunit (part of the NURF complex). However, surprisingly, the authors find that these changes in nucleosomal structure only modestly impact binding of CTCF. Despite these modest changes in CTCF binding, the authors show that loss of BPTF leads to changes in occupancy of Cohesin and associated changes in 3D genome organization. The authors therefore speculate that local accessibility (and not CTCF binding per se) is critical for Cohesin binding and insulation of chromatin domains.

This manuscript presents interesting experiments that give new insight into the function of ISWI remodelers. The data are of high quality, include all relevant controls, are well presented, and clearly described. The finding that BPTF regulates accessibility around CTCF-binding sites without having a major impact on CTCF-binding (in contrast to SNF2H) is convincing and interesting. It also seems clear that loss of BPTF leads to some changes in genome organization. However, I am not convinced that the conclusion that CTCF binding is separated from insulator function is supported by the data.

Major comments

1. In Figure 4c, the authors show changes in nuclear organization in BPTF-mutant cells as compared to WT cells. Cluster 1 (n=9707) shows that when there is no change in CTCF binding and a very modest change in accessibility, there are no clear changes in insulation. Cluster 2 shows that when there is no change in CTCF binding and a moderate change in accessibility (n=10377), there are also no clear changes in insulation. Cluster 3-5 (n=12111+8752+6683=27546) show that when there is a reduction in both CTCF-binding and accessibility, there is a clear reduction in insulation.

In Figure 5b, the authors focus on chromatin insulation changes around CTCF-binding sites at which CTCF occupancy does not change strongly in absence of BPTF. Cluster 1 and 2 show that when there are no/very modest changes in accessibility (n=1358+6941=8299), there is no clear effect on insulation. Cluster 3 shows that a moderate reduction in accessibility (n=6253) is associated with an extremely subtle effect on insulation. Cluster 4 shows that a strong reduction in accessibility (n=1313) is associated with a stronger reduction in insulation.

Although Cluster 4 in Figure 5b clearly shows a reduction in insulation, I think it is important to stress that this is only observed at 1313 sites. Moreover, there is a small reduction in CTCF binding in these sites as well.

Furthermore, Cluster 2 in Figure 4c shows 10377 sites at which there is an almost equally strong reduction in accessibility as in Cluster 3 in Figure 5b, but in Figure 4c/Cluster 2 there are no clear reductions in insulation (i.e. the lower-left and upper-right corners seem to contain roughly as many pink regions as blue regions).

To me it seems that the most numerous and strongest changes in insulation that result from BPTF loss

are associated with a reduction in both CTCF binding and accessibility (Cluster 3-5 in Figure 4c). The conclusion that "These results separate local CTCF binding from insulator function in nuclear organization" therefore seems overstated to me and not fully supported by the current data.

2. Related to the comment above, it would be interesting to directly compare the Hi-C data in the BPTF mutant to the Hi-C data in the SNF2H mutant that was published by the authors in 2019. This may help in determining whether CTCF binding or accessibility is driving insulation. Since the reduction in CTCF binding is larger in the SNF2H mutant compared to the BPTF mutant but the changes in accessibility are similar, a model in which CTCF binding determines insulation would predict that the changes in insulation are larger in the SNF2H mutant compared to the BPTF mutant. Conversely, if insulation is dependent on accessibility and not on CTCF binding, the changes in insulation in the two datasets should be similar.

3. A prominent feature of CTCF function in genome organization is the formation of "CTCF loops" between the CTCF-binding sites at the boundaries of domains. It would be very interesting to analyze if there are any changes in the formation of these loops in absence of BPTF in the Hi-C data.

4. The authors show that mutation of BPTF leads to 693 up- and 1143 down-regulated genes. Are these changes related to the impaired CTCF function in absence of BPTF, i.e. are they associated with changes in genome organization, such as enhancer-promoter interactions?

Minor comments

1. The right panel in Figure 2b is a bit hard to read, but seems to show that nucleosome positioning at TSSs is more distinct in the *Cecr2* mutant. Can the authors comment on this observation?

2. In line with major comment #1, it would be interesting to see the RAD21 and WAPL occupancy aligned with the matrices in Figure 4c as well.

Reviewer #3:

Remarks to the Author:

My expertise lies in chromosome folding, CTCF biology and Hi-C analyses. I therefore was able to review these aspects of the paper with greater attention than the chromatin accessibility data.

This manuscript contains many exciting and novel observations clarifying the role of chromatin remodelers creating chromatin accessibility after transcription factor binding. I found the decoupling of CTCF binding and accessibility in the BPTF mutant especially exciting and compelling – even if it only occurs at a very small number of CTCF binding sites. These results illustrate how transcription factor binding is not always sufficient to create accessibility, which sometimes requires recruitment of dedicated remodelers. These observations have far-reaching implications beyond CTCF biology. I only have minor suggestions, mainly to clarify effect size and some of the Hi-C analyses.

Main suggestions

1. It is very difficult to assess the effect size of genome misfolding from the data presented. Below are some suggestions to help clarify this point.

a) Rewording. The way the abstract is phrased make it sound like one should expect insulation to be impaired at most if not all CTCF sites in BPTF mutants. Yet, as authors show in figures 4 and 5, this is not the case at all. In fact, only 8% (1313) of CTCF sites with unchanged binding (group 4 of figure 4) seem to display insulation alterations. This represents less than 2.8% of all CTCF binding sites. This number is comparable to the fraction of CTCF sites where binding is most decreased (14% cluster 5 of figure 4), yet here authors concluded that CTCF binding is only modestly decreased (abstract). As the analyses presented indicate, genome misfolds but only a handful of CTCF sites. In a nutshell some rewording of the abstract and the manuscript would be helpful to help the reader better appreciate the scale and (limited) pervasiveness of the folding defects.

b) It is quite difficult to appreciate the effect size of genome misfolding from the figures. Brief inspection of the Hi-C heatmaps generated from the .cool file provided (without separating the clusters) does not display pervasive defects – contrasting with the SNF2H or CTCF-AID mutants previously reported. Additional Hi-C metrics could help provide comparative numbers. For example, reporting level of insulation in figure 4c and 5b etc. (e.g. signal enrichment in the top right square) in WT vs KO would help better understand the magnitude of the defects. Showing heatmaps of the actual signal, as opposed to only showing the log2FC of obs/exp, would also help the reader get a sense of how much change compared to baseline is occurring.

Alternatively, aggregate peak analyses centered at the CTCF site clusters/groups of figure 4 and 5 could help understand better the fold change in Hi-C signal between WT and KO.

2. To further help readers appreciate the extent of the Hi-C changes – especially in relation to SNF2H KO and CTCF depletion:

A) it would be especially insightful to report the change in insulation as scatter plots, as authors did in their previous Nature article on SNF2H (fig4a).

B) it would also be most insightful to show at least one Hi-C heatmap with the signal for each genotype (not just the differential heatmaps)

C) adding a supplementary plot of aggregate peak analysis centered at Hi-C loops found in WT (possibly taking a ultra-deep reference loop list – e.g. Bonev et al. 2017) would be helpful. This would help readers understand the extent of looping loss upon BPTF deletion. Again, it could help to show the signal in each genotype in addition to a heatmap reporting the difference (subtraction of signal, not a Z-score or other observed/expected metrics).

D) Adding direct comparison to SNF2H would be insightful. If authors also performed Hi-C upon CTCF depletion (e.g. degron cells available) it would enable readers to understand the quantitative contribution of BPTF to the insulation functions of CTCF. This is not strictly necessary if authors have not already performed Hi-C after CTCF degradation.

3. There seem to be a problem with the Hi-C data deposited on GEO, specifically with GSM7461658_HiC_BPTFko_rep1_matrix.cool.gz

This sample appears to have been normalized in a different way than the other three samples. See attached figures.

It looks like the problem comes from the balancing of the cooler file. When I merged the two replicates and re-balanced the merged file it solved the issue. Please double check the processing of the file for BPTFko_rep1 and update the analyses that may have used it.

Please also make sure to provide on GEO the actual .cool/.mcool files that were used for the analyses presented in the paper (I assume a file containing pooled replicates was used?) – same goes for the

ChIP/MNase/RNA-seq processed files.

Minor suggestions

1. I found the main text description of figure 4c somewhat misleading. Authors state that "CTCF sites with increased loss of accessibility show more significant reduction in long-range interactions" [line 316]. Yet only clusters 3, 4 and 5 display loss of insulation. These are the clusters where CTCF binding is reduced in Bptf mutants. Cluster 2, where CTCF binding is preserved but accessibility is reduced, shows minimal Hi-C defects in panel c. This would argue that changes in Hi-C insulation are attributable to the reduction of CTCF occupancy, rather than some downstream process such as cohesin retention. Authors need to clarify this point or reword this section – or simply wait for the description of figure 5 to make this statement.
2. Any explanation why SNF2H KO caused loss of CTCF binding while none of the subcomplexes? Authors state that "Individual ISWI subunits do not account for the nucleosomal phenotype caused by loss of catalytic activity" – Could authors speculate then about what does? It is very intriguing that none of the subunit deletion recapitulate the NRL increase observed upon SNF2H deletion. It would be interesting if authors could speculate about possible mechanisms.
3. Have authors performed Hi-C and MNase/ATAC-seq SNF2L KO? Does it recapitulate BPTF KO? A simple statement saying whether authors have tried to generate these cells would be helpful (especially if authors already know whether the SNF2L KO cells are viable or not)
4. How does the Hi-C compare between SNF2H (previous Nature article from the authors), BPTF (present data) and, if possible/relevant SNF2L (if these cells are viable)?
5. Given that authors report that BPTF can associate with SNF2H, to what extent the loss of insulation initially reported in the SNF2H KO can be attributed to the BPTF-dependent function of CTCF in cohesin blocking, as opposed to loss of CTCF binding?
6. Authors conclude that "the deletion lines are subcomplex-specific (Extended Data Fig.1). Yet on the Western blot figure 1b it looks like that some deletions affect the other subunits. For example CECR2 displays a strong extra band in the ACF1 deletant, and the stoichiometry of the ACF1 subunits appear to change in the TIP5 and BPTF deletant. Authors may wish to comment on this, either to say that deletion of one subunits consistently affects the other ones (if this effect is reproducible), or to say that these variations are spurious (if effect is not reproducible).
7. Authors ascribe the reduction in MNase accessibility in BPTF at distal DHS to the presence of CTCF sites. However they do not report how MNase patterns change at distal DHS that do not have CTCF. I suggest replacing Ext fig 4c with the non-CTCF distal DHSs. I found this finding especially important for the rest of the paper, and would suggest considering to move Ext fig 4b and c (once replaced with the non-CTCF distal DHSs) to the main figures. If space permits.

8. Authors did not use ChIP-seq spike-ins. It would be helpful if authors could briefly explain to what extent this might affect their conclusions regarding the absolute changes in binding measured in the various mutants. It would also be helpful to explain if the CUT&RUN suffers from the same limitations. Redoing all the experiments with spike-ins would not be necessary at this stage.

9. I found that figure 2 does not clearly convey what happens to CTCF binding. Authors show convincing cherry-picked loci in 2b, but an additional panel reporting the number of peaks in each category (e.g. unaffected, reduced, completely lost, enhanced...) would help readers understand better the general trends. For example authors could add a series of panels splitting the line graphs currently present in figure 3a into several groups (e.g. unaffected, reduced, completely lost, enhanced...) and label the number or fraction of sites in each categories. Another option would be moving Extended data fig 8a in the main figure 2, and indicating the number of CTCF peaks in each cluster.

10. The manuscript would benefit from additional comparisons of the transcriptome datasets. For example a Principal component plot with the genotypes of figure 1d might be helpful?

11. In addition to the GO term analysis enrichment it may be interesting to report on the genomic context of the differentially expressed genes. For example, how many are Polycomb targets? How many had CTCF binding at the promoter? Etc...

12. Line 149 "suggesting either a reduced role for ISWI in the formation and maintenance of nucleosome structure" Reduced compared to what? Do authors simply mean 'minor'?

13. It would be helpful to clarify if ext fig 4e contains all of the differentially ATAC regions or just the ones with lost accessibility. Adding the differential ATAC signal as an extra column might be helpful if it ends up lining up with the clusters.

14. It would be helpful to report in the text the fraction of affected ATAC regions that have CTCF. Judged from figure 4e it is about 50% - which means that other factors seem to depend on BPTF for creating/maintaining accessibility. Also here it would be most helpful to report in a supplementary table the position of the differentially accessible regions.

15. Reporting the expression of CTCF and cohesin regulatory subunits from the RNA-seq would help ascertain that the effects of depleting BPTF are not simply due to mis-expression of these genes.

16. It would be most useful if authors provided the list of DHSs and differential DHSs they used in a supplementary table.

17. Please include a supplementary table with the list of genomic positions in each cluster and group used in figures 4 and 5

18. Please include the mapping statistics for the NGS experiments as a supplementary table (at least final used reads / read-pairs)

19. In Extended figure 7f it would be helpful to have a y axis scale for the gray triangles that denote

the log2FC of CTCF binding in BPTF vs WT

20. The “cell line generation and maintenance” section in the method is missing critical information.

- a) Authors indicate that several clones with several sgRNAs were generated to account for possible off-targeting. It would be most helpful if authors could clarify how many clones were used for how many experiments, and provide a nomenclature (e.g. a unique ID# for each cell line) enabling them to report which experiment was performed with which clones. A table with a unique ID# for each clone and the experiment(s) it was used for would suffice. This would also facilitate future resource sharing.
- b) Authors indicate that separate gRNAs were used to assess off-target and clonal effects yet only one sgRNA sequence per gene target is provided (except for Acf1). Please clarify
- c) It would be helpful if authors provided more technical details such as amount of cell and DNA transfected, which vector(s) carried the puromycin selection whether single colonies were obtained by FACS or limited dilution + manual picking etc...
- d) Authors need to report the sequence of the mutated alleles for gene in each clone used throughout the paper
- e) Transfections needs more detailed protocol for reproducibility (e.g. cell number, plating scheme, DNA and lipofectamine concentrations etc. etc. etc.). Right now it would be impossible to reproduce the experiment with the information provided.

21. Tip5 KO was reported to be lethal to mESCs grown in 2i+LIF medium – but not in serum+LIF (Dalcher santoro biorxiv 2019). I wonder if authors reproduce this phenotype and how the various KOs reported here behave in 2i+LIF. This may unmask essential roles for a specific ISWI subcomplexes and clarify why these subunits are only essential in specific growth conditions. These experiments may be left out for a follow-up study.

Best wishes,
Elphège Nora

Author Rebuttal to Initial comments
--

Reviewers' Comments:

Reviewer #1:

Remarks to the Author:

The overarching observation by Iurlaro and colleagues is that depletion of the NURF component BPTF reduces chromatin accessibility at a subset of CTCF sites with little effect on CTCF chromatin occupancy, but impairing CTCF's architectural function. The authors conclude that CTCF function thus depends in part on its surrounding chromatin environment.

The authors start by investigating the role of individual ISWI subunits at the level of transcription, chromatin accessibility and CTCF factor binding in mouse embryonic stem cells. This follows their previous findings (PMID: 30996347) that the ISWI ATPase SNF2H has a role in nucleosomal phasing, is essential for normal binding of the transcription factor CTCF, and that reduction of CTCF binding (upon SNF2H KO) leads to reduced insulation of topological associated domains (TADs). While most of the ISWI components tested have relatively little effect on the features mentioned above, the NURF-specific subunit BPTF is identified as a cofactor of SNF2H that influences nucleosome occupancy predominantly at distal DNase hypersensitive sites (DHS) but not at TSS and concentrated at CTCF bound sites. Importantly, BPTF Δ reduces chromatin accessibility around CTCF bound sites but in contrast to SNF2H Δ has little impact on CTCF binding. The authors exploit this finding to ask whether such chromatin changes influence CTCF function. Hi-C analysis suggests that the CTCF insulation function is impaired in a manner commensurate with changes in accessibility.

While the data presented are intriguing and generally well presented, the effect sizes are quite modest. Importantly, the results allow for alternative conclusions:

Firstly, the subtle loss in “insulation” at CTCF sites in BPTF Δ cells may be technical in nature, such as, for example, restriction enzyme accessibility bias or ligation bias changes at these regions in BPTF Δ cells.

R1. The reviewer is concerned that chromatin accessibility changes might impact the Hi-C readout. It is important to take into account that we also performed RAD21 and WAPL ChIP-seq in the same conditions as for Hi-C. These independent genome-wide experiments reveal a reduction in both factors specifically at CTCF sites that lose chromatin accessibility (Fig. 5c), in line with the reduction in insulation observed at the same sites (Fig. 5b). This independently supports our interpretation that loss of insulation is not artificial but rather a direct consequence of decreased stalling of the loop-extrusion

activity of cohesin at CTCF sites whose chromatin accessibility is compromised in the absence of BPTF. We nonetheless agree with the reviewer that BPTF depletion only has modest effects on chromosome structure genome-wide. To give a more balanced account of the effect size of BPTF depletion on structural features detected in Hi-C, we have introduced several changes in the figures and main text. We notably expanded the analysis of Hi-C datasets including comparison with other mutants leading to more prominent loss of insulation at CTCF sites, notably *Snf2h* Δ and CTCF degradation (newly added Fig 4c-f, Extended Data Fig. 10c-h, Extended Data Fig. 11, Fig. 5b, Extended Data Fig. 12c-d). Importantly however, comparisons of Hi-C replicates show no changes in insulation: thus, albeit small, the effects we observe following BPTF deletion are not due to technical reasons (Fig. 4c-e, Extended Data Fig. 10c).

Secondly, a very possible alternative explanation is that CTCF-adjacent BPTF-dependent transcription factors account for the reduction in looped contacts and changes in cohesin recruitment in BPTF Δ cells. The nucleosome changes in BPTF Δ cells occur at enhancers rich in TF binding. In other words, CTCF/nucleosome position-independent functions may account for insulation and chromosomal contacts at CTCF sites.

R2: The reviewer raises a point that we already partially addressed in the previous submission, where we showed that less than 10% of sites that show changes in accessibility (cluster 2-5) localize to enhancer regions as defined by ChromHMM (Fig. 4b). While this illustrates that most affected sites do not tend to localize at regions rich in TF binding, we nevertheless explored this further by intersecting the ENCODE annotation of candidate cis regulatory elements (cCREs, Nature 2020) with the respective CTCF sites (Revision Fig. 1a). This analysis revealed that most CTCF sites were either classified as (a) CTCF sites bound alone (CTCF-only/CTCF-bound, out of annotated enhancer/promoters) or (b) CTCF sites bound in distal enhancer like sequences defined using TSS distance and H3K27ac (dELS, CTCF-bound) (Revision Fig. 1a, below). Thus,

we asked whether CTCF binding and accessibility responses upon BPTF depletion differs at these categories (Revision Fig. 1b). This reveals that, CTCF sites overlapping all annotated cCRE had a similar cluster distribution as the sites originally described in the manuscript. Importantly, CTCF sites bound alone were underrepresented in cluster 1 (sites with less changes in both accessibility and binding). In contrast, CTCF sites bound within distal enhancers were overrepresented in cluster 1 (Revision Fig. 1b). Thus, persistent sites (cluster 1) tend to lie in enhancer-like elements rather than being CTCF only bound as shown in Fig. 4b. Additionally, we performed long-range contacts analysis at these regions, separately for each CTCF site category as defined above (newly added Extended Data Fig. 11b). This analysis showed that Hi-C contacts diminished progressively from cluster 1 to 5 in both CTCF only bound sites and CTCF sites bound in distal enhancer as for CTCF overlapping all cCRE. This suggests that changes in chromatin contacts are driven by CTCF itself rather than other elements in the enhancers. We hope the reviewer agrees that these additional analysis and illustration make this point more convincing.

Revision Figure 1. Intersection of CTCF sites with annotated candidate cis regulatory elements

a. Annotation of candidate cis regulatory elements (cCRE, ENCODE 2020) overlapping CTCF bound regions used in the manuscript defines as most prominent CTCF bound-only regions and CTCF bound in distal enhancer like sequence regions (highlighted in red). Annotations: CTCF-only, CTCF-bound (CTCF bound regions outside of promoters and enhancer); dELS, CTCF-bound (CTCF bound regions within distal enhancer like sequences); DNase-H3K4me3, CTCF-bound (high DNaseI regions with high H3K4me3 but low H3K27ac signals outside annotated TSS); pELS, CTCF-bound (CTCF bound regions within enhancer like sequences proximal to TSSs); PLS, CTCF-bound (CTCF bound regions within promoter like sequences). **b.** CTCF sites overlapping all annotated cCREs, overlapping dELS cCREs, overlapping CTCF-only bound cCREs divided by clusters as for Fig. 4a.

1. Fig.2b,c. Please show *Snf2h* Δ in the same panel for comparison.

R3: We now report in the newly added Extended Data Fig. 5a-b, the MNase signal for SNF2H deletion and WT (data from Barisic et al., 2019) together with a panel for BPTF deletion and WT as comparison. Data are displayed separately for *Snf2h* and the WT of the same batch as the experiments from the 2019 study were done via single-end sequencing while the samples sequenced for this study are all paired-end, a difference that affects the read shift used in this analysis.

2. In order to interpret the data, such as the selectivity of BPTF for DHS vs TSS, BPTF ChIP-seq or CUT&RUN data should be included.

R4: We now provide BPTF localization in mESCs using ChIP-seq. We note that genomic localization of remodeler components has largely been proven more difficult than of e.g., histone modifications or transcription factors thus we performed BPTF ChIP-seq not only in WT but also in BPTF KO cells as a rigorous negative control. This reveals indeed specific signal for BPTF at CTCF sites in both distal DHSs and promoter regions in a BPTF dependent manner as it is reduced in the knockout cells (newly added Extended Data Fig. 7a-b). We conclude that BPTF localizes to CTCF sites in line with a recent report published during the revision (Radzishchenskaya et al., 2023). Additionally, we report BPTF localization at CTCF sites overlapping both promoters and distal DHSs which makes us speculate that the chromatin changes seen specifically at distal elements could be due to a potential redundancy of remodelers at TSSs.

3. Line 157: “Since a large fraction of distal DHSs in mammalian genomes are sites of CTCF binding”. Please explain how this was assessed. What fraction of DHS are bound by CTCF and what size window was applied?

R5: Intersecting distal DHSs (using each DHS width without further extension) with CTCF bound sites (CTCF motif of 20 bp bound at least two-fold in our dataset) reveals that about one-third of the DHSs overlaps a bound CTCF site (34%). This is now specified in the text at line 169 as suggested.

4. The sub-header “NURF-component BPTF mediates accessibility upon CTCF binding” is unsupported. To demonstrate BPTF dependence on CTCF requires degradation of CTCF or editing CTCF binding sites. CTCF degron cell lines are in circulation and this is a doable experiment.

R6: While it is generally assumed that transcription factors recognize their target motif first before remodeler activity, the reviewer is correct that we do not formally prove a stepwise model. We thus changed the sub header to “CTCF sites lose accessibility yet largely remain bound in the absence of BPTF” to avoid the impression of overinterpretation and corrected the related part in the discussion (line 437). We further note that showing that accessibility around CTCF sites is dependent on CTCF as suggested would only reveal (an expected) genetic dependency but would similarly not allow a formal conclusion on temporal order.

5. Please simplify for the reader. How many cluster 2 regions are there and how many have CTCF sites and within what radius? How many CTCF sites are unchanged upon BPTF Δ and Snf2h Δ ? How does this compare to other transcription factors?

R7: We have now added information about the number of regions in each cluster in the main text at line 180 (cluster1 = 36386, cluster2 = 40166, cluster3 = 12426). As evident from the visualization (Extended Data fig. 6b), only cluster 2 shows clear enrichment for CTCF sites. In fact, 47.5% of cluster 2 regions have a bound CTCF motif in them (considering the ATAC peak width without further extensions). This is in stark contrast with cluster 1 and 3, where bound CTCF sites correspond to less than 1% of the regions. This is also now clarified in the text at line 193.

Upon BPTF and SNF2H depletion, CTCF bound sites which do not lose accessibility represent 9.6% and 14.3% of the total number of sites (these have been defined as sites with accessibility loss higher than 20% of the original ATAC signal). We have now added

this information in the main text at line 203.

Regarding other transcription factors affected by BPTF depletion we refer to Extended Data Fig. 6c where we took an unbiased approach and performed motif analysis on the regions called as differentially accessible divided by cluster as in Fig. 2d. This analysis reveals that no other motif displays an enrichment comparable to CTCF motif within cluster 2. We thus conclude that we do not find evidence at the level of accessibility, for a contribution of other transcription factor motifs comparable to the one of CTCF. Our conclusions are compatible with other studies in different mouse and human cellular models at a subset of individual CTCF sites (Qiu et al., 2015), the phenotype upon depletion of BPTF in leukemic cells (Radzisheuskaya et al., 2023, published during the revision) as well as reported nucleosomal changes upon siRNA depletion of different ISWI accessory subunits (Wiechens et al., 2016). These results are in line with the fact that many affected sites do not reside in enhancers but in CTCF only bound DHSs (also see R2).

6. Fig.3: please show whether CTCF binding to chromatin is also affected at sites not co- bound by SNF2H.

R8: To address this question, we now compared SNF2H to CTCF localization in WT vs BPTF knockout (Revision Fig. 2). This reveals that SNF2H always localizes to CTCF bound sites and that its intensity scales with CTCF initial binding strength.

Revision Figure 2. SNF2H and CTCF binding changes upon BPTF depletion
SNF2H and CTCF localization (assessed by CUT&RUN and ChIP-seq respectively)

shown as single locus heatmap at all bound CTCF sites in WT and upon depletion of BPTF. The black arrow indicates the position of CTCF motif.

7. This reviewer struggles with the interpretation of the footprinting experiments. The window shown is only 400 base pairs which corresponds to maximally 2 nucleosomes, perhaps less depending on the size of the nucleosome-free region. What is being measured here? What is the definition of the “flanking region”? Can this analysis include a broader window?

R9: While the NoMeSeq approach is well established as a measure of DNA accessibility (Pardo et al. 2011, Kelly et al. 2012, Sönmezer et al. 2021) it is indeed not as widely used as e.g., ATACseq. Here nuclei are exposed to a DNA methyltransferase as a DNA modifying enzyme (analogous to a nuclease (DNaseI) or a transposase (ATAC)), whose activity is a function of the local accessibility. The important difference is that the enzymatic activity is read out at the single molecule level using bisulfite sequencing. Due to its resolution and quantifiability, accessible DNA as well as presence of nucleosomes and transcription factors can be detected at base-pair resolution. In our study we focused on CTCF bound sites revealing strong footprint for the TF as observed before (Sönmezer et al. 2021) enabling to quantify binding in presence or absence of BPTF which fully confirms CTCF binding as measured by ChIP as well as the accessibility changes detected by ATAC-seq. We now provide additional loci to make this point clearer (newly added Extended Data Fig. 8c-d). Due to the bisulfite treatment needed for reading out cytosine methylation, the amplification of longer PCR fragments is unfortunately not possible. Several groups are working on alternative footprinting approaches (Abdulhay et al. 2020, Abdulhay et al. 2023), which at this point however do not allow to look at individual regions as they are not PCR compatible. We acknowledge that despite having become more popular, this approach indeed benefits from additional description which we now provide in the main text starting at line 242.

8. A major open question is whether loss of accessibility near CTCF sites in BPTF Δ cells is due to loss of nearby transcription factor binding. A motif search around relevant CTCF sites might be an entry point to address this issue.

R10: We thank the reviewer for this suggestion and now provide additional analysis. Our initial motif analysis in differentially accessible regions divided by clusters mainly highlighted CTCF motif as enriched at sites losing accessibility in BPTF and SNF2H depleted cells (Figure 2d and Extended Data Figure 6c). We now focused only on CTCF bound sites excluding promoters (to correct for proximal vs distal motif occurrence and different GC composition). These sites were binned by accessibility delta in Bptf Δ -WT (Revision Fig. 3a), and motif enrichment was performed using monaLisa (Machlab et al., 2022) searching for known sequences enriched in every bin vs the “zero bin” (bin without changes in ATAC upon BPTF depletion, Revision Fig. 3b). We would expect motifs responsible for ATAC changes to scale in enrichment from less changing bin to the more changing bin (from the white to the deep green bin). We show the result of this analysis by selecting motifs only by significance ($-\log_{10}$ adjusted p-value ≥ 4) (Revision Fig. 3b). Even within CTCF distal sites we retrieve different versions of CTCF motif. In addition, we see an enrichment for NFYA, NFYC and CTCFL (another CTCF variant) motif. To investigate the occurrence of other motifs not related to CTCF we plotted the total motif frequency per bin (Revision Fig. 3c). This reveals that NFYA and NFYC motifs occur at a very low frequency rendering them improbable as a candidate to explain the accessibility changes observed. We additionally checked the co-occurrence of CTCF and NFYA binding by using our CTCF ChIP-seq dataset and our previously generated NFYA dataset (Tiwari et al., Nature Genetics, 2011, GSM632039). In a window of 1kb around CTCF bound distal sites we found 352 NFYA binding events (defined as 2-fold bound in IP/input), corresponding to 0.8% of CTCF distal sites considered. This very low frequency of co-binding argues that any potential contribution of NFYA to the phenotype observed at CTCF sites could only account for a low number of events if any.

Revision Figure 3. Motif enrichment analysis at CTCF regions losing accessibility upon BPTF depletion.

a. MA plots showing average ATAC read counts (x-axis) and ATAC changes upon BPTF depletion (y-axis) for CTCF bound sites. Sites are binned by increasing ATAC loss upon BPTF depletion (from white to dark green). Bin size and related ATAC changes reported in c. **b.** Motif enrichment analysis using monaLisa at CTCF regions binned by ATAC delta as described in a. Motif enrichment has been calculated for each bin vs the zero bin (sites not changing in accessibility). All motifs with $-\log_{10}$ adjusted p value ≥ 4 are reported (standard threshold). **c.** Fraction of CTCF sites with motifs as for b binned by ATAC delta as in a. All motifs retrieved by the analysis in b not related to CTCF are shown.

9. Please clarify and reconcile these two findings: SNF2H binding at distal CTCF sites is BPTF dependent, and BPTF Δ has little impact on CTCF binding while SNF2H Δ does.

R11: The distinct phenotypes resulting from deleting either SNF2H or BPTF is indeed a key and intriguing observation of our study. As already pointed out in the previous version of the manuscript, it is important to also consider that in *Snf2h* Δ also linker length is affected genome-wide while this is not the case for *Bptf* Δ . From these combined observations our working hypothesis is that SNF2H localization in absence of BPTF is very transient at CTCF sites and thus non detected in our experiments but sufficient to

partially maintain CTCF binding. This could be ensured by the combined effect of the remaining ISWI complexes via unspecific nucleosome mobilization (as for the NRL regulation which occurs genome-wide and not at specific regions). While this awaits a molecular understanding (like most, if not all, remodeler contributions to TF binding) it is conceivable that the combined effect of all SNF2H containing complexes would overall impact TF binding in a different way than in the *Bptf* Δ . In the revised manuscript we also excluded the contribution at CTCF sites of the SNF2H homologous protein SNF2L by generating a stable deletion of this protein in mESCs (newly added Extended Data Fig. 1a-f) and we further show that BPTF indeed localizes to CTCF sites (also see R4). Our new experiments in the *Snf2l* Δ line confirms that in mESCs BPTF partners with SNF2H for maintaining CTCF sites open while other BPTF-independent functions of SNF2H allow TF binding. We have tried to make the discussion of such scenarios clearer in the revised version at line 425 of the discussion.

10. Fig.4c: The Hi-C data are described insufficiently. What happened to compartment boundaries, TADs, loops etc in *BPTF* Δ cells? How was a “reduction in long-range interactions” defined?

R12: In response to the Reviewer’s question as well as point 1 and 2 from Reviewer 3 we now provide more in-depth analysis of the Hi-C data. Additionally, to contextualize the extent of the changes observed we added in all analysis comparisons with other mutants previously described to affect insulation (*Snf2h* Δ , Barisic et al., 2019 and CTCF-degron line, Nora et al., 2017). In brief, upon BPTF depletion we observe changes in insulation scores at TAD boundaries which are less pronounced than upon SNF2H depletion or CTCF degradation (newly added Extended data Fig. 10c-d). Importantly these are specific to BPTF depletion as they are absent from comparison of biological replicates of the same sample and compatible with the very recent observations by Radzisheuskaya

and colleagues (2023) showing that BPTF depletion affects chromatin organization in human leukemic lines. We then compared the values of eigenvector 1 (compartment signal) between WT and Bptf Δ datasets and observe this time no changes in compartment organization (newly added Extended data Fig. 10e). We also called TADs using the insulation score method (from the reference Bonev et al., 2017) and quantified TAD strength across conditions using pileups centered around these (newly added Fig. 4c and Extended data Fig. 10f). We observed a reduction in contact enrichment at TAD edges in Bptf Δ vs WT and included comparison with SNF2H mutant and CTCF degron, without detectable difference between replicates. We performed the same analysis for TAD boundaries (newly added Fig. 4d and Extended data Fig. 10g), and consistently observed a small loss of insulation upon BPTF depletion, with again no difference between replicates. Finally, we called loops in the Bonev et al. dataset, and quantified loop strength across conditions using pileups (newly added Fig. 4e and Extended data Fig. 10h). We observe a small but clear loss of contact frequency in loops in BPTF knockout cells relative to WT. These extensive new analyses were done by our collaborators Luca Giorgetti, a leading scientist in the study of genome folding, and his postdoc Ilya Flyamer, who have now become coauthors on our study.

It would be helpful to stratify the CTCF anchors against all fragments, CTCF with other CTCF sites, CTCF with TSS.

R13: As requested by the reviewer, we performed pileup analysis of CTCF sites vs TSS in WT and Bptf Δ (Revision Fig. 4a and 4b respectively) by taking all pairwise interactions between them in the range of 50 kb to 1 Mb distance. We observed a subtle change in contact enrichment within the stripes corresponding to CTCF sites, and the loops between them upon BPTF depletion. We did not observe changes related to TSSs.

Revision Figure 4. Hi-C contacts analysis at CTCF against TSSs in BPTF Δ vs WT.

a. Mean observed/expected interactions at CTCF sites vs TSSs in WT by taking all pairwise interactions between them in the range of 50 kb to 1 Mb distance. **b.** Same as a in BPTF depleted cells.

11. Insulation changes seem modest at best. In fact, in extended data Fig.8d it appears that insulation is stronger for group 1 CTCF sites in BPTF Δ cells. It would be helpful to show some examples on the Hi-C heat maps.

R14: We now provide as requested Hi-C interaction map for a representative region harboring a CTCF site which strongly loses accessibility but maintains binding in absence of BPTF (newly added Extended data Fig. 12c-d). To exclude that the effect observed is due to changes in binding in neighboring CTCF sites we only focused on regions which do not contain any CTCF sites changing binding in a window of 100kb. Additionally, to contextualize the extent of the changes observed upon BPTF depletion we added in all Hi-C analysis comparisons with other mutants previously described to affect insulation (SNF2H deletion and CTCF depletion, also see R12).

12. Also, changes in insulation scores can be driven by changes in intra- or inter-TAD

interactions. For example, a reduction in -gene contacts in gene rich TADs can manifest in reduced insulation. Fewer intra-TAD interactions in BPTF Δ cells are not a reflection of CTCF insulation function.

R15: We thank the reviewer for raising this possibility. We do not observe an average loss of contacts within TADs, only at their edges, where contacts are largely driven by CTCF. This is visible in average TAD pileups in the newly added Fig 4c. We also quantified this by looking at the average contact frequency within a progressively larger window centered on the center of the TAD (Revision Fig. 5a). We actually observe a slight increase of interaction frequency in the knockout conditions relative to WT while the window size is below ~ 0.75 of the TAD size, which suggests that if anything, the intra-TAD interactions might be increased when interactions at CTCF sites are reduced (Revision Fig. 5b).

Revision Figure 5. Hi-C contacts frequency changes at TADs upon BPTF or SNF2H depletion

a. Schematic of the analysis performed in b: average observed/expected contact probability is calculated in windows of increasing size centered on the center of the TAD. **b.** Top: average contact frequency within a progressively larger window centered on the center of the TAD upon BPTF or SNF2H depletion in contrast to

their respective controls. Bottom: same as top, but for comparison of WT replicates in BPTF and SNF2H datasets.

13. Related to point 12: The CTCF sites in question are at distal enhancers. Are these

enhancers at TAD boundaries? Are their target genes weakened in BPTF Δ cells?

R16: As shown in Figure 4b of the previous submission, only a minority of affected CTCF sites do reside in enhancers as defined by ChromHMM (clusters 2-5 Fig. 4b), while being more enriched in cluster 1 which is mildly affected by BPTF depletion. Additionally, as illustrated above, we further validate this notion by intersecting our CTCF site annotation with the ENCODE cCRE annotation. This analysis (also see R2) highlighted that persistent sites (cluster 1) tend to lie in enhancer-like elements thus we did not further investigate the downstream effects at these regions.

14. Reduction in insulation is expected to result in inter-TAD contacts and perhaps illegitimate enhancer-promoter contacts. Can gene expression changes in BPTF Δ cells be accounted for in this manner?

R17: Assigning transcriptional dependencies over long distances has been proven difficult not only at the level of enhancers and dependent genes but even more as a function of insulation activity. We have nevertheless tried to address this question by looking at gene expression changes as a function of the closest bound CTCF site. As a reference, we included in the analysis the RNA-seq data from SNF2H and CTCF depleted cells (Barisic et al., 2019, Nora et al., 2017). The results of this analysis are reported in Revision Fig. 6. In agreement with previous reports (Nora et al., 2017), this reveals that upon degradation of CTCF, genes where the strongest bound CTCF site is very close to the TSS are more likely to be downregulated at the earliest time point (Revision Fig. 6a, first panel from the left). Additionally, when looking at genes where the closest strongly bound

CTCF site is more distal, we observe, on average, a progressive upregulation of genes over time (Revision Fig. 6a). This appears compatible with a model where loss of CTCFs-mediated insulation leads to additional enhancer-promoter contacts, leading to further gene activation. When performing the same analysis in the background of the depletions of SNF2H or BPTF, we observe in both cases, if anything, a small effect proximally (Revision Fig. 6b). Additionally, we do not observe a distal effect in *Bptf* Δ and only a minor trend in case of *Snf2h* Δ . This appears compatible with the increase in insulation changes from the BPTF to the SNF2H mutant to the CTCF degron line. Importantly, it has to be stressed that in the case of BPTF and SNF2H mutants, we are not looking at gene expression changes upon acute depletion but in stable knock-out cells, which are more likely confounded by secondary effects.

In summary, there seems to be no detectable genome-wide relationship between gene expression and the position of the closest bound CTCF site in the *BPTF* Δ line except for a very small downregulation of genes harboring CTCF sites very close to TSS. Thus, there seems to be no strong effect on gene expression mediated by changes in insulation, at least in stable knock-out cells.

Revision Figure 6. CTCF proximity to TSSs does not explain genes expression changes in the remodeler mutants

a. Bar plot representing RNA changes upon CTCF degradation (y-axis) for 1, 2 or 4 days (panels from left to right) binned by distance of the closest strongly bound CTCF motif from the TSS (x-axis). **b.** Same analysis as in a for Bptf Δ (left) and Snf2h Δ (right) gene expression changes.

15. Extended data Fig.8c,d: Why are the pile ups asymmetrical? What are the stripes at the bottom? Why are there no stripes at the top?

R18: The CTCF motif is asymmetrical and it only pauses cohesin loop extrusion in one orientation. We always orient the CTCF sites in one direction in the Hi-C analyses hence, it preferentially interacts in one direction. The direction of the CTCF motif is now indicated more clearly by small arrowheads (pointing in the cohesin-stalling direction) across all figures to clarify this point.

16. Line 319: “BPTF-mediated accessibility around CTCF sites is required for efficient insulation”. This sentence implies causation but is in fact a correlation, and accessibility may not be “required.” The effect on chromatin contacts or insulation may be mediated by TFs that contribute to contacts directly or indirectly via cohesin loading.

R19: The reviewer is correct in that we did not specifically search for additional motifs at those CTCF sites with reduced cohesin. To address this possibility, we now focused on CTCF distal sites binned by RAD21 ChIP-seq changes in BPTF KO vs WT (Revision Fig. 7) and performed motif enrichment at these bins using monaLisa (also see R10). We would expect motifs responsible for RAD21 changes to scale in enrichment from less changing bin to the more changing bin. This analysis did not retrieve any enriched motif. We thus conclude that there is no evidence at these sites for other TFs than CTCF in mediating changes of cohesin localization.

Revision Figure 7. Motif enrichment analysis at CTCF sites binned by Rad21 changes upon BPTF depletion

MA plots showing average RAD21 read counts (x-axis) and its binding changes upon BPTF depletion (y-axis) for CTCF bound sites. Sites are binned by increasing RAD21 loss upon BPTF depletion (from white to dark green). Motif enrichment at these bins using monaLisa vs the non-changing bin did not retrieve any sequence with a $-\log_{10}$ adjusted p-value of at least 4.

Minor:

1. Extended Data Fig. 4e could be made a main figure because it displays data central to this report.

R20: Due to the limited space available in the main figures we would prefer to keep this figure in the Extended Data unless the reviewer deems it absolutely necessary.

2. Extended Data Fig. 6a: “log₂ fold-change IP/input”. What “change”?

R21: This denotes IP over input condition. This was made clearer in the figure legend (now Extended data Fig.8a).

3. Why are cohesin and CTCF ChIP-seq peaks doublets? This is unusual.

R22: This is due to the fact that in these ChIP metaprofiles the reads are shown without shifting by fragment size, in agreement with a similar analysis in our previous publication (Iurlaro et al, 2021). We thank the reviewer for highlighting this point, which we now clarify in the appropriate Methods section.

4. Line 243: replace “loose” with “lose”

R23: We thank the reviewer for the comment and corrected the wording.

5. Break up extended data Fig.4b into individual panels.

R24: We now provide previous Extended Data Fig 4b split by cell line as requested by the reviewer in the new Extended Data Fig. 5f-k.

6. Lines 293 and 368: “Affinity” to the motif has not been measured in this study and other mechanisms may account for the persistent CTCF binding in BPTF Δ cells. “CTCF affinity for its motif seems to be crucial in retaining binding in absence of NURF” is an overstatement.

R25: The presence of M2 has been shown previously to coincide with evolutionary conserved binding events which are furthermore less sensitive to CTCF depletion (Schmidt et al., 2012) in line with an expected higher affinity to a larger motif. However, the reviewer is correct in that we did not provide affinity measurement in the manuscript. Thus, we changed accordingly the discussion (line 414).

7. I have difficulty in interpreting this sentence: “However, we are not aware of an example where absence of a cofactor causes loss of chromatin opening yet, largely persistent TF binding as we show here for CTCF in absence of BPTF”. There are many cases of TFs binding to regions of closed chromatin with no ATAC signal or any other indications of chromatin remodelling co-factors.

R26: The reviewer is correct that there are indeed cases of TF binding that do not cause ATAC signal including p53, on which our group recently reported on (Isbel et al., 2023). In our statement we refer specifically to the phenotype obtained upon remodeler depletion, where thus far reduced accessibility coincided with reduced TF binding (King & Klose, 2017; Barisic et al., 2019; Minderjahn et al., 2020; Iurlaro et al., 2021). If the reviewer is aware of such examples, we are happy to include them and revise our

statement accordingly.

Reviewer #2:

Remarks to the Author:

In this manuscript, Iurlaro/Masoni et al investigate the function of the ISWI subunits RSF1, ACF1, WSTF, TIP5, CECR2 and BPTF, by making individual loss-of-function mutations in mESCs. Although most deletions have only moderate effects on nucleosomal structure and gene expression, the authors identify strong changes in chromatin accessibility around CTCF sites after mutating the BPTF subunit (part of the NURF complex). However, surprisingly, the authors find that these changes in nucleosomal structure only modestly impact binding of CTCF. Despite these modest changes in CTCF binding, the authors show that loss of BPTF leads to changes in occupancy of Cohesin and associated changes in 3D genome organization. The authors therefore speculate that local accessibility (and not CTCF binding per se) is critical for Cohesin binding and insulation of chromatin domains.

This manuscript presents interesting experiments that give new insight into the function of ISWI remodelers. The data are of high quality, include all relevant controls, are well presented, and clearly described. The finding that BPTF regulates accessibility around CTCF-binding sites without having a major impact on CTCF-binding (in contrast to SNF2H) is convincing and interesting. It also seems clear that loss of BPTF leads to some changes in genome organization. However, I am not convinced that the conclusion that CTCF binding is separated from insulator function is supported by the data.

R27: Through additional analysis, revisions of figures and text we made clearer that at a

class of CTCF sites loss of BPTF reduces accessibility but not CTCF binding and that at these sites insulator function is also impaired as detected by HiC as well as cohesin presence (see individual points below). While it was already evident in the initial submission, particularly in the figures, that this relates only to a class of sites, but not all, we now refined our wording in the abstract, results and discussion sections to avoid the impression of a generalization.

Major comments

1. In Figure 4c, the authors show changes in nuclear organization in BPTF-mutant cells as compared to WT cells. Cluster 1 (n=9707) shows that when there is no change in CTCF binding and a very modest change in accessibility, there are no clear changes in insulation. Cluster 2 shows that when there is no change in CTCF binding and a moderate change in accessibility (n=10377), there are also no clear changes in insulation

Cluster 3-5 (n=12111+8752+6683=27546) show that when there is a reduction in both CTCF-binding and accessibility, there is a clear reduction in insulation

In Figure 5b, the authors focus on chromatin insulation changes around CTCF-binding sites at which CTCF occupancy does not change strongly in absence of BPTF. Cluster 1 and 2 show that when there are no/very modest changes in accessibility (n=1358+6941=8299), there is no clear effect on insulation. Cluster 3 shows that a moderate reduction in accessibility (n=6253) is associated with an extremely subtle effect on insulation. Cluster 4 shows that a strong reduction in accessibility (n=1313) is associated with a stronger reduction in insulation.

Although Cluster 4 in Figure 5b clearly shows a reduction in insulation, I think it is important to stress that this is only observed at 1313 sites. Moreover, there is a small

reduction in CTCF binding in these sites as well.

Furthermore, Cluster 2 in Figure 4c shows 10377 sites at which there is an almost equally strong reduction in accessibility as in Cluster 3 in Figure 5b, but in Figure 4c/Cluster 2 there are no clear reductions in insulation (i.e. the lower-left and upper-right corners seem to contain roughly as many pink regions as blue regions).

To me it seems that the most numerous and strongest changes in insulation that result from BPTF loss are associated with a reduction in both CTCF binding and accessibility (Cluster 3-5 in Figure 4c).

The conclusion that "These results separate local CTCF binding from Insulator function in nuclear organization" therefore seems overstated to me and not fully supported by the current data.

R28: The reviewer is correct in that those CTCF sites that lose accessibility and binding show the strongest effect in terms of insulation as expected from CTCF function and in line with previous observations by us and others (cluster 3,4,5 newly added Fig. 4f, Extended Data Fig. 11a). To better highlight this point we changed the main text at line 347.

However, in addition to this pattern, it is evident that clusters presenting the same level of CTCF loss but different levels of ATAC loss also show different changes in insulation at a high number of sites. Indeed cluster 4 vs 3 display stronger ATAC loss and stronger insulation changes despite similar variation in binding (newly added Fig. 4f and Extended Data Fig. 11a). This argues that accessibility plays an additional role in CTCF insulation function. To separate the accessibility and binding components, grouping of the sites was necessary to focus on the regions with only accessibility reduction but persistent binding as specified above. Of 15865 sites that show no change in binding (Fig. 5a) 47% display a strong accessibility loss coupled with a reduction in insulation (Fig. 5a-b group3,4 and

Extended Data Fig 12b). Extended Data Fig 12a reports the absolute data for all loci in these groups highlighting the persistent CTCF binding at these sites despite strong changes in accessibility and changes in Hi-C. This phenotype is unique for the deletion of BPTF and is furthermore independently confirmed by reduced presence of RAD21 and WAPL (Fig. 5c, also see R33 below) and to our knowledge has not been reported before (also see general comments from Reviewer 3). While the fact that this could be observed at a specific class of sites was evident in figures and description in the previous version, we now made this point clearer in the text and abstract to avoid the false impression that this is happening at all CTCF sites (in the main text at line 361,380, in the abstract at line 22 and discussion at line 393, 442).

2. Related to the comment above, it would be interesting to directly compare the Hi-C data in the BPTF mutant to the Hi-C data in the SNF2H mutant that was published by the authors in 2019. This may help in determining whether CTCF binding or accessibility is driving insulation. Since the reduction in CTCF binding is larger in the SNF2H mutant compared to the BPTF mutant but the changes in accessibility are similar, a model in which CTCF binding determines insulation would predict that the changes in insulation are larger in the SNF2H mutant compared to the BPTF mutant. Conversely, if insulation is dependent on accessibility and not on CTCF binding, the changes in insulation in the two datasets should be similar.

R29: The main conclusion of our study is that BPTF dependent accessibility around bound CTCF contributes at a class of sites to CTCF function. We never claimed that CTCF binding itself is not highly critical. However, we now better highlight this point as described above in R28. We thank the reviewer for the suggestion to compare the Hi-C changes observed upon BPTF depletion to the ones observed upon SNF2H loss. We expanded the analysis of Hi-C datasets including comparison with other mutants leading to loss of insulation at CTCF sites, notably *Snf2h* Δ (Barisic et al., 2019) and CTCF degra-

line (Nora et al., 2017), in newly added Fig 4c-f, Extended Data Fig. 10c-h, Extended Data Fig. 11, Fig. 5b, Extended Data Fig. 12c-d. This illustrates, as expected, that Snf2h deletion does not only result in stronger reduction of CTCF binding but also insulation compared to BPTF. As clarified in the point R28 above and in the text the strongest changes in insulation also in Bptf Δ are at sites losing both binding and accessibility, in line with the phenotype observed in absence of SNF2H. This does not disprove an additional role of accessibility in CTCF structural function that we could isolate only in the Bptf Δ line where changes in binding are more restricted. These changes are less pronounced than the one observed in the SNF2H mutant, which we clarified in the newly added figures and in the main text at line 332-341.

3. A prominent feature of CTCF function in genome organization is the formation of "CTCF loop" between the CTCF-binding sites at the boundaries of domains. It would be very interesting to analyze if there are any changes in the formation of these loops in absence of BPTF in the Hi-C data.

R30: We have now added a more in-depth analysis of various features in the Hi-C data, including loops. To do so, we used loops called using the Bonev et al. 2017 mESC dataset and quantified loop strength changes using pileups upon depletion of BPTF, and for comparison SNF2H deletion and CTCF depletion (newly added Fig. 4e, Extended Data Fig. 10h). We observe a small but clear loss of contact frequency in loops in BPTF knockout cells relative to WT.

4. The authors show that mutation of BPTF leads to 693 up- and 1143 down-regulated genes. Are these changes related to the impaired CTCF function in absence of BPTF, i.e. are they associated with changes in genome organization, such as enhancer-promoter interactions?

R31: Assigning transcriptional dependencies over long distances has been proven difficult not only at the level of enhancers and dependent genes but even more as a function of insulation activity. Thus, to investigate this point we generally tried to investigate the dependency of gene expression changes relative to the closest strongly bound CTCF site (Also see R17). As a reference, we included in the analysis the RNA-seq time course data of auxin-induced degradation of CTCF (Nora et al. 2017) as well as the RNA-seq data from SNF2H mutant cells (Barisic et al., 2019). In brief, in agreement with previous reports (Nora et al., 2017), we find that upon degradation of CTCF, genes where the strongest bound CTCF site is proximal to TSS are more likely to be downregulated but only at the earliest time point (1 day). Additionally, when looking at genes where the closest strongly bound CTCF site is more distal, we observe a progressive average upregulation of genes with time (day 1 to day 4). This appears compatible with a model where loss of CTCFs-mediated insulation leads to additional enhancer-promoter contacts. There seems to be no detectable relationship between gene expression and the position of the closest bound CTCF site in the BPTF Δ line except for a very small downregulation of genes harboring CTCF sites very close to TSS. Thus, there seems to be no strong effect on gene expression mediated by changes in insulation.

Minor comments

1. The right panel in Figure 2b is a bit hard to read, but seems to show that nucleosome positioning at TSSs is more distinct in the *Cecr2* mutant. Can the authors comment on this observation?

R32: Given that the overall shape is identical and since the baseline in the valley in the *Cecr2* mutant is also a bit higher than the control we do not consider this to be a distinctive feature. This is also in line with the fact that the *Cecr2* mutant displays very modest changes in accessibility and gene expression (Fig. 1c-d, Fig.2d).

2. In line with major comment #1, it would be interesting to see the RAD21 and WAPL occupancy aligned with the matrices in Figure 4c as well.

R33: As requested by the reviewer we now provide in Revision Fig. 8a-b Rad21 and Wapl binding profiles for each cluster (as for Figure 4a) in WT and Bptf Δ conditions. This illustrates for both Rad21 and Wapl increasing loss of signal following cluster 1 to 5 in agreement with loss of chromatin contacts as shown in Fig. 4f. Importantly in the context of our study this also shows that both Rad21 and Wapl signal decrease between cluster 1 to 2. Unique to the BPTF depletion these clusters only differ in loss of accessibility while CTCF binding is unchanged in both groups. This again highlights that at a class of sites, loss of accessibility alone coincides with reduced cohesin and long range contacts despite persistent CTCF binding. We thank the reviewer for suggesting this analysis which we included only in the rebuttal due to limited space in the manuscript.

a.

b.

Revision Figure 8. RAD21 and WAPL localization changes upon BPTF depletion divided by clusters

a-b. Average ChIP-seq signal for RAD21 (a) and WAPL (b), at CTCF sites divided by the same clusters as in Fig. 4a and 4f.

Reviewer #3:

Remarks to the Author:

My expertise lies in chromosome folding, CTCF biology and Hi-C analyses. I therefore was able to review these aspects of the paper with greater attention than the chromatin accessibility data.

This manuscript contains many exciting and novel observations clarifying the role of chromatin remodelers creating chromatin accessibility after transcription factor binding. I found the decoupling of CTCF binding and accessibility in the BPTF mutant especially exciting and compelling – even if it only occurs at a very small number of CTCF binding sites. These results illustrate how transcription factor binding is not always sufficient to create accessibility, which sometimes requires recruitment of dedicated remodelers. These observations have far-reaching implications beyond CTCF biology. I only have minor suggestions, mainly to clarify effect size and some of the Hi-C analyses.

We thank the reviewer for these positive comments and for pointing out the relevance of our key observation for the field, even though limited to a class of CTCF sites.

Main suggestions

1. It is very difficult to assess the effect size of genome misfolding from the data presented. Below are some suggestions to help clarify this point.

a) Rewording. The way the abstract is phrased make it sound like one should expect

insulation to be impaired at most if not all CTCF sites in BPTF mutants. Yet, as authors show in figures 4 and 5, this is not the case at all. In fact, only 8% (1313) of CTCF sites with unchanged binding (group 4 of figure 4) seem to display insulation alterations. This represents less than 2.8% of all CTCF binding sites.

This number is comparable to the fraction of CTCF sites where binding is most decreased (14% cluster 5 of figure 4), yet here authors concluded that CTCF binding is only modestly decreased (abstract). As the analyses presented indicate, genome misfolds but only a handful of CTCF sites. In a nutshell some rewording of the abstract and the manuscript would be helpful to help the reader better appreciate the scale and (limited) pervasiveness of the folding defects.

R34: In our study we focused on our novel observation that upon BPTF depletion a class of CTCF sites show drastic changes in accessibility despite CTCF binding being weakly or not at all impaired. However, the reviewer is correct in that the sites where the accessibility effect can be clearly separated from loss of binding represents a subset of all bound CTCF regions. While this was already reported in figures and description in the previous version, we further made this clearer it in the abstract and the main text to avoid the unintended impression that this would be the case for all CTCF sites (see R28 and changes in the main text at line 361,380, in the abstract at line 22 and discussion at line 393, 442).

b) It is quite difficult to appreciate the effect size of genome misfolding from the figures. Brief inspection of the Hi-C heatmaps generated from the .cool file provided (without separating the clusters) does not display pervasive defects – contrasting with the SNF2H or CTCF-AID mutants previously reported. Additional Hi-C metrics could help provide comparative numbers. For example, reporting level of insulation in figure 4c and

5b etc. (e.g. signal enrichment in the top right square) in WT vs KO would help better understand the magnitude of the defects.

Showing heatmaps of the actual signal, as opposed to only showing the log₂FC of obs/exp, would also help the reader get a sense of how much change compared to baseline is occurring.

Alternatively, aggregate peak analyses centered at the CTCF site clusters/groups of figure 4 and 5 could help understand better the fold change in Hi-C signal between WT and KO.

R35: To address the reviewer's point, we now added a metric of insulation strength to figures reporting local pileups centered on CTCF sites or insulating boundaries for each genotype and controls (newly added Extended Data Fig. 10 f-h, Extended Data Fig. 11a, Extended Data Fig. 12b). We calculated the ratio of the average contact frequency between the top-left and bottom-right quarters of the pileup, and the top-right and bottom-left quarters, while ignoring the central 3 bins to reduce the contribution of stripes to the measurement. We hope that effect sizes have become clearer, especially as we now provide expanded analysis of Hi-C datasets including comparison with other mutants leading to more prominent loss of insulation at CTCF sites, notably *Snf2h*Δ (Barisic et al., 2019) and CTCF degron (Nora et al., 2017) in the newly added Fig 4c-f, Extended Data Fig. 10f-h, Extended Data Fig. 11a, Fig. 5b, Extended Data Fig. 12b-c. Additionally, we now show pileups with the absolute signal (observed/expected) in each condition for all the reported analyses (see Extended Data Fig. 10f-h, Extended Data Fig. 11a, Extended Data Fig. 12b). Related to the last comment we have performed the aggregate peak analyses centered at CTCF sites divided by clusters and groups and display the pileups of average contact frequencies in the newly added Fig 4f, Extended Data Fig. 11a, Fig. 5b, Extended Data Fig. 12b.

2. To further help readers appreciate the extent of the Hi-C changes – especially in relation to SNFH KO and CTCF depletion:

A) it would be especially insightful to report the change in insulation as scatter plots, as

authors did in their previous Nature article on SNF2H (fig4a).

R36: We thank the reviewer for this suggestion. We have added scatterplots comparing insulation scores of boundaries (from Bonev et al., 2017 data) for BPTF, SNF2H and CTCF loss (see newly added Extended Fig.10 d). Insulation score changes in such analysis are virtually undetectable in BPTF KO conditions, especially when compared to SNF2H and CTCF depletion. This could however be expected based on the smaller effect size in pileup analysis, which is in general more sensitive to local changes in insulation than insulation analysis. Importantly, although modest, the insulation changes detected appear specific to BPTF deletion as they cannot be detected in analysis of biological replicates e.g. of WT samples (see Extended Fig.10 c). Importantly as well, small changes in insulation scores in BPTF deletion conditions tend to occur at boundaries that also change insulation upon SNF2H deletion (see R46 below).

B) it would also be most insightful to show at least one Hi-C heatmap with the signal for each genotype (not just the differential heatmaps)

R37: In response to the reviewer's comment, we provided Hi-C heatmaps for a representative region, harboring a CTCF site which strongly loses accessibility but maintains binding in absence of BPTF. To exclude that the effect observed is due to changes in binding in neighboring CTCF sites we only focused on regions which do not contain any CTCF sites changing binding in a window of 100kb. (see newly added Extended Data Fig. 12c and d).

C) adding a supplementary plot of aggregate peak analysis centered at Hi-C loops found in WT (possibly taking a ultra-deep reference loop list – e.g. Bonev et al. 2017) would be helpful. This would help readers understand the extent of looping loss upon BPTF deletion. Again, it could help to show the signal in each genotype in addition to a heatmap reporting the difference (subtraction of signal, not a Z-score or other

observed/expected metrics).

R38: We called loops using the Bonev et al. (2017) dataset, and quantified loop strength across conditions using pileups (see newly added Fig. 4e and Extended Data Fig. 10h). We observe a small loss of contact frequency in loops in BPTF knockout cells relative to WT, which is absent from replicate analysis.

D) Adding direct comparison to SNF2H would be insightful. If authors also performed Hi-C upon CTCF depletion (e.g. degron cells available) it would enable readers to understand the quantitative contribution of BPTF to the insulation functions of CTCF. This is not strictly necessary if authors have not already performed Hi-C after CTCF degradation.

R39: We now include our previously published SNF2H data (Barisic et al, 2019), and also the published CTCF depletion dataset from Nora et al, 2017, in all Hi-C analyses for reference (see newly added Fig 4c-f, Extended Data Fig. 10d and f-h, Extended Data Fig. 11a-b, Fig. 5b, Extended Data Fig. 12b-c).

3. There seem to be a problem with the Hi-C data deposited on GEO, specifically with GSM7461658_HiC_BPTFko_rep1_matrix.cool.gz

This sample appears to have been normalized in a different way than the other three samples. See attached figures.

It looks like the problem comes from the balancing of the cooler file. When I merged the two replicates and re-balanced the merged file it solved the issue. Please double check the processing of the file for BPTFko_rep1 and update the analyses that may have used it.

R40: As the analysis of all Hi-C samples was repeated during revisions, in parallel with the publicly available datasets now used in the manuscript, all related figures and deposited data were modified accordingly. This should have corrected the reported issue and we thank the reviewer for highlighting it.

Please also make sure to provide on GEO the actual .cool/.mcool files that were used for the analyses presented in the paper (I assume a file containing pooled replicates was used?) – same goes for the CHIP/MNase/RNA-seq processed files.

R41: We updated the GEO submission with the processed files used for the Hi-C analysis (as outlined above) and also included processed data files derived from pooled replicates. Regarding the other NGS experiments, we specify in the methods that all plots report the mean signal from two or more independent biological replicate thus we provide the processed files for each replicate separately in the GEO submission.

Minor suggestions

1. I found the main text description of figure 4c somewhat misleading. Authors state that “CTCF sites with increased loss of accessibility show more significant reduction in long- range interactions” [line 316]. Yet only clusters 3, 4 and 5 display loss of insulation. These are the clusters where CTCF binding is reduced in Bptf mutants. Cluster 2, where CTCF binding is preserved but accessibility is reduced, shows minimal Hi-C defects in panel c. This would argue that changes in Hi-C insulation are attributable to the reduction of CTCF occupancy, rather than some downstream process such as cohesin retention. Authors need to clarify this point or reword this section – or simply wait for the description of figure 5 to make this statement.

R42: We have modified the description of Figures 4 and 5 accordingly and thank the reviewer for the suggestion (also see R28 and main text changes at line 347).

2. Any explanation why SNF2H KO caused loss of CTCF binding while none of the subcomplexes? Authors state that “Individual ISWI subunits do not account for the nucleosomal phenotype caused by loss of catalytic activity” – Could authors speculate then about what does? It is very intriguing that none of the subunit deletion recapitulate the NRL increase observed upon SNF2H deletion. It would be interesting if authors could speculate about possible mechanisms.

R43: The distinct response in binding observed when deleting either SNF2H or BPTF is indeed an intriguing observation and a key finding of our study combined with the result that none of the other ISWI complex seem to play a role at CTCF regions based on single KO results. In the revised manuscript we also excluded that the SNF2H homologous protein SNF2L could play a role at these same regions by generating a stable KO in mESCs (see R44 below and newly added Extended Data Fig.1). While these findings remain to be fully understood in molecular detail, which is the case for most remodeler functions and contributions to TF binding, it is conceivable that the combined effect of all Snf2h containing complexes would overall impact TF binding in a different way than in the BPTF KO.

Based on our findings we speculate that ISWI complexes could control CTCF sites by two different mechanisms: unspecific nucleosomal mobility mediated by different SNF2H-containing complexes could enable CTCF binding while specific recruitment of NURF is required for chromatin opening. We have tried to make the discussion of such scenarios clearer in the revised version (see line 425-433 of the discussion).

Regarding the second part of the reviewer comment it is correct in that no single ISWI subcomplex accounts for the changes in nucleosome repeat length observed when depleting the enzymatic activity SNF2H. As highlighted in the discussion this finding would argue for a redundancy by two or more subcomplexes in regulating nucleosome distance. Additionally, it could be hypothesized that the nucleosome spacing activity of

ISWI is intrinsic of the Snf2h enzymatic activity and does not depend on the coupled accessory subunits. This is compatible with the fact that all three ACF, RSF and WICH mammalian complexes show nucleosome spacing activity in vitro (see line 399 of the discussion).

3. Have authors performed Hi-C and MNase/ATAC-seq SNF2L KO? Does it recapitulate BPTF KO? A simple statement saying whether authors have tried to generate these cells would be helpful (especially if authors already know whether the SNF2L KO cells are viable or not)

R44: Motivated by the reviewer comment we now provide a newly generated SNF2L deletion in mES cells (newly added Extended Data Fig.1a-f). Its characterization reveals very limited to no changes at the transcriptome level and only subtle changes in ATAC-seq and importantly, outside of CTCF sites. From these findings we conclude that SNF2L has no particular function in ES cells compared to SNF2H. This is in line with the fact that SNF2H is expressed to a higher level in this cell type and with the observation by Radzisheuskaya and colleagues (2023) that in human leukemic lines where SNF2L is also weakly expressed, SNF2H interacts with BPTF in the NURF complex. In line with this apparent lack of phenotype, we did not investigate Snf2l Δ lines further at the level of genome folding.

4. How does the Hi-C compare between SNF2H (previous Nature article from the authors), BPTF (present data) and, if possible/relevant SNF2L (if these cells are viable)?

R45: SNF2L depleted lines did not show significant changes in gene expression or accessibility at CTCF sites in contrast to the phenotype observed when depleting BPTF or SNF2H (see R44 above). This is in line with the finding that in our system SNF2H interacts with BPTF. For these reasons we did not further evaluate Hi-C changes in the Snf2l Δ line.

As discussed above we now included our previously published SNF2H data (Barisic et

al, 2019) in all Hi-C analyses for reference (see newly added Fig 4c-f, Extended Data Fig. 10d and f-h, Extended Data Fig. 11a-b, Fig. 5b, Extended Data Fig. 12b-c). Comparison between *Snf2h*Δ and *Bptf*Δ Hi-C changes revealed stronger effect in the SNF2H depletion as expected by much stronger loss of CTCF binding. This has been clarified in the main text at line 332-341 in the main text.

5. Given that authors report that BPTF can associate with SNF2H, to what extent the loss of insulation initially reported in the SNF2H KO can be attributed to the BPTF-dependent function of CTCF in cohesin blocking, as opposed to loss of CTCF binding?

R46: The effect of BPTF loss on insulation is weaker than that of SNF2H (see second point of R45) so we expect that the contribution of BPTF-dependent cohesin-blocking function is lower. However, plotting the log fold-change of insulation scores at TAD boundaries in BPTF-KO/WT vs SNF2H-KO/WT conditions (Revision Fig. 9) reveals a modest, yet highly significant correlation since boundaries that lose insulation (positive logFC) upon knockout of BPTF are also more likely to lose it in SNF2H KO, and vice versa (Fisher's exact p-value 9×10^{-17}).

Revision Figure 9. Insulation changes at TAD boundaries upon BPTF vs SNF2H depletion

Log2 fold-change of insulation score at TAD boundaries upon BPTF depletion (x axis) compared to SNF2H depletion (y axis).

6. Authors conclude that “the deletion lines are subcomplex-specific (Extended Data Fig.1). Yet on the Western blot figure 1b it looks like that some deletions affect the other subunits. For example, CECR2 displays a strong extra band in the ACF1 deletant, and the stoichiometry of the ACF1 subunits appear to change in the TIP5 and BPTF deletant. Authors may wish to comment on this, either to say that deletion of one subunit consistently affects the other ones (if this effect is reproducible), or to say that these variations are spurious (if effect is not reproducible).

R47: We thank the reviewer for highlighting this point. The mentioned change in band stoichiometry of Cecr2 and Acf1 had been observed in other repetition of the western-blot and Figure1b is representative of this observation. Thus, this was specified in the figure legend of Fig1b.

7. Authors describe the reduction in MNase accessibility in BPTF at distal DHS to the presence of CTCF sites. However they do not report how MNase patterns change at distal DHS that do not have CTCF. I suggest replacing Ext fig 4c with the non-CTCF distal DHSs. I found this finding especially important for the rest of the paper, and would suggest considering to move Ext fig 4b and c (once replaced with the non-CTCF distal DHSs) to the main figures. If space permits.

R48: The reviewer is asking whether distal DHSs non overlapping CTCF are affected upon loss of BPTF. To answer this question, we now compare MNase signal at DHSs overlapping CTCF sites vs DHSs non-overlapping CTCF sites (newly added Extended Data Fig. 5c-d). This illustrates that non-CTCF bound sites tend to show weak nucleosome phasing on average since most are bound by multiple TFs with different spacing. Their average spacing is unchanged in the BPTF KO. If we focus on distal DHSs

occupied by the TF REST (Extended Data Fig. 5e), which tends to bind alone leading to strong phasing, comparable to CTCF, we observe no changes in nucleosome organization in the BPTFKO. This, combined with the fact we do not identify additional motifs with enrichment comparable to CTCF when looking at differentially accessible regions (Extended Data Fig. 6c, also see R10) lets us conclude that CTCF accounts for the observed changes at distal DHSs. We thank the reviewer for this suggestion.

8. Authors did not use ChIP-seq spike-ins. It would be helpful if authors could briefly explain to what extent this might affect their conclusions regarding the absolute changes in binding measured in the various mutants. It would also be helpful to explain if the CUT&RUN suffers from the same limitations. Redoing all the experiments with spike-ins would not be necessary at this stage.

R49: Since chromatin IP is an enrichment-based assay, it is by nature not quantitative. Instead of using external references in this assay, we opted for an orthologous approach that is quantitative. We employed single molecule footprinting as measure of occupancy that directly relates to ChIP signal (Sönmezer et al. 2021). Importantly this confirms independently not only the observed CTCF binding but also the reduced accessibility adjacent to CTCF. To make this important control more visible we now provide new footprinting data for additional individual loci (newly added Extended Data 8c-d). This confirms persistent binding and reduced accessibility as measured by ChIP-seq and ATAC-seq. We believe that this important control and orthologous approach validates our conclusion of continued CTCF binding at many sites in the BPTF mutant. In the case of the remodeler location analysis, we provide in addition to the IgG negative control also lines stably depleted from the target protein to contrast the enrichment in WT and BPTF KO condition. Additionally, we validate that the changes observed at CTCF sites are not of technical nature as we show that Snf2h localization at the remaining DHS is unchanged upon BPTF depletion (Extended Data Fig. 7f-g).

9. I found that figure 2 does not clearly convey what happens to CTCF binding. Authors show convincing cherry-picked loci in 2b, but an additional panel reporting the number of peaks in each category (e.g. unaffected, reduced, completely lost, enhanced...) would help readers understand better the general trends. For example authors could add a

series of panels splitting the line graphs currently present in figure 3a into several groups (e.g. unaffected, reduced, completely lost, enhanced...) and label the number or fraction of sites in each categories. Another option would be moving Extended data fig 8a in the main figure 2, and indicating the number of CTCF peaks in each cluster.

R50: We assume the reviewer refers to Figure 3 and not 2. We now added in Extended Data Fig. 8b metaprofiles illustrating CTCF binding now divided based on their response upon BPTF depletion as suggested by the reviewer. We hope that this additional figure illustrates more clearly the CTCF binding changes observed upon BPTF depletion.

10. The manuscript would benefit from additional comparisons of the transcriptome datasets. For example a Principal component plot with the genotypes of figure 1d might be helpful?

R51: We now added a PCA plot in Extended Data Fig. 3c. This illustrates that the first principal component (explaining 38% of the total variance) distinguishes *Snf2h* Δ and *Bptf* Δ lines from all other genotypes. The second principal component (explaining 29% of the remaining variance) separates both genotypes. Importantly the PCA illustrates the similarity between *Acf1*, *Rsf1*, *Tip5* and *Cecr2* deletion lines, while *Wstf* Δ appears different in line with presenting the highest number of DEGs after *Snf2h* Δ and *Bptf* Δ .

11. In addition to the GO term analysis enrichment it may be interesting to report on the

genomic context of the differentially expressed genes. For example, how many are Polycomb targets? How many had CTCF binding at the promoter? Etc...

R52: We now added the ChromHMM annotation of the DEGs promoter regions for each deletion line (see Extended Data Fig. 3d). This analysis shows that in each line promoters of affected genes (both up and down regulated) are enriched for the bivalent chromatin state. In Revision Fig. 10 we also show, as requested, the fraction of differentially expressed genes promoters bound by CTCF, in comparison to all annotated promoters.

Revision Figure 10. Fraction of changing promoters bound by CTCF upon ISWI deletions

Percentage of annotated promoters and differentially expressed genes promoter bound by CTCF (in a window of +/-500 bp around the TSS). Promoters with CTCF are reported in blue.

12. Line 149 “suggesting either a reduced role for ISWI in the formation and maintenance of nucleosome structure” Reduced compared to what? Do authors simply mean ‘minor’?

R53: We thank the reviewer for pointing out this mistake. We changed “either a reduced” to “a minor.”

13. It would be helpful to clarify if ext fig 4e contains all of the differentially ATAC regions

or just the ones with lost accessibility. Adding the differential ATAC signal as an extra column might be helpful if it ends up lining up with the clusters.

R54: The figure mentioned by the reviewer indeed included all differential ATAC regions and the differential ATAC signal for the same regions in the same order as provided in Fig2d. We thank the reviewer for highlighting the missing specifications now both added in the dedicated figure legend.

14. It would be helpful to report in the text the fraction of affected ATAC regions that have CTCF. Judged from figure 4e it is about 50% - which means that other factors seem to depend on BPTF for creating/maintaining accessibility. Also here it would be most helpful to report in a supplementary table the position of the differentially accessible regions.

R55: We now provide the requested data in the newly added Supplementary Table2. Considering all the differentially accessible regions divided by clusters as displayed in Extended Data Figure 4e, only cluster 2 has clear enrichment for CTCF sites. In fact, 47.5% of cluster 2 regions have a bound CTCF motif in them as rightly expected by the reviewer. This is in stark contrast with cluster 1 and 3, where bound CTCF sites correspond to less than 1% of the regions. This is now specified in the text at line 193.

Regarding the reviewer comment on other transcription factors affected by BPTF depletion we refer to Extended Data Fig. 6c where we took an unbiased approach and performed motif analysis on the regions called as differentially accessible divided by cluster as in figure 2. This analysis reveals that while other motifs are found enriched in clusters 1 and 3, no other motif display an enrichment comparable to CTCF motif within cluster 2 (representing regions with strong accessibility loss in both Snf2h and Bptf KO). We thus conclude that at these sites we do not find evidence at the level of accessibility, for a contribution of other transcription factor motifs comparable to the one of CTCF. This is in line with the fact that many affected sites do not reside in enhancers but in CTCF

only bound DHSs (also see R2).

15. Reporting the expression of CTCF and cohesin regulatory subunits from the RNA-seq would help ascertain that the effects of depleting BPTF are not simply due to mis-expression of these genes.

R56: In response to the reviewer request we now provide gene expression data for CTCF and cohesin complex subunits and regulatory proteins for WT and BPTF depleted lines, which all show limited changes. Moreover, we provide protein quantification by western-blotting of Rad21 in WT and BPTFKO cells in addition to CTCF protein levels already provided in Figure 1b. Both RNA and protein do not show significant difference in the factors examined and we now included these plots as important controls of our study as newly added Extended Data Fig. 12e and f.

16. It would be most useful if authors provided the list of DHSs and differential DHSs they used in a supplementary table.

R57: We now provide the genomic position of DHSs used in the manuscript in the newly added Supplementary Table1. We did not call differential DHSs but differentially accessible peaks which we now provide in Supplementary Table2.

17. Please include a supplementary table with the list of genomic positions in each cluster and group used in figures 4 and 5

R58: Now provided as Supplementary Table3.

18. Please include the mapping statistics for the NGS experiments as a supplementary table (at least final used reads / read-pairs)

R59: Now provided as Supplementary Table5.

19. In Extended figure 7f it would be helpful to have a y axis scale for the gray triangles that denote the log₂FC of CTCF binding in BPTF vs WT

R60: The scale of CTCF binding changes is already provided for each bin in the y axis label we hope this answers the reviewer comment.

20. The “cell line generation and maintenance” section in the method is missing critical information.

a) Authors indicate that several clones with several sgRNAs were generated to account for possible off-targeting. It would be most helpful if authors could clarify how many clones were used for how many experiments, and provide a nomenclature (e.g. a unique ID# for each cell line) enabling them to report which experiment was performed with which clones. A table with a unique ID# for each clone and the experiment(s) it was used for would suffice. This would also facilitate future resource sharing.

b) Authors indicate that separate gRNAs were used to assess off-target and clonal effects yet only one sgRNA sequence per gene target is provided (except for Acf1). Please clarify

R61: We thank the reviewer for highlighting this point which was not clearly explained in the Method section. Separate clones were picked for each line and (where possible) using gRNAs targeting different exons. One representative clone was then selected for each genotype and all experiments presented in the manuscript were carried out on one clone per KO. We specified this point in the Methods section at line 498. As the reviewer correctly points out, we are showing two gRNAs sequences for Acf1. This is because both were in fact used to generate the Acf1 KO line used in the study, in order to delete a second alternative isoform that was still expressed following the use of a single gRNA.

This clarification was now also added in the method section. The sequence of gRNAs used for obtaining all the lines presented in the study are reported in the same section.

c) It would be helpful if authors provided more technical details such as amount of cell and DNA transfected, which vector(s) carried the puromycin selection whether single colonies were obtained by FACS or limited dilution + manual picking etc...

R62: We thank the reviewer for highlighting the missing information which we are now providing in the Methods section from line 485.

d) Authors need to report the sequence of the mutated alleles for gene in each clone used throughout the paper

R63: As detailed in point R61, for each gene deletion the same clone is used for all the experiments throughout the study. We have now updated the Methods section and added Supplementary Table4 reporting the genotype details for each cell lines used.

e) Transfections needs more detailed protocol for reproducibility (e.g. cell number, plating scheme, DNA and lipofectamine concentrations etc. etc. etc.). Right now it would be impossible to reproduce the experiment with the information provided.

R64: As described in R62 we now expanded this part of the Methods including the information required by the reviewer.

21. Tip5 KO was reported to be lethal to mESCs grown in 2i+LIF medium – but not in serum+LIF (Dalcher santoro biorxiv 2019). I wonder if authors reproduce this phenotype

and how the various KOs reported here behave in 2i+LIF. This may unmask essential roles for a specific ISWI subcomplexes and clarify why these subunits are only essential in specific growth conditions. These experiments may be left out for a follow-up study.

R65: We thank the reviewer for highlighting the previous study by Dalcher et al, 2020. We did not test our lines in other conditions than the one described in the manuscript (serum+LIF) and in these settings the TIP5 KO indeed did not show a strong phenotype. Considering the previous study cited by the reviewer (which we now cite at line 146), it would be an interesting starting point for follow-up experiments to test if the ISWI mutants show different phenotypes under different conditions.

Decision Letter, first revision:

29th Jan 2024

Dear Professor Schübeler,

Your Article, "Systematic assessment of ISWI subunits reveals that NURF creates local accessibility for proper CTCF function" has now been seen by 3 referees. You will see from their comments below that while they find your work of interest, some important points are raised. We are interested in the possibility of publishing your study in Nature Genetics, but would like to consider your response to these concerns in the form of a revised manuscript before we make a final decision on publication.

To guide the scope of the revisions, the editors discuss the referee reports in detail within the team, including with the chief editor, with a view to identifying key priorities that should be addressed in revision and sometimes overruling referee requests that are deemed beyond the scope of the current study. In this case, we kindly ask you to address Reviewers' comments in full. Notably, we strongly invite you to change the title according to the Reviewers' suggestions, to tone down all the statements that allude to a strong dispensability of the ISWI subunit for CTCF function and openly state, throughout the manuscript, that the effect of BPTF on 3D genome organization is subtle. We hope that you will find the prioritized set of referee points to be useful when revising your study. Please do not hesitate to get in touch if you would like to discuss these issues further.

We therefore invite you to revise your manuscript taking into account all reviewer and editor comments. Please highlight all changes in the manuscript text file. At this stage we will need you to upload a copy of the manuscript in MS Word .docx or similar editable format.

*2) If you have not done so already please begin to revise your manuscript so that it conforms to our Article format instructions, available here.
Refer also to any guidelines provided in this letter.

Please be aware of our guidelines on digital image standards.

[redacted]

We hope to receive your revised manuscript within four to eight weeks. If you cannot send it within this time, please let us know.

Sincerely,

Chiara Anania, PhD
Associate Editor
Nature Genetics
<https://orcid.org/0000-0003-1549-4157>

Referee expertise:

Referee #1:

Referee #2:

Referee #3:

Reviewers' Comments:

Reviewer #1:

Remarks to the Author:

The authors commendably tried their best to address all my comments.

Unfortunately, there is no getting around the fact that the effect sizes are still minuscule, calling into question the relevance of the findings, and whether the advance of this (carefully done) study over the author's previous paper on this subject merits publication in Nature Genetics.

For example, Figs.4c,d the scales cover a sliver of a window, e.g. 0.85-1.2 fold and the differences between WT and BPTFΔ are still barely noticeable. The effects on loops in Fig.4e are essentially null, which may not be surprising as RAD21 and WAPL changes at these CTCF sites might cancel each other. Similar for Fig.4f cluster 2, Fig.5b (WT/BPTFΔ), Extended data Fig.10f, g, h.

Therefore, I cannot help reaffirming that the title of this paper that NURF is required for proper CTCF function is technically correct, but at the face of it, one could also state that BPTF is dispensable for CTCF function.

Additional points:

1. The Discussion attributes the changes in CTCF function in BPTFΔ cells to reduced RAD21 but does not consider a similar loss of WAPL.
2. R8: The answer to my previous question 6 (Fig.3) about whether CTCF binding to chromatin changes in SNF2HΔ cells even at sites not co-bound by the two. Just to clarify, are the authors saying that there are no CTCF sites that are not also bound by SNF2H?

Reviewer #2:

Remarks to the Author:

The authors have addressed all my concerns.

Reviewer #3:

Remarks to the Author:

Thank you to the authors for extensive new analyses. Most of my suggestions were addressed and I find the effect sizes much easier to appreciate with the new figures.

I still find the title of the manuscript somewhat obscure "Systematic assessment of ISWI subunits reveals that NURF creates local accessibility for proper CTCF FUNCTION".

Authors report the novel discovery that in BPTF mutants CTCF binding is diminished about 2-fold at about 60% of sites (according to figure 4), with resulting mild but statistically reproducible Hi-C changes but only where CTCF binding is altered. The 22% of sites that depend on BPTF for accessibility but not for CTCF binding do not exhibit pronounced Hi-C defects (cluster 2 in figure 4). Therefore CTCF binding and insulation remain coupled in the mutant conditions presented. And given the very small effects on Hi-C it would seem unjustified to stress in the title the role of NURF in CTCF function (as in, 3D genome folding). Therefore it seems that the novelty of the article resides in the discovery of a subtle but perceptible contribution of NURF in CTCF binding – not so much that alterations in CTCF binding result in altered Hi-C patterns.

Therefore, I feel "Systematic assessment of ISWI subunits reveals that NURF creates local accessibility for proper CTCF BINDING" may better reflect the main novelty of the study.

Minor additional suggestions

1. R34: In the abstract the proposed modification "it only modestly impacts CTCF binding itself. at a class of sites" sounds confusing. Maybe something along the lines of "it only modestly impacts CTCF binding itself, and only at about half the sites" would be more explicit (given the numbers in new supp fig. 8)?
2. Figure 4 legend panel e "same analysis at CTCF loops". Please clarify what are "CTCF loops"? If these are Hi-C loops called by Bonev et al it would be better to just say "Hi-C loop". If CTCF binding was somehow intersected to call these specifically "CTCF loops" please briefly explain how this subset of Hi-C loops was obtained in the legend

Author Rebuttal, first revision:

NG-A62885R Point-by-point response to editor's and reviewer's comments:

We very much appreciated the positive responses by all reviewers to our revisions and the amount and quality of the additional experiments and analysis.

Remaining requests were to reconsider the title and to identify potential overstatements on the effect size on CTCF function. We have edited the manuscript accordingly and furthermore suggest changing the title to "Systematic assessment of ISWI subunits reveals that NURF creates local accessibility for CTCF". This captures, in our view, the main observation of the drastic effect on accessibility and allows to provide the needed detailed context on binding and other consequences in the text itself.

We hope the reviewers agree to these edits and that the manuscript is now acceptable. Please find below comments to each specific point.

Reviewers' Comments:

Reviewer #1:

Remarks to the Author:

The authors commendably tried their best to address all my comments.

Unfortunately, there is no getting around the fact that the effect sizes are still minuscule, calling into question the relevance of the findings, and whether the advance of this (carefully done) study over the author's previous paper on this subject merits publication in Nature Genetics.

For example, Figs.4c,d the scales cover a sliver of a window, e.g. 0.85-1.2 fold and the differences between WT and BPTF Δ are still barely noticeable. The effects on loops in Fig.4e are essentially null, which may not be surprising as RAD21 and WAPL changes at these CTCF sites might cancel each other. Similar for Fig.4f cluster 2, Fig.5b (WT/BPTF Δ), Extended data Fig.10f, g, h.

R1. We thank the reviewer for acknowledging that we addressed all comments and that our study is done carefully. While it is inherently difficult to define actual advance, we would like to stress that we phenotype and compare all ISWI subcomplexes, a total of eight remodeler mutants (seven of which generated for this study). Only this comprehensive approach identifies the specificity of NURF function for CTCF in addition to revealing the unexpected and novel observation of a separation of binding from remodeler dependent accessibility with coinciding reduction of cohesin and its unloading factor and detectable effects on nuclear organization.

Therefore, I cannot help reaffirming that the title of this paper that NURF is required for proper CTCF function is technically correct, but at the face of it, one could also state that BPTF is dispensable for CTCF function.

R2. While our previous title appears technically correct, we nevertheless changed it to avoid a false expectation and now suggest:

"Systematic assessment of ISWI subunits reveals that NURF creates local accessibility for CTCF". This places the focus on our main observation of the drastic effect on accessibility observed only for NURF among all new ISWI mutants as also outlined above.

Additional points:

1. The Discussion attributes the changes in CTCF function in BPTF Δ cells to reduced RAD21 but does not consider a similar loss of WAPL.

R3. While WAPL depletion can lead to cohesin loss at cell-type specific active chromatin regions, it leads to *increased* cohesin stalling at CTCF sites (Liu et al., Nature Genetics 2021). We do not observe increased cohesin occupancy at these sites, or a structural phenotype that could be

associated with WAPL loss and expected increased occupancy of cohesin (e.g. increased stripe and long-range CTCF-peak signal). We thus favor the interpretation that decreased WAPL occupancy is a consequence of cohesin loss at CTCF sites that lose chromatin accessibility in BPTF Δ cells.

We have now edited the discussion in respect to the reviewer comment at page 25:

“Since this coincides with partial loss of binding of both cohesin, and as a result of its release factor WAPL, it is tempting to speculate that proper cohesin accumulation at CTCF sites requires chromatin opening mediated by NURF”.

2. R8: The answer to my previous question 6 (Fig.3) about whether CTCF binding to chromatin changes in SNF2H Δ cells even at sites not co-bound by the two. Just to clarify, are the authors saying that there are no CTCF sites that are not also bound by SNF2H?

R4. It was not evident for us that the reviewer asked for comparison of CTCF and SNF2H binding in Snf2h Δ cells, which we now added in the plot below. The SNF2H and CTCF signals correlate in WT conditions and these sites are globally affected by loss of the ATPase as is evident from this plot. However, the limited dynamic range of the remodeler ChIP as shown in Extended Data Fig.7g does not allow to rigorously discriminate bound from unbound sites to be compared with loss of TF binding. In turn, the low remodeler signal to noise-ratio also precludes to state that they always co-bind, which was an oversight in the rebuttal. While this was never stated in the manuscript, where the limited dynamic range of remodelers localization is evident throughout the figures, we apologize for the misleading wording in the previous answer.

SNF2H and CTCF binding changes upon SNF2H depletion

Snf2h and CTCF localization (assessed by CUT&RUN and ChIP-seq respectively) shown as single locus heatmap at all bound CTCF sites in WT and upon depletion of SNF2H. The black arrow indicates the position of CTCF motif.

Reviewer #2:

Remarks to the Author:

The authors have addressed all my concerns.

R5. We are very pleased to see that we addressed all their concerns.

Reviewer #3:

Remarks to the Author:

Thank you to the authors for extensive new analyses. Most of my suggestions were addressed and I find the effect sizes much easier to appreciate with the new figures.

R6. We thank the reviewer for this positive feedback and again for the insightful suggestions in the first review that guided our revisions.

I still find the title of the manuscript somewhat obscure “Systematic assessment of ISWI subunits reveals that NURF creates local accessibility for proper CTCF FUNCTION”.

Authors report the novel discovery that in BPTF mutants CTCF binding is diminished about 2-fold at about 60% of sites (according to figure 4), with resulting mild but statistically reproducible Hi-C changes but only where CTCF binding is altered. The 22% of sites that depend on BPTF for accessibility but not for CTCF binding do not exhibit pronounced Hi-C defects (cluster 2 in figure 4). Therefore CTCF binding and insulation remain coupled in the mutant conditions presented.

And given the very small effects on Hi-C it would seem unjustified to stress in the title the role of NURF in CTCF function (as in, 3D genome folding). Therefore it seems that the novelty of the article resides in the discovery of a subtle but perceptible contribution of NURF in CTCF binding – not so much that alterations in CTCF binding result in altered Hi-C patterns.

Therefore, I feel “Systematic assessment of ISWI subunits reveals that NURF creates local accessibility for proper CTCF BINDING” may better reflect the main novelty of the study.

R7. In our view the key finding of our comprehensive study is the function of NURF in creating accessibility at CTCF sites. Given that this has a highly site specific effect on

binding of CTCF and in order not to simplify, we suggest changing the title to “Systematic assessment of ISWI subunits reveals that NURF creates local accessibility for CTCF” as outlined above.

Minor additional suggestions

1. R34: In the abstract the proposed modification “it only modestly impacts CTCF binding itself. at a class of sites” sounds confusing. Maybe something along the lines of “it only modestly impacts CTCF binding itself, and only at about half the sites” would be more explicit (given the numbers in new supp fig. 8)?

R8. In response to the reviewer comment we modified the related part in the abstract. We hope this made the abstract clearer.

2. Figure 4 legend panel e “same analysis at CTCF loops”. Please clarify what are “CTCF loops”? If these are Hi-C loops called by Bonev et al it would be better to just say “Hi-C loop”. If CTCF binding was somehow intersected to call these specifically “CTCF loops” please briefly explain how this subset of Hi-C loops was obtained in the legend

R9. Indeed Figure 4e shows Hi-C loops called in the Bonev et al. dataset, this has now been corrected in the figure legend (Fig.4 and Extended Data Fig.10).

Decision Letter, second revision:

9th Feb 2024

Dear Dr. Schübeler,

Thank you for submitting your revised manuscript "Systematic assessment of ISWI subunits reveals that NURF creates local accessibility for CTCF" (NG-A62885R1). We find that the paper has improved in revision, and therefore we'll be happy in principle to publish it in Nature Genetics, pending minor revisions to comply with our editorial and formatting guidelines.

Congratulations!

Sincerely,
Chiara

Chiara Anania, PhD
Associate Editor
Nature Genetics
<https://orcid.org/0000-0003-1549-4157>

Final Decision Letter:

23rd Apr 2024

Dear Dr. Schübeler,

I am delighted to say that your manuscript "Systematic assessment of ISWI subunits shows that NURF creates local accessibility for CTCF" has been accepted for publication in an upcoming issue of Nature Genetics.

Your paper will be published online after we receive your corrections and will appear in print in the next available issue. You can find out your date of online publication by contacting the Nature Press Office (press@nature.com) after sending your e-proof corrections.

Please note that *Nature Genetics* is a Transformative Journal (TJ). Authors may publish their research with us through the traditional subscription access route or make their paper immediately open access through payment of an article-processing charge (APC). Authors will not be required to make a final decision about access to their article until it has been accepted. Find out more about Transformative Journals

Authors may need to take specific actions to achieve compliance with funder and institutional open access mandates. If your research is supported by a funder that requires immediate open access (e.g. according to Plan S principles) then you should select the gold OA route, and we will direct you to the compliant route where possible. For authors selecting the subscription publication route, the journal's standard licensing terms will need to be accepted, including [a href="https://www.nature.com/nature-portfolio/editorial-policies/self-archiving-and-license-to-publish"](https://www.nature.com/nature-portfolio/editorial-policies/self-archiving-and-license-to-publish). Those licensing terms will supersede any other terms that the author or any third party may assert apply to any version of the manuscript.

If you have not already done so, we invite you to upload the step-by-step protocols used in this manuscript to the Protocols Exchange, part of our on-line web resource, natureprotocols.com. If you complete the upload by the time you receive your manuscript proofs, we can insert links in your article that lead directly to the protocol details. Your protocol will be made freely available upon publication of your paper. By participating in natureprotocols.com, you are enabling researchers to more readily reproduce or adapt the methodology you use. [Natureprotocols.com](http://natureprotocols.com) is fully searchable, providing your protocols and paper with increased utility and visibility. Please submit your protocol to <https://protocolexchange.researchsquare.com/>. After entering your nature.com username and password you will need to enter your manuscript number (NG-A62885R2). Further information can be found at <https://www.nature.com/nature-portfolio/editorial-policies/reporting-standards#protocols>

Sincerely,
Chiara

Chiara Anania, PhD
Associate Editor
Nature Genetics
<https://orcid.org/0000-0003-1549-4157>